# PINNacle: A Comprehensive Benchmark of Physics-Informed Neural Networks for Solving PDEs

**Zhongkai Hao**[1]*, **Jiachen Yao**[1]*, **Chang Su**[1]*, **Hang Su**[1], **Ziao Wang**[1],

**Fanzhi Lu**[1], **Zeyu Xia**[1], **Yichi Zhang**[1], **Songming Liu**[1], **Lu Lu**[2], **Jun Zhu**[1]†

[1]Dept. of Comp. Sci. and Tech., Institute for AI, BNRist Center, THBI Lab,
Tsinghua-Bosch Joint ML Center, Tsinghua University
[2] Department of Statistics and Data Science, Yale University
hzj21@mails.tsinghua.edu.cn;    dcszj@tsinghua.edu.cn

## Abstract

While significant progress has been made on Physics-Informed Neural Networks (PINNs), a comprehensive comparison of these methods across a wide range of Partial Differential Equations (PDEs) is still lacking. This study introduces PINNacle, a benchmarking tool designed to fill this gap. PINNacle provides a diverse dataset, comprising over 20 distinct PDEs from various domains, including heat conduction, fluid dynamics, biology, and electromagnetics. These PDEs encapsulate key challenges inherent to real-world problems, such as complex geometry, multi-scale phenomena, nonlinearity, and high dimensionality. PINNacle also offers a user-friendly toolbox, incorporating about 10 state-of-the-art PINN methods for systematic evaluation and comparison. We have conducted extensive experiments with these methods, offering insights into their strengths and weaknesses. In addition to providing a standardized means of assessing performance, PINNacle also offers an in-depth analysis to guide future research such as domain decomposition methods and loss reweighting for handling multi-scale problems. To the best of our knowledge, it is the largest benchmark with a diverse and comprehensive evaluation that will undoubtedly foster further research in PINNs.

## 1 Introduction

Partial Differential Equations (PDEs) are of paramount importance in science and engineering, as they often underpin our understanding of intricate physical systems such as fluid flow, heat transfer, and stress distribution [37]. The computational simulation of PDE systems has been a focal point of research for an extensive period, leading to the development of numerical methods such as finite difference [6], finite element [45], and finite volume methods [14].

Recent advancements have led to the use of deep neural networks to solve forward and inverse problems involving PDEs [44, 62, 11, 54]. Among these, Physics-Informed Neural Networks (PINNs) have emerged as a promising alternative to traditional numerical methods in solving such problems [44, 25]. PINNs leverage the underlying physical laws and available data to effectively handle various scientific and engineering applications. The growing interest in this field has spurred the development of numerous PINN variants, each tailored to overcome specific challenges or to enhance the performance of the original framework.

---

*Equal contribution.    †The corresponding author.

38th Conference on Neural Information Processing Systems (NeurIPS 2024) Track on Datasets and Benchmarks.

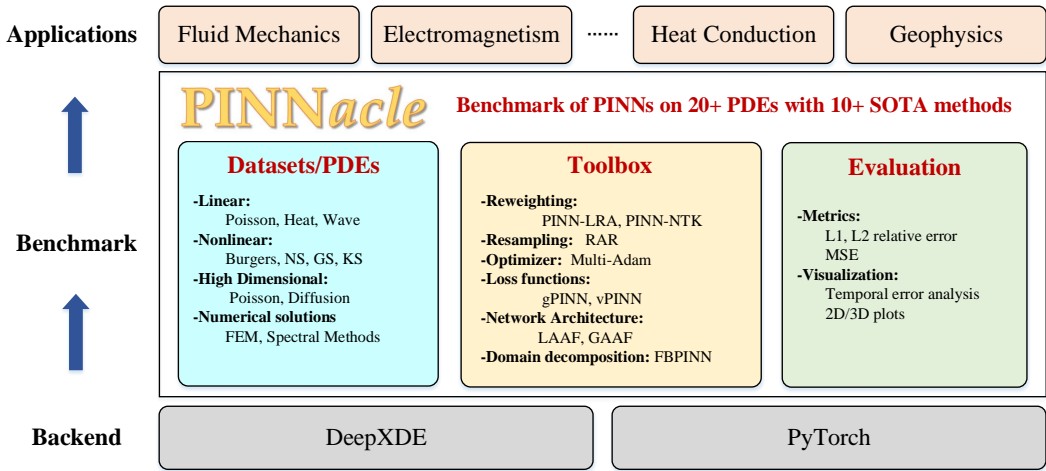

Figure 1: Architecture of PINNacle. It contains a dataset covering more than 20 PDEs, a toolbox that implements about 10 SOTA methods, and an evaluation module. These methods have a wide range of application scenarios like fluid mechanics, electromagnetism, heat conduction, geophysics, and so on.

While PINN methods have achieved remarkable progress, a comprehensive comparison of these methods across diverse types of PDEs is currently lacking. Establishing such a benchmark is crucial as it could enable researchers to more thoroughly understand existing methods and pinpoint potential challenges. Despite the availability of several studies comparing sampling methods [60] and reweighting methods [2], there has been no concerted effort to develop a rigorous benchmark using challenging datasets from real-world problems. The sheer variety and inherent complexity of PDEs make it difficult to conduct a comprehensive analysis. Moreover, different mathematical properties and application scenarios further complicate the task, requiring the benchmark to be adaptable and exhaustive.

To resolve these challenges, we propose PINNacle, a comprehensive benchmark for evaluating and understanding the performance of PINNs. As shown in Fig. 1, PINNacle consists of three major components — a diverse dataset, a toolbox, and evaluation modules. The dataset comprises tasks from over 20 different PDEs from various domains, including heat conduction, fluid dynamics, biology, and electromagnetics. Each task brings its own set of challenges, such as complex geometry, multi-scale phenomena, nonlinearity, and high dimensionality, thus providing a rich testing ground for PINNs. The toolbox incorporates more than 10 state-of-the-art (SOTA) PINN methods, enabling a systematic comparison of different strategies, including loss reweighting, variational formulation, adaptive activations, and domain decomposition. These methods can be flexibly applied to the tasks in the dataset, offering researchers a convenient way to evaluate the performance of PINNs which is also user-friendly for secondary development. The evaluation modules provide a standardized means of assessing the performance of different PINN methods across all tasks, ensuring consistency in comparison and facilitating the identification of strengths and weaknesses in various methods.

PINNacle provides a robust, diverse, and comprehensive benchmark suite for PINNs, contributing significantly to the field's understanding and application. It represents a major step forward in the evolution of PINNs which could foster more innovative research and development in this exciting field. Code and data are publicly available at `https://github.com/i207M/PINNacle`.

In a nutshell, our contributions can be summarized as follows:

- We design a dataset encompassing over 20 challenging PDE problems. These problems encapsulate several critical challenges faced by PINNs, including handling complex geometries, multi-scale phenomena, nonlinearity, and high-dimensional problems.

- We systematically evaluate more than 10 carefully selected representative variants of PINNs. We conducted thorough experiments and ablation studies to evaluate their performance. To the best of our knowledge, this is the largest benchmark comparing different PINN variants.

- We provide an in-depth analysis to guide future research. We show using loss reweighting and domain decomposition methods could improve the performance on multi-scale and complex geometry problems. Variational formulation achieves better performance on inverse problems. However, few methods can adequately address nonlinear problems, indicating a future direction for exploration and advancement.

## 2 Related Work

### 2.1 Benchmarks and datasets in scientific machine learning

The growing trend of AI in scientific research has stimulated the development of various benchmarks and datasets, which differ greatly in data formats, sizes, and governing principles. For instance, [33] presents a benchmark for comparing neural operators, while [3, 41] benchmarks methods for learning latent Newtonian mechanics. Furthermore, domain-specific datasets and benchmarks exist in fluid mechanics [20], climate science [42, 5], quantum chemistry [1], and biology [4].

Beyond these domain-specific datasets and benchmarks, physics-informed machine learning has received considerable attention [18, 9] since the advent of Physics-Informed Neural Networks (PINNs) [44]. These methods successfully incorporate physical laws into model training, demonstrating immense potential across a variety of scientific and engineering domains. Various papers have compared different components within the PINN framework; for instance, [10] and [60] investigate the sampling methods of collocation points in PINNs, and [2] compare reweighting techniques for different loss components. PDEBench [52] and PDEArena [17] design multiple tasks to compare different methods in scientific machine learning such as PINNs, FNO, and U-Net. Nevertheless, a comprehensive comparison of various PINN approaches remains absent in the literature.

### 2.2 Softwares and Toolboxes

A plethora of software solutions have been developed for solving PDEs with neural networks. These include SimNet [19], NeuralPDE [43], TorchDiffEq [8], and PyDEns [29]. More recently, DeepXDE [34] has been introduced as a fundamental library for implementing PINNs across different backends. However, there remains a void for a toolbox that provides a unified implementation for advanced PINN variants. Our PINNacle fills this gap by offering a flexible interface that facilitates the implementation and evaluation of diverse PINN variants. We furnish clear and concise code for researchers to execute benchmarks across all problems and methods.

### 2.3 Variants of Physics-informed neural networks

The PINNs have received much attention due to their remarkable performance in solving both forward and inverse PDE problems. However, vanilla PINNs have many limitations. Researchers have proposed numerous PINN variants to address challenges associated with high-dimensionality, non-linearity, multi-scale issues, and complex geometries [18, 9, 25, 30]. Broadly speaking, these variants can be categorized into: loss reweighting/resampling [57, 58, 53, 60, 40], innovative optimizers [61], novel loss functions such as variational formulations [62, 26, 27, 28] or regularization terms [63, 50], and novel architectures like domain decomposition [21, 31, 38, 24] and adaptive activations [23, 22]. These variants have enhanced PINN's performance across various problems. Here we select representative methods from each category and conduct a comprehensive analysis using our benchmark dataset to evaluate these variants.

## 3 PINNacle: A Hierarchical Benchmark for PINNs

In this section, we first introduce the preliminaries of PINNs. Then we introduce the details of datasets (tasks), PINN methods, the toolbox framework, and the evaluation metrics.

### 3.1 Preliminaries of Physics-informed Neural Networks

Physics-informed neural networks are neural network-based methods for solving PDEs as well as inverse problems of PDEs, which have received much attention recently. Specifically, let's consider a

general Partial Differential Equation (PDE) system defined on $\Omega$, which can be represented as:

$$\mathcal{F}(u(x); x) \quad = \quad 0, \quad x \in \Omega, \tag{1}$$
$$\mathcal{B}(u(x); x) \quad = \quad 0, \quad x \in \partial\Omega. \tag{2}$$

where $\mathcal{F}$ is a differential operator and $\mathcal{B}$ is the boundary/initial condition. PINN uses a neural network $u_\theta(x)$ with parameters $\theta$ to approximate $u(x)$. The objective of PINN is to minimize the following loss function:

$$\mathcal{L}(\theta) = \frac{w_c}{N_c} \sum_{i=1}^{N_c} ||\mathcal{F}(u_\theta(x_c^i); x_c^i)||^2 + \frac{w_b}{N_b} \sum_{i=1}^{N_b} ||\mathcal{B}(u_\theta(x_b^i); x_b^i)||^2 + \frac{w_d}{N_d} \sum_{i=1}^{N_d} ||u_\theta(x_d^i) - u(x_d^i)||^2. \tag{3}$$

where $w_c, w_b, w_d$ are weights. The first two terms enforce the PDE constraints on $\{x_c^i\}_{1...N_c}$ and boundary conditions on $\{x_b^i\}_{1...N_b}$. The last term is data loss, which is optional when there is data available. However, PINNs have several inherent drawbacks. First, PINNs optimize a mixture of imbalance loss terms which might hinder its convergence as illustrated in [57]. Second, nonlinear or stiff PDEs might lead to unstable optimization [58]. Third, the vanilla MLPs might have difficulty in representing multi-scale or high-dimensional functions. For example, [30] shows that vanilla PINNs only work for a small parameter range, even in a simple convection problem. To resolve these challenges, numerous variants of PINNs are proposed. However, a comprehensive comparison of these methods is lacking, and thus it is imperative to develop a benchmark.

## 3.2 Datasets

To effectively compare PINN variants, we've curated a set of PDE problems (datasets) representing a wide range of challenges. We chose PDEs from diverse domains, reflecting their importance in science and engineering. Our dataset includes 22 unique cases, with further details in Appendix B.

- The **Burgers' Equation**, fundamental to fluid mechanics, considering both one and two-dimensional problems.
- The **Poisson's Equation**, widely used in math and physics, with four different cases.
- The **Heat Equation**, a time-dependent PDE that describes diffusion or heat conduction, demonstrated in four unique cases.
- The **Navier-Stokes Equation**, describing the motion of viscous fluid substances, showcased in three scenarios: a lid-driven flow (NS2d-C), a geometrically complex backward step flow (NS2d-CG), and a time-dependent problem (NS2d-LT).
- The **Wave Equation**, modeling wave behavior, exhibited in three cases.
- **Chaotic PDEs**, featuring two popular examples: the Gray-Scott (GS) and Kuramoto-Sivashinsky (KS) equations.
- **High Dimensional PDEs**, including the high-dimensional Poisson equation (PNd) and the high-dimensional diffusion or heat equation (HNd).
- **Inverse Problems**, focusing on the reconstruction of the coefficient field from noisy data for the Poisson equation (PInv) and the diffusion equation (HInv).

It is important to note that we have chosen PDEs encompassing a wide range of mathematical properties. This ensures that the benchmarks do not favor a specific type of PDE. The selected PDE problems introduce several core challenges, which include:

- **Complex Geometry**: Many PDE problems involve complex or irregular geometry, such as heat conduction or wave propagation around obstacles. These complexities pose significant challenges for PINNs in terms of accurate boundary behavior representation.
- **Multi-Scale Phenomena**: Multi-scale phenomena, where the solution varies significantly over different scales, are prevalent in situations such as turbulent fluid flow. Achieving a balanced representation across all scales is a challenge for PINNs in multi-scale scenarios.
- **Nonlinear Behavior**: Many PDEs exhibit nonlinear or even chaotic behavior, where minor variations in initial conditions can lead to substantial divergence in outcomes. The optimization of PINNs becomes intriguing on nonlinear PDEs.

- **High Dimensionality**: High-dimensional PDE problems, frequently encountered in quantum mechanics, present significant challenges for PINNs due to the "curse of dimensionality". This term refers to the increase in computational complexity with the addition of each dimension, accompanied by statistical issues like data sparsity in high-dimensional space.

These challenges are selected due to their frequent occurrence in numerous real-world applications. As such, a method's performance in addressing these challenges serves as a reliable indicator of its overall practical utility. Table 1 presents a detailed overview of the dataset, the PDEs, and the challenges associated with these problems. We generate data using FEM solver provided by COMSOL 6.0 [39] for problems with complex geometry and spectral method provided by Chebfun [12] for chaotic problems. More details can be found in Appendix B.

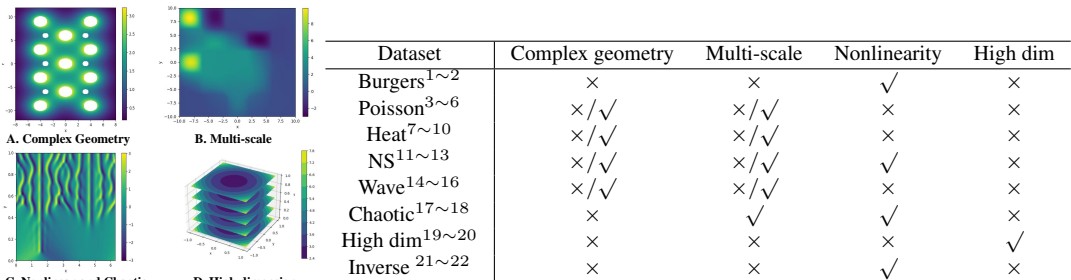

| Dataset | Complex geometry | Multi-scale | Nonlinearity | High dim |
|---|---|---|---|---|
| Burgers[1~2] | $\times$ | $\times$ | $\checkmark$ | $\times$ |
| Poisson[3~6] | $\times/\checkmark$ | $\times/\checkmark$ | $\times$ | $\times$ |
| Heat[7~10] | $\times/\checkmark$ | $\times/\checkmark$ | $\times$ | $\times$ |
| NS[11~13] | $\times/\checkmark$ | $\times/\checkmark$ | $\checkmark$ | $\times$ |
| Wave[14~16] | $\times/\checkmark$ | $\times/\checkmark$ | $\times$ | $\times$ |
| Chaotic[17~18] | $\times$ | $\checkmark$ | $\checkmark$ | $\times$ |
| High dim[19~20] | $\times$ | $\times$ | $\times$ | $\checkmark$ |
| Inverse [21~22] | $\times$ | $\times$ | $\checkmark$ | $\times$ |

A. Complex Geometry  B. Multi-scale  C. Nonlinear and Chaotic  D. High dimension

Table 1: Overview of our datasets along with their challenges. We chose 22 cases in total to evaluate the methods of PINNs. The left picture shows the visualization of cases with these four challenges, i.e., complex geometry, multi-scale, nonlinearity, and high dimension.

## 3.3 Methods and Toolbox

After conducting an extensive literature review, we present an overview of diverse PINNs approaches for comparison. Then we present the high-level structure of our PINNacle.

### 3.3.1 Methods

As mentioned above, variants of PINNs are mainly based on loss functions, architecture, and optimizer [18]. The modifications to loss functions can be divided into reweighting existing losses and developing novel loss functions like regularization and variational formulation. Variants of architectures include using domain decomposition and adaptive activations.

The methods discussed are directly correlated with the challenges highlighted in Table 1. For example, domain decomposition methods are particularly effective for problems involving complex geometries and multi-scale phenomena. Meanwhile, loss reweighting strategies are adept at addressing imbalances in problems with multiple losses. We have chosen variants from these categories based on their significant contributions to the field.

Here, we list the primary categories and representative methods as summarized in Table 2:

- **Loss reweighting/Resampling (2~4)**: PINNs are trained with a mixed loss of PDE residuals, boundary conditions, and available data losses shown in Eq 3. Various methods [57, 59, 2, 35, 47] propose different strategies to adjust these weights $w_c$, $w_b$ and $w_d$ at different epochs or resample collocation points $\{x_c^i\}$ and $\{x_b^i\}$ in Eq 3, which indirectly adjust the weights [60, 40]. We choose three famous examples, i.e., reweighting using gradient norms (PINN-LRA) [57], using neural tangent kernel (PINN-NTK) [59], and residual-based resampling (RAR)[34, 60].
- **Novel optimizer (5)**: To handle the problem of multi-scale objectives, some new optimizers [32, 61] are proposed. We chose MultiAdam, which is resistant to domain scale changes.
- **Novel loss functions (6~7)**: Some works introduce novel loss functions like variational formulation [49, 28, 27] and regularization terms to improve training. We choose hp-VPINN [27] and gPINN [63, 50], which are representative examples from these two categories.

| | Complex Geometry | Multi-scale | Nonlinearity | High dim |
|---|:---:|:---:|:---:|:---:|
| Vanilla PINN [1] | × | × | × | × |
| Reweighting/Resampling[2~4] | √ | √ | × | × |
| Novel Optimizer[7] | × | √ | × | × |
| Novel Loss Functions[5~6] | × | × | × | × |
| Novel Architecture[8~10] | √ | √ | × | × |

Table 2: Overview of methods in our PINNacle. √ denotes the method is potentially designed to solve or show empirical improvements for problems encountering the challenge and vice versa.

- **Novel activation architectures (8~10)**: Some works propose various network architectures, such as using CNN and LSTM [64, 15, 46], custom activation functions [22, 23], and domain decomposition [21, 48, 24, 38]. Among adaptive activations for PINNs, we choose LAAF [22] and GAAF [23]. Domain decomposition is a method that divides the whole domain into multiple subdomains and trains subnetworks on these subdomains. It is helpful for solving multi-scale problems, but multiple subnetworks increase the difficulty of training. XPINNs, cPINNs, and FBPINNs [21, 24, 38] are three representative examples. We choose FBPINNs which is the state-of-the-art domain decomposition that applies domain-specific normalization to stabilize training.

### 3.3.2 Structure of Toolbox

We provide a user-friendly and concise toolbox for implementing, training, and evaluating diverse PINN variants. Specifically, our codebase is based on DeepXDE and provides a series of encapsulated classes and functions to facilitate high-level training and custom PDEs. These utilities allow for a standardized and streamlined approach to the implementation of various PINN variants and PDEs. Moreover, we provided many auxiliary functions, including computing different metrics, visualizing predictions, and recording results.

Despite the unified implementation of diverse PINNs, we also design an adaptive multi-GPU parallel training framework to enhance the efficiency of systematic evaluations of PINN methods. It addresses the parallelization phase of training on multiple tasks, effectively balancing the computational loads of multiple GPUs. It allows for the execution of larger and more complex tasks. In a nutshell, we provide an example code for training and evaluating PINNs on two Poisson equations using our PINNacle framework in Appendix D.

### 3.4 Evaluation

To comprehensively analyze the discrepancy between the PINN solutions and the true solutions, we adopt multiple metrics to evaluate the performance of the PINN variants. Generally, we choose several metrics that are commonly used in literature that apply to all methods and problems. We suppose that $\boldsymbol{y} = (y_i)_{i=1}^n$ is the prediction and $\boldsymbol{y}' = (y_i')_{i=1}^n$ to is ground truth, where $n$ is the number of testing examples. Specifically, we use $\ell_2$ relative error (L2RE), and $\ell_1$ relative error (L1RE) which are two most commonly used metrics to measure the global quality of the solution,

$$\text{L2RE} = \sqrt{\frac{\sum_{i=1}^n (y_i - y_i')^2}{\sum_{i=1}^n y_i'^2}}, \ \text{L1RE} = \frac{\sum_{i=1}^n |y_i - y_i'|}{\sum_{i=1}^n |y_i'|}. \tag{4}$$

We also compute max error (mERR in short), mean square error (MSE), and Fourier error (fMSE) for a detailed analysis of the prediction. These three metrics are computed as follows:

$$\text{MSE} = \frac{1}{n} \sum_{i=1}^n (y_i - y_i')^2, \ \text{mERR} = \max_i |y_i - y_i'|, \ \text{fMSE} = \frac{\sqrt{\sum_{k_{\min}}^{k_{\max}} |\mathcal{F}(\boldsymbol{y}) - \mathcal{F}(\boldsymbol{y}')|^2}}{k_{\max} - k_{\min} + 1}, \tag{5}$$

where $\mathcal{F}$ denotes Fourier transform of $\boldsymbol{y}$ and $k_{\min}, k_{\max}$ are chosen similar to PDEBench [52]. Besides, for time-dependent problems, investigating the quality of the solution with time is important. Therefore we compute the L2RE error varying with time in Appendix E.2.

We assess the performance of PINNs against the reference from numerical solvers. Experimental results utilizing the $\ell_2$ relative error (L2RE) metric are incorporated within the main text, while a more exhaustive set of results, based on the aforementioned metrics, is available in the Appendix E.1.

| L2RE | Name | Vanilla | | | Loss Reweighting/Sampling | | | Optimizer | Loss functions | | Architecture | | |
|---|---|---|---|---|---|---|---|---|---|---|---|---|---|
| – | | PINN | PINN-w | LBFGS | LRA | NTK | RAR | MultiAdam | gPINN | vPINN | LAAF | GAAF | FBPINN |
| Burgers | 1d-C | 1.45E-2 | 2.63E-2 | **1.33E-2** | 2.61E-2 | 1.84E-2 | 3.32E-2 | 4.85E-2 | 2.16E-1 | 3.47E-1 | 1.43E-2 | 5.20E-2 | 2.32E-1 |
| | 2d-C | 3.24E-1 | 2.70E-1 | 4.65E-1 | **2.60E-1** | 2.75E-1 | 3.45E-1 | 3.33E-1 | 3.27E-1 | 6.38E-1 | 2.77E-1 | 2.95E-1 | – |
| Poisson | 2d-C | 6.94E-1 | 3.49E-2 | NaN | 1.17E-1 | **1.23E-2** | 6.99E-1 | 2.63E-2 | 6.87E-1 | 4.91E-1 | 7.68E-1 | 6.04E-1 | 4.49E-2 |
| | 2d-CG | 6.36E-1 | 6.08E-2 | 2.96E-1 | 4.34E-2 | **1.43E-2** | 6.48E-1 | 2.76E-1 | 7.92E-1 | 2.86E-1 | 4.80E-1 | 8.71E-1 | 2.90E-2 |
| | 3d-CG | 5.60E-1 | 3.74E-1 | 7.05E-1 | **1.02E-1** | 9.47E-1 | 5.76E-1 | 3.63E-1 | 4.85E-1 | 7.38E-1 | 5.79E-1 | 5.02E-1 | 7.39E-1 |
| | 2d-MS | 6.30E-1 | 7.60E-1 | 1.45E+0 | 7.94E-1 | 7.48E-1 | 6.44E-1 | **5.90E-1** | 6.16E-1 | 9.72E-1 | 5.93E-1 | 9.31E-1 | 1.04E+0 |
| Heat | 2d-VC | 1.01E+0 | 2.35E-1 | 2.32E-1 | **2.12E-1** | 2.14E-1 | 9.66E-1 | 4.75E-1 | 2.12E+0 | 9.40E-1 | 6.42E-1 | 8.49E-1 | 9.52E-1 |
| | 2d-MS | 6.21E-2 | 2.42E-1 | **1.73E-2** | 8.79E-2 | 4.40E-2 | 7.49E-2 | 2.18E-1 | 1.13E-1 | 9.30E-1 | 7.40E-2 | 9.85E-1 | 8.20E-2 |
| | 2d-CG | 3.64E-2 | 1.45E-1 | 8.57E-1 | 1.25E-1 | 1.16E-1 | 2.72E-2 | 7.12E-2 | 9.38E-2 | – | **2.39E-2** | 4.61E-1 | 9.16E-2 |
| | 2d-LT | 9.99E-1 | 9.99E-1 | 1.00E+0 | 9.99E-1 | 1.00E+0 | 9.99E-1 | 1.00E+0 | 1.00E+0 | 1.00E+0 | 9.99E-1 | 9.99E-1 | 1.01E+0 |
| NS | 2d-C | 4.70E-2 | 1.45E-1 | 2.14E-1 | NaN | 1.98E-1 | 4.69E-1 | 7.27E-1 | 7.70E-2 | 2.92E-1 | **3.60E-2** | 3.79E-2 | 8.45E-2 |
| | 2d-CG | 1.19E-1 | 3.26E-1 | NaN | 3.32E-1 | 2.93E-1 | 3.34E-1 | 4.31E-1 | 1.54E-1 | 9.94E-1 | **8.24E-2** | 1.74E-1 | 8.27E+0 |
| | 2d-LT | 9.96E-1 | 1.00E+0 | 9.70E-1 | 1.00E+0 | 9.99E-1 | 1.00E+0 | 1.00E+0 | 9.95E-1 | 1.73E+0 | 9.98E-1 | 9.99E-1 | 1.00E+0 |
| Wave | 1d-C | 5.88E-1 | 2.85E-1 | NaN | 3.61E-1 | **9.79E-2** | 5.39E-1 | 1.21E-1 | 5.56E-1 | 8.39E-1 | 4.54E-1 | 6.77E-1 | 5.91E-1 |
| | 2d-CG | 1.84E+0 | 1.66E+0 | 1.33E+0 | 1.48E+0 | 2.16E+0 | 1.15E+0 | 1.09E+0 | 8.14E-1 | 7.99E-1 | 8.19E-1 | **7.94E-1** | 1.06E+0 |
| | 2d-MS | 1.34E+0 | 1.02E+0 | 1.37E+0 | 1.02E+0 | 1.04E+0 | 1.35E+0 | 1.01E+0 | 1.02E+0 | 9.82E-1 | 1.06E+0 | 1.06E+0 | 1.03E+0 |
| Chaotic | GS | 3.19E-1 | 1.58E-1 | NaN | 9.37E-2 | 2.16E-1 | 9.46E-2 | 9.37E-2 | 2.48E-1 | 1.16E+0 | 9.47E-2 | 9.46E-2 | **7.99E-2** |
| | KS | 1.01E+0 | 9.86E-1 | NaN | 9.57E-1 | 9.64E-1 | 1.01E+0 | 9.61E-1 | 9.94E-1 | 9.72E-1 | 1.01E+0 | 1.00E+0 | 1.02E+0 |
| High dim | PNd | 3.04E-3 | 2.58E-3 | 4.67E-4 | **4.58E-4** | 4.64E-3 | 3.59E-3 | 3.98E-3 | 5.05E-3 | – | 4.14E-3 | 7.75E-3 | – |
| | HNd | 3.61E-1 | 4.59E-1 | **1.19E-4** | 3.94E-1 | 3.97E-1 | 3.57E-1 | 3.02E-1 | 3.17E-1 | – | 5.22E-1 | 5.21E-1 | – |
| Inverse | PInv | 9.42E-2 | 1.66E-1 | NaN | 1.54E-1 | 1.93E-1 | 9.35E-2 | 1.30E-1 | 8.03E-2 | **2.45E-2** | 1.30E-1 | 2.54E-1 | 8.44E-1 |
| | HInv | 1.57E+0 | 5.26E-2 | NaN | **5.09E-2** | 7.52E-2 | 1.52E+0 | 8.04E-2 | 4.84E+0 | 4.56E-1 | 5.59E-1 | 2.12E-1 | 9.27E-1 |

Table 3: Mean L2RE of different PINN variants on PINNacle. Best results are highlighted in blue and second-places in lightblue. We do not bold any result if errors of all methods are about $100\%$. "NaN" means the method does not converge and "–" means the method is not suitable for the problem.

# 4 Experiments

## 4.1 Main Results

We now present experimental results. Except for the ablation study in Sec 4.3 and Appendix E.2, we use a learning rate of 0.001 and train all models with 20,000 epochs. We repeat all experiments three times and record the mean and std. We run all experiments on a Linux server with 20 Intel(R) Xeon(R) Silver 4210 CPUs @ 2.20GHz and eight NVIDIA GeForce RTX 2080 Ti each with 12 GB GPU memory. All experiments in Table E.1 require a total about 776 GPU hours, which can be completed in about 4 days on our cluster. Table 3 presents the main results for all methods on our tasks and shows their average $\ell_2$ relative errors (with standard deviation results available in Appendix E.1).

**PINN.** We use PINN-w to denote training PINNs with larger boundary weights. The vanilla PINNs struggle to accurately solve complex physics systems, indicating substantial room for improvement. Using an $\ell_2$ relative error (L2RE) of $10\%$ as a threshold for a successful solution, we find that vanilla PINN only solves 10 out of 22 tasks, most of which involve simpler equations (e.g., $1.45\%$ on Burgers-1d-C). They encounter significant difficulties when faced with physics systems characterized by complex geometries, multi-scale phenomena, nonlinearity, and longer time spans. This shows that directly optimizing an average of the PDE losses and initial/boundary condition losses leads to critical issues such as loss imbalance, suboptimal convergence, and limited expressiveness.

**PINN variants.** PINN variants offer approaches to addressing some of these challenges to varying degrees. Methods involving loss reweighting and resampling have shown improved performance in some cases involving complex geometries and multi-scale phenomena (e.g., $1.43\%$ on Poisson-2d-CG). This is due to the configuration of loss weights and sampled collocation points, which adaptively place more weight on more challenging domains during the training process. However, these methods still struggle with Wave equations, Navier-Stokes equations, and other cases with higher dimensions or longer time spans. MultiAdam, a representative of novel optimizers, solves several simple cases and the chaotic GS equation ($9.37\%$), but does not significantly outperform other methods. The new loss term of variational form demonstrates significant superiority in solving inverse problems (*e.g.,* $1.19\%$ on HInv for vPINN), but no clear improvement in fitting error over standard PINN in forward cases. Changes in architecture can enhance expressiveness and flexibility for cases with complex

| L2RE | Name | Vanilla | Loss Reweighting/Sampling | | | Optimizer | Loss functions | | Architecture | | |
|---|---|---|---|---|---|---|---|---|---|---|---|
| – | – | PINN | LRA | NTK | RAR | MultiAdam | gPINN | vPINN | LAAF | GAAF | FBPINN |
| Burgers-P | 2d-C | 4.74E-01 | 4.36E-01 | **4.13E-01** | 4.71E-01 | 4.93E-01 | 4.91E-01 | 2.82E+0 | 4.37E-01 | 4.34E-01 | - |
| Poisson-P | 2d-C | 2.24E-01 | 7.07E-02 | **1.66E-02** | 2.33E-01 | 8.24E-02 | 4.03E-01 | 5.51E-1 | 1.84E-01 | 2.97E-01 | 2.87E-2 |
| Heat-P | 2d-MS | 1.73E-01 | 1.23E-01 | 1.50E-01 | 1.53E-01 | 4.00E-01 | 4.59E-01 | 5.12E-1 | **6.27E-02** | 1.89E-01 | 2.46E-1 |
| NS-P | 2d-C | 3.89E-01 | - | 4.52E-01 | 3.91E-01 | 9.33E-01 | 7.19E-01 | 3.76E-1 | **3.63E-01** | 4.85E-01 | 3.99E-1 |
| Wave-P | 1d-C | 5.22E-01 | 3.44E-01 | **2.69E-01** | 5.05E-01 | 6.89E-01 | 7.66E-01 | 3.58E-1 | 4.03E-01 | 9.00E-01 | 1.15E+0 |
| High dim-P | HNd | 7.66E-03 | 6.53E-03 | 9.04E-03 | 8.07E-03 | **2.22E-03** | 7.87E-03 | - | 6.97E-03 | 1.94E-01 | - |

Table 4: Results of different PINN variants on parametric PDEs. We report average L2RE on all examples within a class of PDE. We **bold** the best results across all methods.

geometries and multi-scale systems. For example, FBPINN achieves the smallest error on the chaotic GS equation (7.99%), while LAAF delivers the best fitting result on Heat-2d-CG (2.39%).

**Discussion.** For challenges related to complex geometries and multi-scale phenomena, some methods can mitigate these issues by implementing mechanisms like loss reweighting, novel optimizers, and better capacity through adaptive activation. This holds true for the 2D cases of Heat and Poisson equations, which are classic linear equations. However, when systems have higher dimensions (Poisson3d-CG) or longer time spans (Heat2d-LT), all methods fail to solve, highlighting the difficulties associated with complex geometries and multi-scale systems.

In contrast, nonlinear, long-time PDEs like 2D Burgers, NS, and KS pose challenges for most methods. These equations are sensitive to initial conditions, resulting in complicated solution spaces and more local minima for PINNs [51]. The Wave equation, featuring a second-order time derivative and periodic behavior, is particularly hard for PINNs, which often become unstable and may violate conservation laws [24, 55]. Although all methods perform well on Poisson-Nd, only PINN with LBFGS solves Heat-Nd, indicating the potential of a second-order optimizer for solving high dimensional PDEs[53].

## 4.2 Parameterized PDE Experiments

To investigate whether PINNs could handle a class of PDEs, we design this experiment to solve the same PDEs with different parameters. We choose 6 PDEs, i.e., Burgers2d-C, Poisson2d-C, Heat2d-MS, NS-C, Wave1d-C, and Heat-Nd (HNd), with each case containing five parameterized examples. Details of the parametrized PDEs are shown in Appendix B. Here we report the average L2RE metric on these parameterized PDEs for every case, and results are shown in the following Table 4. First, we see that compared with the corresponding cases in Table E.1, the mean L2RE of parameterized PDEs is usually higher. We suggest that this is because there are some difficult cases under certain parameters for these PDEs with very high errors. Secondly, we find that PINN-NTK works well on parameterized PDE tasks which achieve three best results among all six experiments. We speculate that solving PDEs with different parameters requires different weights for loss terms, and PINN-NTK is a powerful method for automatically balancing these weights.

## 4.3 Hyperparameter Analysis

The performance of PINNs is strongly affected by hyperparameters, with each variant introducing its own unique set. The results are shown in Figure 2. We focus on a set of problems, i.e., Burgers1d, GS, Heat2d-CG, and Poisson2d-C. Detailed results and additional findings are in Appendix E.2.

**Batch size and training epochs.** Figure 19 presents the effects of varying batch sizes and training epochs. Larger batch sizes generally yield better outcomes due to more accurate gradient estimations, though saturation is observed beyond a batch size of 2048 for the GS and Poisson2d-C problems. Similarly, increasing the number of training epochs reduces the L2RE, indicating an improvement in model accuracy. However, this benefit plateaus around 20k to 80k epochs, where further increases in epochs do not significantly reduce the error.

**Learning Rates.** The performance of standard PINNs under various learning rates and learning rate schedules is shown in Figure 3. We observe that the influence of the learning rate on performance is intricate, with optimal learning rates varying across problems. Furthermore, PINN training tends to be unstable. High learning rates, such as $10^{-2}$, often lead to error spikes, while low learning rates,

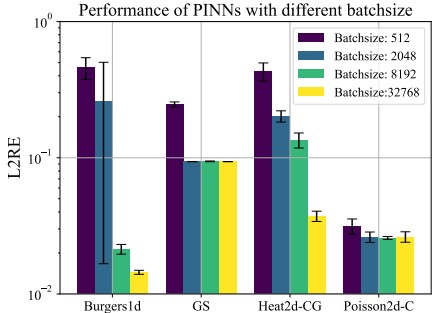 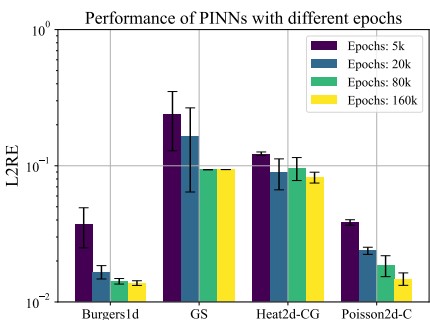

Figure 2: Performance of vanilla PINNs under different batch sizes (number of collocation points), which is shown in the left figure; and number of training epochs, which is shown in the right figure.

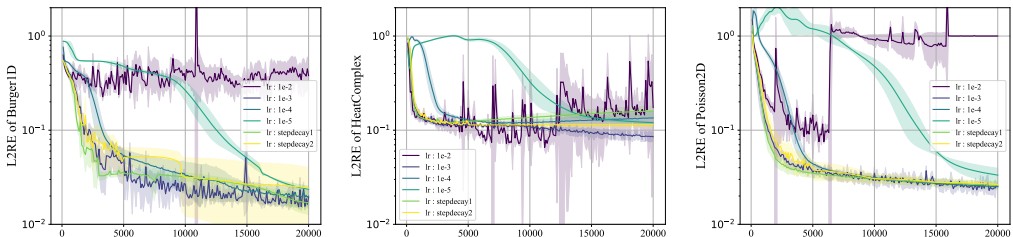

Figure 3: Convergence curve of PINNs with different learning rate schedules on Burgers1d, Heat2d-CG, and Poisson2d-C.

like $10^{-5}$, result in slow convergence. Our findings suggest that a moderate learning rate, such as $10^{-3}$ or $10^{-4}$, or a step decay learning rate schedule, tends to yield more stable performance.

## 5 Limitations

First, real-world problems are often more complex, with giant geometric domains or chaotic behaviors. Good performance on PINNacle does not guarantee it solves practical problems. We could explore larger-scale PINN training methods or efficient domain decomposition methods[7]. Second, the issues of safety in the PINN methods pose potential roadblocks. Developing theoretical convergence for PINN like stability and convergence analysis [13] could help resolve these limitations.

## 6 Conclusion

In this work, we introduced PINNacle, a comprehensive benchmark offering a user-friendly toolbox that encompasses over 20 PDE problems and 10 PINN methods with extensive experiments and ablation studies. Looking forward, we plan to expand the benchmark by integrating additional state-of-the-art methods and incorporating more practical problem scenarios. Our analysis of the experimental results yields several key insights. First, domain decomposition is beneficial for addressing problems characterized by complex geometries, and PINN-NTK is a strong method for balancing loss weights as experiments show. Second, selecting appropriate hyperparameters is crucial to the performance of PINNs. However, the best hyperparameters usually vary with PDEs. Third, we identify high-dimensional and nonlinear problems as a pressing challenge. The overall performance of PINNs is not yet on par with traditional numerical methods [16]. Fourth, there are only a few attempts exploring of PINNs' loss landscape [30]. Finally, integrating the strengths of neural networks with numerical methods like preconditioning, weak formulation, and multigrid may present a promising avenue toward overcoming the challenges[36].

## Acknowledgement

This work was supported by NSFC Projects (Nos. 62350080, 92370124, 92248303, 62276149, U2341228, 62061136001, 62076147), BNRist (BNR2022RC01006), Tsinghua Institute for Guo Qiang, and the High Performance Computing Center, Tsinghua University. J. Zhu was also supported by the XPlorer Prize.

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

## A  Overview of Appendices

We provide supplementary details about problems and experiments for the main text in the Appendix. In Appendix B, we provide mathematical descriptions and visualization for all PDEs in this paper. In Appendix C, we list the detailed hyperparameters and training/testing settings. In Appendix D, we provide a high-level overview of the codebase of the toolbox. In Appendix E, the results for the main experiments, i.e., the performance of L2RE, L1RE, MSE, and runtime for all methods on all PDEs are displayed. In Appendix F, we show the visualization results for several methods on some problems.

## B  Details of PDEs and Methods

Here provide details of PDE tasks used for evaluating different variants of PINNs. Denote $u$ to be the function to solve and $x, t$ to be spatial and temporal variables.

### B.1  Definitions for PDEs in main experiments

#### 1. One-dimensional Burgers Equation (Burgers1d)

The Burgers 1D equation is given by

$$u_t + uu_x = \nu u_{xx}. \tag{6}$$

The domain is defined as

$$(x,t) \in \Omega = [-1,1] \times [0,1]. \tag{7}$$

The initial and boundary conditions are

$$u(x,0) = -\sin \pi x, \tag{8}$$
$$u(-1,t) = u(1,t) = 0. \tag{9}$$

The parameter is

$$\nu = \frac{0.01}{\pi}. \tag{10}$$

#### 2. 2D Coupled Burgers equation (Burgers 2d)

The 2D Coupled Burgers equation is given by

$$\boldsymbol{u}_t + \boldsymbol{u} \cdot \nabla \boldsymbol{u} - \nu \Delta \boldsymbol{u} = 0, \tag{11}$$
$$\boldsymbol{u}(0,y,t) = \boldsymbol{u}(L,y,t), \quad \boldsymbol{u}(x,0,t) = \boldsymbol{u}(x,L,t), \tag{12}$$
$$\{x,y\} \in [0,L], \quad t \in [0,T], \tag{13}$$

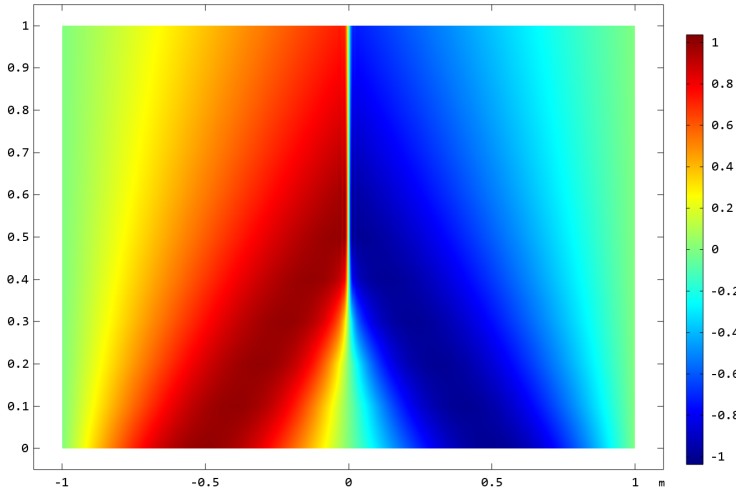

Figure 4: Reference solution of Burgers1d using FEM solver.

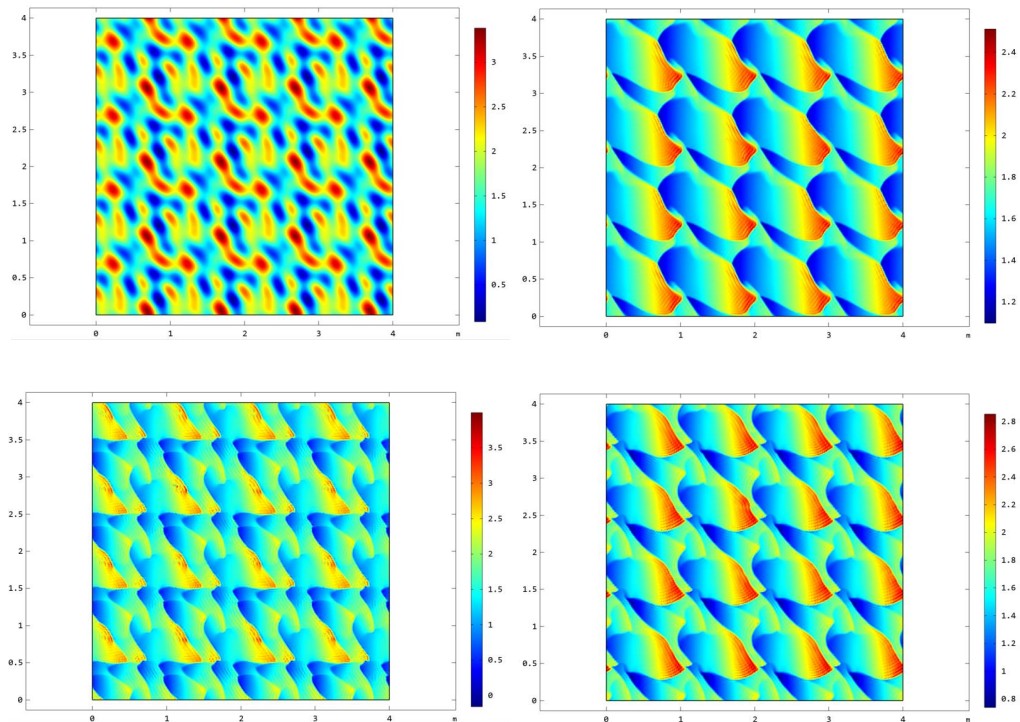

Figure 5: Reference solution of Burgers2d at timesteps $t = 0, 0.2, 0.4, 1.0$ using FEM solver.

The domain is defined as

$$(x, y, t) \in \Omega = [0, L]^2 \times [0, 1]. \tag{14}$$

The initial conditions are given by

$$\boldsymbol{w}(x, y) = \sum_{i=-L}^{L} \sum_{j=-L}^{L} \boldsymbol{a}_{ij} \sin(2\pi(ix + jy)) + \boldsymbol{b}_{ij} \cos(2\pi(ix + jy)), \tag{15}$$

$$\boldsymbol{u}(x, y, 0) = 2\boldsymbol{w}(x, y) + \boldsymbol{c} \tag{16}$$

where $a, b, c \sim N(0, 1)$. The parameters are

$$L = 4, \quad T = 1, \quad \nu = 0.001. \tag{17}$$

## 3. Poisson 2D Classic (Poisson2d-C)

The Poisson 2D equation is given by

$$-\Delta u = 0. \tag{18}$$

The domain is a rectangle minus four circles $\Omega = \Omega_{rec} \setminus R_i$ where $\Omega_{rec} = [-0.5, 0.5]^2$ is the rectangle and $R_i$ denotes four circle areas:

$$\begin{align}
R_1 &= \{(x, y) : (x - 0.3)^2 + (y - 0.3)^2 \leq 0.1^2\}, \tag{19}\\
R_2 &= \{(x, y) : (x + 0.3)^2 + (y - 0.3)^2 \leq 0.1^2\}, \tag{20}\\
R_3 &= \{(x, y) : (x - 0.3)^2 + (y + 0.3)^2 \leq 0.1^2\}, \tag{21}\\
R_4 &= \{(x, y) : (x + 0.3)^2 + (y + 0.3)^2 \leq 0.1^2\}. \tag{22}
\end{align}$$

The boundary condition is

$$\begin{align}
u &= 0, x \in \partial R_i, \tag{23}\\
u &= 1, x \in \partial \Omega_{rec}. \tag{24}
\end{align}$$

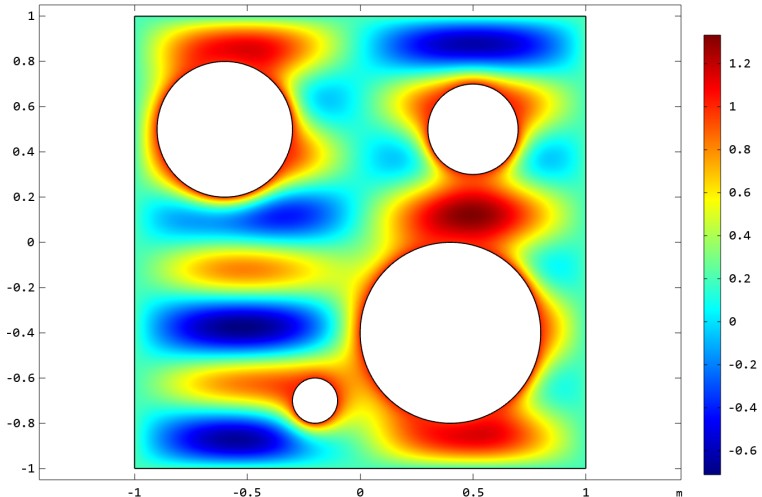

Figure 6: Reference solution of Poisson2d-CG by FEM solver.

## 4. Poisson-Boltzmann (Helmholtz) 2D Irregular Geometry (Poisson2d-CG)

The Poisson-Boltzmann (Helmholtz) 2D equation is given by

$$-\Delta u + k^2 u = f(x, y). \tag{25}$$

The function $f(x)$ is defined as

$$f(x) = A \cdot \left( \sum_i \mu_i^2 + x_i^2 \right) \sin(\mu_1 \pi x_1) \sin(\mu_2 \pi x_2). \tag{26}$$

The domain is $[-1, 1]^2$ and the boundary conditions are

$$u = 0.2, \quad x \in \partial\Omega_{\text{rec}}, \tag{27}$$
$$u = 1, \quad x \in \partial\Omega_{\text{circle}}. \tag{28}$$

Parameter references are

$$\mu_1 = 1, \quad \mu_2 = 4, \quad k = 8, \quad A = 10. \tag{29}$$

The domain is $[-1, 1]^2$ with several circles removed. The circles $\Omega_{circle} = \cup_{i=1}^4 R_i$ are

$$R_1 = \{(x, y) : (x - 0.5)^2 + (y - 0.5)^2 \leq 0.2^2\} \tag{30}$$
$$R_2 = \{(x, y) : (x - 0.4)^2 + (y + 0.4)^2 \leq 0.4^2\} \tag{31}$$
$$R_3 = \{(x, y) : (x + 0.2)^2 + (y + 0.7)^2 \leq 0.1^2\} \tag{32}$$
$$R_4 = \{(x, y) : (x + 0.6)^2 + (y - 0.5)^2 \leq 0.3^2\} \tag{33}$$

## 5. Poisson 3D Complex Geometry with Two Domains (Poisson3d-CG)

The Poisson 3D equation with two domains is given by

$$-\mu_i \Delta u + k_i^2 u = f(x, y, z), \quad i = 1, 2. \tag{34}$$

The function $f(x, y, z)$ is defined as

$$f(x, y, z) = A_1 \frac{\exp(\sin m_1 \pi x + \sin m_2 \pi y + \sin m_3 \pi z)}{x^2 + y^2 + z^2 + 1} (x^2 + y^2 + z^2 - 1) + A_2 \sin(m_1 \pi x) \sin(m_2 \pi y) \sin(m_3 \pi z). \tag{35}$$

The coefficients are defined as $\begin{cases} \mu = \mu_1, & k = k_1, \quad x \in \Omega_1, \\ \mu = \mu_2, & k = k_2, \quad x \in \Omega_2. \end{cases}$

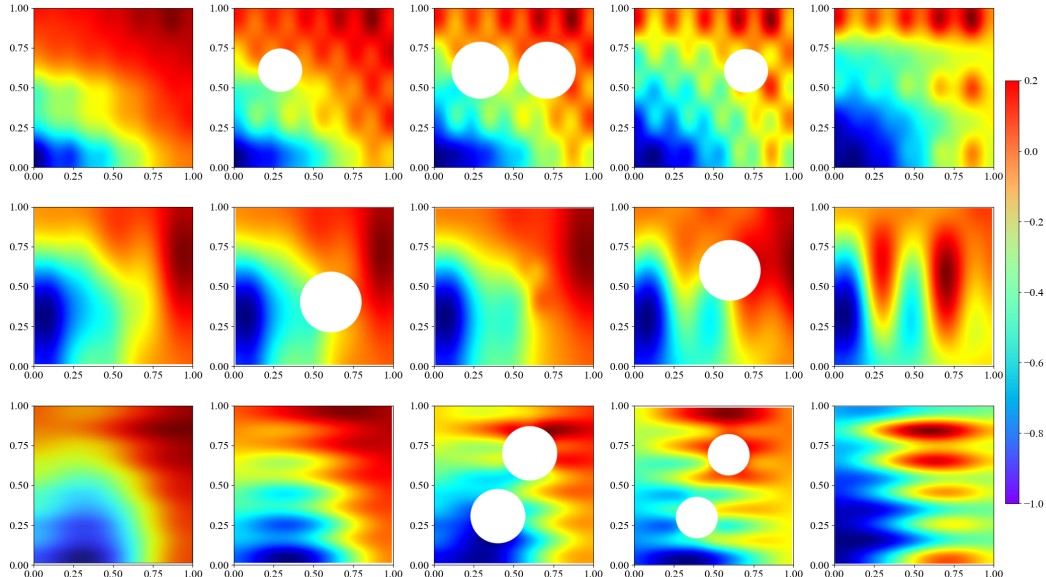

Figure 7: Reference solution of Poisson3d-CG by FEM solver. The top row displays the solution at 5 YZ planes with $x = 0, 0.25, 0.5, 0.75, 1.0$. The medium row displays it at XZ planes with $y = 0.0, 0.25, 0.5, 0.75, 1.0$. The bottom row displays it at XY planes with $z = 0.0, 0.25, 0.5, 0.75, 1.0$.

The boundary condition is

$$\frac{\partial u}{\partial n} = 0, \quad x \in \partial\Omega. \tag{36}$$

The domains and other parameters are defined as follows:

$$\Omega_1 = [0,1] \times [0,1] \times [0,0.5] / \cup_{i=1}^{4} R_i, \tag{37}$$

$$\Omega_2 = [0,1] \times [0,1] \times [0.5,1] / \cup_{i=1}^{4} R_i. \tag{38}$$

The circular regions $R_i$ are

$$R_1 = \{(x,y,z) : (x-0.4)^2 + (y-0.3)^2 + (z-0.6)^2 \le 0.2^2\} \tag{39}$$

$$R_1 = \{(x,y,z) : (x-0.6)^2 + (y-0.7)^2 + (z-0.6)^2 \le 0.2^2\} \tag{40}$$

$$R_1 = \{(x,y,z) : (x-0.2)^2 + (y-0.8)^2 + (z-0.7)^2 \le 0.1^2\} \tag{41}$$

$$R_1 = \{(x,y,z) : (x-0.6)^2 + (y-0.2)^2 + (z-0.3)^2 \le 0.1^2\} \tag{42}$$

$$\tag{43}$$

Other parameters are

$$m_1 = 1, m_2 = 10, m_3 = 5, \mu_1 = 1, \mu_2 = 1, k_1 = 8, k_2 = 10, A_1 = 20, A_2 = 100. \tag{44}$$

### 6. 2D Poisson equation with many subdomains (Poisson2d-MS)

The PDE and boundary condition is given by

$$-\nabla(a(x)\nabla u) = f(x,y), x in \Omega \tag{45}$$

$$\frac{\partial u}{\partial n} + u = 0, x \in \partial\Omega. \tag{46}$$

Here the domain is $(x,y) \in \Omega = [-10, 10]^2$. We divide the whole domain into many small squares, and $a(x)$ is a piecewise linear function in each square. We store the $a(x)$ in a file in practical implementation.

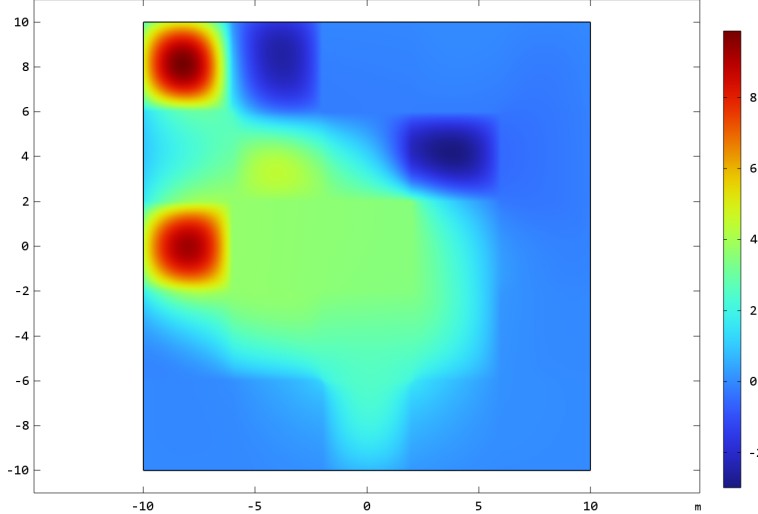

Figure 8: Reference solution of Poisson2d-MS by FEM solver.

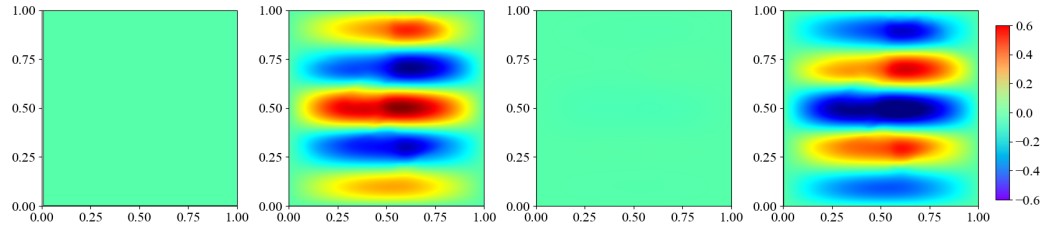

Figure 9: Reference solution of Heat2d-VC by FEM solver at timesteps $t = 0, 0.5, 2.0, 3.5$.

## 7. 2D Heat with Varying Coefficients (Heat2d-VC)

The 2D heat equation with a varying source is given by

$$\frac{\partial u}{\partial t} - \nabla(a(x)\nabla u) = f(x, t). \tag{47}$$

The domain is $\Omega \times T = [0, 1]^2 \times [0, 5]$. The function $a(x)$ is chosen similarly to Darcy flow but with an exponential GRF. The function $f(x, t)$ is defined as

$$f(x, t) = A \sin(m_1 \pi x) \sin(m_2 \pi y) \sin(m_3 \pi t). \tag{48}$$

with $A = 200, m_1 = 1, m_2 = 5, m_3 = 1$. The initial and boundary conditions are

$$
\begin{align}
u(x, y, 0) &= 0, x \in \Omega \tag{49}\\
u(x, y, t) &= 0, x \in \partial\Omega. \tag{50}
\end{align}
$$

## 8. 2D Heat Multi-Scale (Heat2d-MS)

The 2D heat multi-scale equation is given by

$$\frac{\partial u}{\partial t} - \frac{1}{(500\pi)^2} u_{xx} - \frac{1}{\pi^2} u_{yy} = 0, \tag{51}$$

with domain $\Omega \times T = [0, 1]^2 \times [0, 5]$.

The initial and boundary conditions are

$$
\begin{align}
u(x, y, 0) &= \sin(20\pi x)\sin(\pi y), \quad x \in \Omega, \tag{52}\\
u(x, y, t) &= 0, \quad x \in \partial\Omega. \tag{53}
\end{align}
$$

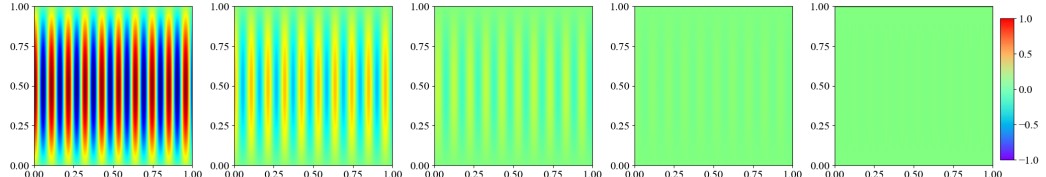

Figure 10: Reference solution of Heat2d-MS by FEM solver.

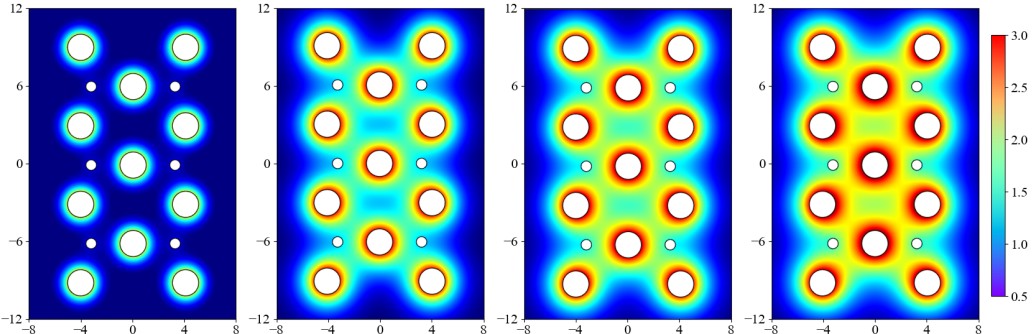

Figure 11: reference solution of Heat2d-CG by FEM solver at timesteps $t = 0.5, 2.0, 2, 5, 3.0$.

## 9. 2D Heat Complex Geometry (Heat Exchanger, Heat2d-CG)

The 2D heat equation for a complex geometry is given by

$$\frac{\partial u}{\partial t} - \Delta u = 0. \tag{54}$$

The domain is defined as $\Omega \times T = ([-8, 8] \times [-12, 12] \setminus \cup_i R_i) \times [0, 3]$.

The boundary condition is

$$-n \cdot (-c\nabla u) = g - qu. \tag{55}$$

Here we choose $c = 1$. The positions of large circles are

$$(\pm 4, \pm 3), \quad (\pm 4, \pm 9), \quad (0, 0), \quad (0, \pm 6), \quad r = 1 \tag{56}$$

with $g = 5$ and $q = 1$. The positions of small circles are

$$(\pm 3.2, \pm 6), \quad (\pm 3.2, 0), \quad r = 0.4 \tag{57}$$

with $g = 1$ and $q = 1$. For the rectangular boundary conditions, $g = 0.1$ and $q = 1$.

## 10. 2D Heat Long Time (Heat2d-LT)

The governing PDE is

$$\frac{\partial u}{\partial t} = 0.001\Delta u + 5\sin(ku^2)\left(1 + 2\sin\left(\frac{\pi t}{4}\right)\right)\sin(m_1\pi x)\sin(m_2\pi y) \tag{58}$$

with domain $\Omega \times T = [0, 1]^2 \times [0, 100]$, $m_1 = 4$, $m_2 = 2$, and $k = 1$.

The initial and boundary conditions are given by

$$u(x, y, 0) = \sin(4\pi x)\sin(3\pi y), x \in \Omega \tag{59}$$
$$u(x, y, t) = 0, \quad x \in \partial\Omega. \tag{60}$$

## 11. 2D NS lid-driven flow (NS2d-C).

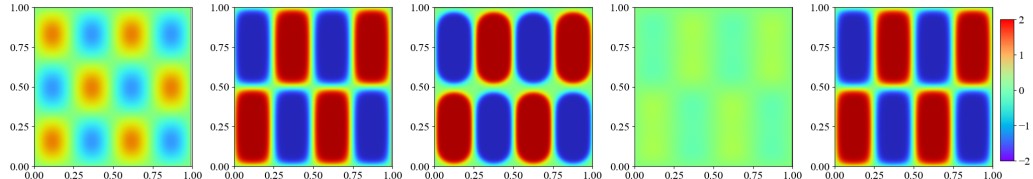

Figure 12: reference solution of Heat2d-LT by FEM solver at timesteps $t = 0, 20, 50, 80, 100$.

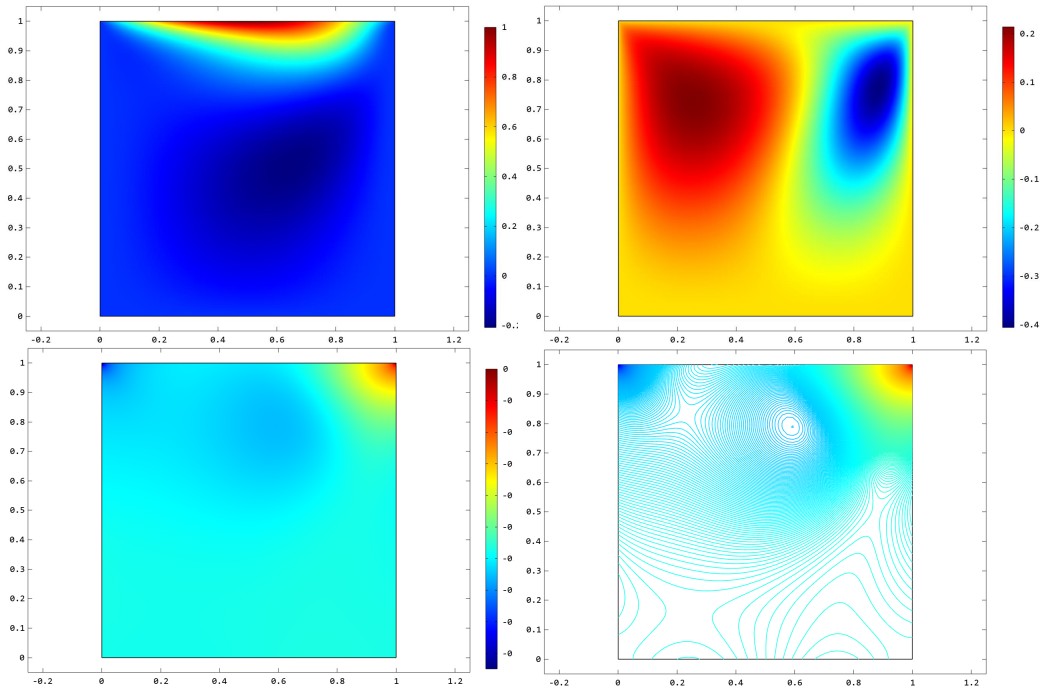

Figure 13: Reference solution of NS2d-Ld by FEM solver.

The PDE is given by

$$\boldsymbol{u} \cdot \nabla \boldsymbol{u} + \nabla p - \frac{1}{Re} \Delta \boldsymbol{u} \quad = \quad 0, x \in \Omega \tag{61}$$

$$\nabla \cdot \boldsymbol{u} \quad = \quad 0, x \in \Omega \tag{62}$$

The domain is $\Omega = [0,1]^2$, the top boundary is $\Gamma_1$, the left, right and bottom boundary is $\Gamma_2$.

The boundary conditions are

$$\boldsymbol{u}(\boldsymbol{x}) \quad = \quad (4x(1-x), 0), x \in \Gamma_1 \tag{63}$$

$$\boldsymbol{u}(\boldsymbol{x}) \quad = \quad (0,0), x \in \Gamma_2 \tag{64}$$

$$p \quad = \quad 0, x = (0,0). \tag{65}$$

The Reynolds number Re $= 100$.

### 12. 2D Back Step Flow (NS-CG)

The equations and boundary conditions are given by

$$u \cdot \nabla u + \nabla p - \frac{1}{Re} \Delta u \quad = \quad 0, \tag{66}$$

$$\nabla \cdot u \quad = \quad 0. \tag{67}$$

The domain is defined as $\Omega = [0,4] \times [0,2] \setminus ([0,2] \times [1,2] \bigcup R_i)$ (excluding the top-left quarter).

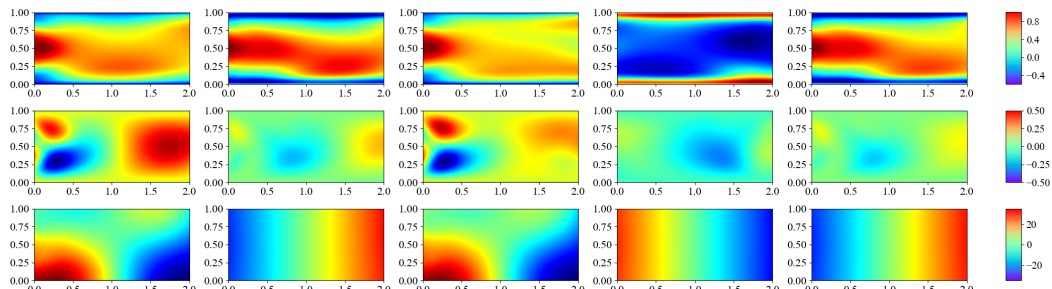

Figure 14: Reference fields $u, v, p$ from top to bottom of NS2d-LT by FEM solver at timesteps $t = 0.5, 1.0, 2.5, 4.0, 5.0$.

The inlet velocity is given by $u_{\text{in}} = 4y(1 - y)$, the outlet pressure is $p = 0$, and the boundary condition is no-slip: $\mathbf{u} = 0$. The Reynolds number of Re $= 100$.

13. **2D NS Long Time (NS2d-LT)**

The PDE of this case is given by

$$\frac{\partial u}{\partial t} + u \cdot \nabla u + \nabla p - \frac{1}{\text{Re}}\Delta u = f(x, y, t), \tag{68}$$

$$\nabla \cdot u = 0. \tag{69}$$

The domain is $\Omega \times T = ([0, 2] \times [0, 1]) \times [0, 5]$, and the forcing term $f(x, y, t)$ can be given as

$$f(x, y, t) = (0, -\sin(\pi x)\sin(\pi y)\sin(\pi t)). \tag{70}$$

The boundary conditions are similar to case 12, and the left inlet initial condition can be given as an oscillatory form:

$$u(0, y, t) = \sin(\pi y)(A_1 \sin(\pi t) + A_2 \sin(3\pi t) + A_3 \sin(5\pi t)). \tag{71}$$

where $A_1 = 1, A_2 = 1, A_3 = 1$.

The initial condition in the domain is

$$u(x, y, 0) = 0. \tag{72}$$

14. **Basic 1D Wave Equation (Wave1d-C)**

The governing PDE is

$$u_{tt} - 4u_{xx} = 0 \tag{73}$$

The domain is $\Omega \times T = [0, 1] \times [0, 1]$. The boundary conditions are

$$u(0, t) = u(1, t) = 0 \tag{74}$$

The initial condition:

$$u(x, 0) \quad = \quad \sin(\pi x) + \frac{1}{2}\sin(4\pi x) \tag{75}$$

$$u_t(x, 0) \quad = \quad 0 \tag{76}$$

The analytical solution of this problem is

$$u(x, t) = \sin(\pi x)\cos(2\pi t) + \frac{1}{2}\sin(4\pi x)\cos(8\pi t). \tag{77}$$

15. **2D Wave Equation in Heterogeneous Medium (Wave2d-CG)**

The governing PDE is given by

$$\left[\nabla^2 - \frac{1}{c(x)}\frac{\partial^2}{\partial t^2}\right]u(x, t) = 0 \tag{78}$$

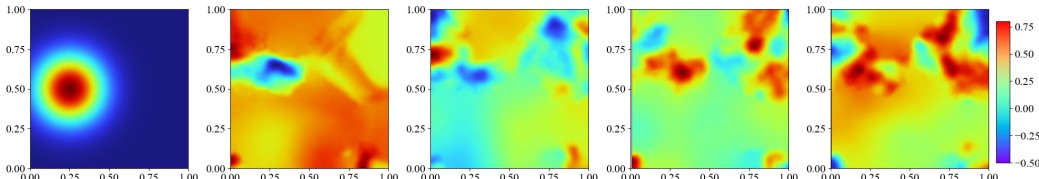

Figure 15: Reference solution of Wave2d-CG by FEM solver at timesteps $t = 0, 0.5, 2.0, 4.0, 5.0$.

The Domain is $\Omega = [-1, 1] \times [-1, 1]$ and the initial condition is

$$u(x, 0) = \exp\left(-\frac{\|x - \mu\|^2}{2\sigma^2}\right), x \in \Omega \tag{79}$$

$$\frac{\partial u}{\partial t}(x, 0) = 0, x \in \Omega \tag{80}$$

The boundary conditions are

$$\frac{\partial u}{\partial n} = 0, x \in \partial\Omega \tag{81}$$

The parameters are

$$\mu = (-0.5, 0), \sigma = 0.3, \tag{82}$$

and $c(x)$ are generated by a Gaussian random field.

## 16. 2D Multi-Scale Long Time Wave Equation (Wave2d-MS)

The governing PDE is

$$u_{tt} - (u_{xx} + a^2 u_{yy}) = 0 \tag{83}$$

The domain is defined as $\Omega = [0, 1]^2 \times [0, 100]$ and the boundary and initial conditions are

$$u(x, y, t) = c_1 \sinh(m_1 \pi x) \sinh(n_1 \pi y) \cos(p_1 \pi t), (x, y) \in \partial\Omega. \tag{84}$$

$$\frac{\partial u}{\partial t}(x, y, 0) = 0 \tag{85}$$

The exact solution to this problem is

$$u(x, y, t) = c_1 \sinh(m_1 \pi x) \sinh(n_1 \pi y) \cos(p_1 \pi t), \tag{86}$$

where $a = \sqrt{2}, m_1 = 1, n_1 = 1, p_1 = \sqrt{3}$ and $c_1 = 1$.

## 17. 2D Diffusion-Reaction Gray-Scott Model (GS)

The governing PDE is

$$u_t = \varepsilon_1 \Delta u + b(1 - u) - uv^2 \tag{87}$$
$$v_t = \varepsilon_2 \Delta v - dv + uv^2 \tag{88}$$

The domain is $\Omega \times T = [-1, 1]^2 \times [0, 200]$ and parameters are

$$b = 0.04, d = 0.1, \varepsilon_1 = 1 \times 10^{-5}, \varepsilon_2 = 5 \times 10^{-6} \tag{89}$$

The initial conditions are

$$u(x, y, 0) = 1 - \exp(-80((x + 0.05)^2 + (y + 0.02)^2)) \tag{90}$$
$$v(x, y, 0) = \exp(-80((x - 0.05)^2 + (y - 0.02)^2)) \tag{91}$$

The visualization of the reference solution of this case is in Figure 16.

## 18. Kuramoto-Sivashinsky Equation (KS)

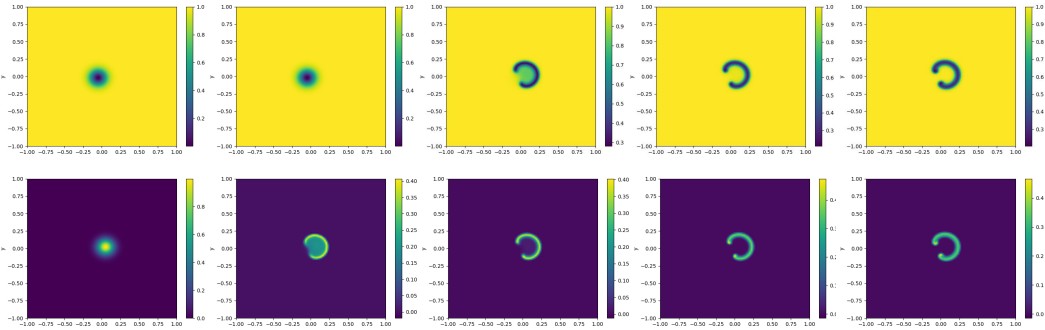

Figure 16: Reference solution of GS equation at timestep $t = 0.0, 2.5, 5.0, 7.5, 10.0$.

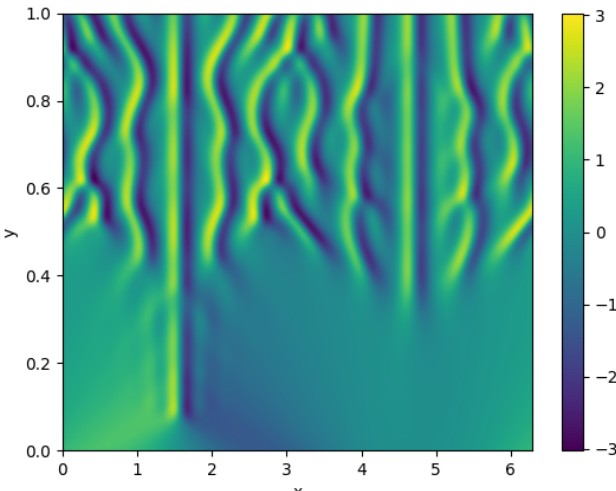

Figure 17: Reference solution of KS equation.

The governing PDE is

$$u_t + \alpha u u_x + \beta u_{xx} + \gamma u_{xxxx} = 0 \tag{92}$$

The domain is $\Omega \times T = [0, 2\pi] \times [0, 1]$. (Note: Error may increase rapidly in chaotic problems.)

$$\alpha = \frac{100}{16}, \beta = \frac{100}{16^2}, \gamma = \frac{100}{16^4} \tag{93}$$

The initial condition is

$$u(x, 0) = \cos(x)(1 + \sin(x)) \tag{94}$$

The reference solution of KS equation is shown in Figure B.1.

### 19. N-Dimensional Poisson equation (PNd)

The governing PDE is

$$-\Delta u = \frac{\pi^2}{4} \sum_{i=1}^{n} \sin\left(\frac{\pi}{2} x_i\right) \tag{95}$$

The domain is defined by $\Omega = [0, 1]^n$. The exact solution is

$$u = \sum_{i=1}^{n} \sin\left(\frac{\pi}{2} x_i\right) \tag{96}$$

We choose $n = 5$ in our code.

## 20. N-Dimensional Heat Equation (HNd)

The governing PDE is

$$\frac{\partial u}{\partial t} = k\Delta u + f(x,t), x \in \Omega \times [0,1] \tag{97}$$

$$\boldsymbol{n} \cdot \nabla u = g(x,t), x \in \partial\Omega \times [0,1] \tag{98}$$

$$u(x,0) = g(x,0), x \in \Omega \tag{99}$$

The geometric domain $\Omega = \{x : |x|_2 \leqslant 1\}$ is a unit sphere in $d$-dimensional space. We choose dimension $d = 5$.

$$k = \frac{1}{d} \tag{100}$$

The two functions are

$$f(x,t) = -\frac{1}{d}|x|_2^2 \exp\left(\frac{1}{2}|x|_2^2 + t\right) \tag{101}$$

$$g(x,t) = \exp\left(\frac{1}{2}|x|_2^2 + t\right) \tag{102}$$

We can see that the exact solution of the equation is $g(x,t)$.

## 21. Poisson inverse problem (PInv)

The governing PDE is

$$-\nabla(a\nabla u) = f \tag{103}$$

The geometric domain is $\Omega = [0,1]^2$, and

$$u = \sin \pi x \sin \pi y. \tag{104}$$

The source term $f$ is

$$f = \frac{2\pi^2 \sin \pi x \sin \pi y}{1 + x^2 + y^2 + (x-1)^2 + (y-1)^2} + \frac{2\pi((2x-1)\cos \pi x \sin \pi y + (2y-1)\sin \pi x \cos \pi y)}{(1 + x^2 + y^2 + (x-1)^2 + (y-1)^2)^2}. \tag{105}$$

To ensure the uniqueness of the solution, we impose a boundary condition of $a(x,y)$, i.e.,

$$a(x,y) = \frac{1}{1 + x^2 + y^2 + (x-1)^2 + (y-1)^2}, x \in \partial\Omega \tag{106}$$

We sample data of $u(x,y)$ with 2500 uniformly distributed $50 \times 50$ points and add Gaussian noise $\mathcal{N}(0, 0.1)$ to it. The goal is to reconstruct the diffusion coefficients. We see that the ground truth of $a(x,y)$ is

$$a(x,y) = \frac{1}{1 + x^2 + y^2 + (x-1)^2 + (y-1)^2}, x \in \Omega. \tag{107}$$

## 22. Heat (Diffusion) inverse problem (HInv)

The governing PDE of this inverse problem is

$$u_t - \nabla(a\nabla u) = f \tag{108}$$

The geometric domain is $\Omega \times T = [-1,1]^2 \times [0,1]$, and

$$u = e^{-t} \sin \pi x \sin \pi y \tag{109}$$

Similarly, we impose a boundary condition for the diffusion coefficient field:

$$a(x,y) = 2, \partial x \in \Omega. \tag{110}$$

Then the source function $f$ is

$$f = ((4\pi^2 - 1)\sin \pi x \sin \pi y + \pi^2(2\sin^2 \pi x \sin^2 \pi y - \cos^2 \pi x \sin^2 \pi y - \sin^2 \pi x \cos^2 \pi y))e^{-t} \tag{111}$$

We sample data of $u(x,y,t)$ randomly with 2500 points from the temporal domain $\Omega \times T$ and add Gaussian noise $\mathcal{N}(0, 0.1)$ to it. The goal is to reconstruct the diffusion coefficients. We see that the ground truth is

$$a(x,y) = 2 + \sin \pi x \sin \pi y, x \in \Omega. \tag{112}$$

## B.2 Definitions and design choices for parametric PDEs

We design a set of parametric PDEs and evaluate the average performance of PINN variants on cases with different parameters. We choose Burgers2d-C, Poisson2d-C, Heat2d-MS, NS2d-C, Wave2d-C, and Heat-Nd to design these parametric cases.

### 1. 2D Coupled Burgers equation (Burgers2d-C) with different initial values.

The initial values of this case are shown in Eq 16 where $a$ and $b$ are sampled from Gaussian Random Field. Here the initial values are used as parameters and we sample 5 different $a$ and $b$ from GRF and test the performance of PINN variants on all 5 cases. Each parametrized PDE is solved using COMSOL. In PDEBench, the authors similarly tested the average effect of PDEs sampled multiple times from the GRF with the same equation. Since the GRF has not changed, there is not much variation in the magnitude and frequency of the initial flow velocity, but there may be significant differences in their spatial distribution. This can also lead to differences in difficulty when solving with the PINN method. From the 4, we see that the error of the best method increased from 26% to 41%, indicating a significant influence of the flow distribution on the solution.

### 2. Poisson 2d Classic (Poisson2d-C)

This PDE is defined on $\Omega = [-L, L]^2$. We parametrize this case by using different domain scales $L$ from $\{1, 2, 4, 8, 16\}$. Since this PDE is linear, we could compute the ground truth solution by linearly scaling the original PDE where $L = 0.5$. Some papers [61] pointed out that the effect of PINN is influenced by the size of the domain. This is because scaling the domain directly to $[0, 1]^d$ may be suboptimal and can lead to an imbalanced ratio of PDE loss to boundary loss. This is because PINNs are sensitive to initialization, so different domain scales might lead to different results. Here the real solution of this linear PDE can be obtained through a linear transformation from a solution of another domain scale $L$. The condition number does not differ when we change $L$, making it suitable to study the influence of domain scale on PINN's performance. We observed from the results that some methods (PINN-NTK, MultiAdam, FBPINN) are relatively robust to domain scale.

### 3. 2D Heat Multi-Scale (Heat2d-MS)

We parameterize this case using different initial conditions in Eq 53,

$$u(x, y, 0) = \sin(a\pi x)\sin(\pi y). \tag{113}$$

Here we choose $(a, b)$ from $\{(20, 1), (1, 20), (10, 2), (2, 10), (5, 4)\}$. The reference solutions for different parameters are solved using COMSOL. Changes in the frequency of the initial condition will lead to changes in the frequency of the solution, which allows us to study the influence of the initial condition frequency on PINN. Comparing the results of several experiments, we found that the loss reweighting strategy of PINN-NTK and the adaptive activation function of LAAF perform well for multi-scale problems overall. However, when the frequency variation range is more significant, both their performances decline, suggesting room for improvement.

### 4. 2D NS lid-driven flow (NS2d-C)

We parametrize NS2d-C by setting different speeds at the top boundary in Eq 65,

$$\boldsymbol{u}(\boldsymbol{x}) = (ax(1 - x), 0), x \in \Gamma_1, \tag{114}$$

where $a$ is chosen from $\{2, 4, 8, 16, 32\}$. The reference solutions for different parameters are solved using COMSOL. Different flow rates imply different Reynolds numbers, thus altering the difficulty of solving the equation. As the Reynolds number increases, the condition number of the equation will also increase. Generally, the higher the Reynolds number, the more likely turbulence or some small-scale complex flow states will occur. Testing different Reynolds numbers is a natural idea. Specifically, we chose a velocity $u = ax(1 - x)$, where $a$ ranges between 2 and 32. Compared to the main experiment with $a = 4$, the Reynolds number increased eightfold when $a = 32$.

### 5. 1D Wave Equation

We parametrize this case with different initial conditions in Eq 76,

$$u(x, 0) = \sin(\pi x) + \frac{1}{2}\sin(a\pi x), \tag{115}$$

where $a$ is chosen from $\{2, 4, 6, 8, 10\}$. The ground truth solution is given by,

$$u(x, t) = \sin(\pi x)\cos(2\pi t) + \frac{1}{2}\sin(\pi a x)\cos(2a\pi t). \tag{116}$$

## 6. N-Dimensional Heat Equation

We parametrize this case by choosing a different number of dimensions $n$ from $\{4, 5, 6, 8, 10\}$. The solutions are given by Eq 102. Although neural networks are theoretically universal function approximators, the ability to fit the solution of high-dimensional PDEs still needs to be studied. So, we chose heat equations of different dimensions to compare the effects of various PINN methods. We observed that for high-dimensional heat equations, the improved optimizer MultiAdam is very helpful in solving high-dimensional problems.

| PDE type | | Software | Solver | #Mesh |
|---|---|---|---|---|
| Burgers | 1d-C | Comsol | BDF,MUMPS | 1000 |
| | 2d-C | Comsol | BDF,MUMPS | 24912 |
| Poisson | 2d-C | Comsol | MUMPS | 40000 |
| | 2d-CG | Comsol | MUMPS | 22420 |
| | 3d-CG | Comsol | MUMPS | 371024 |
| | 2d-MS | Comsol | MUMPS | 24912 |
| Heat | all | Comsol | BDF,MUMPS | 24912 |
| Naiver-Stokes | 2d-C | Comsol | PARDISO | 10000 |
| | 2d-CG | Comsol | PARDISO | 39294 |
| | 2d-LT | Comsol | BDF,PARDISO | 43250 |
| Wave | 1d-C | | Analytical | |
| | 2d-CG | Comsol | Generalized alpha, MUMPS | 24912 |
| | 2d-MS | Comsol | Generalized alpha, MUMPS | 25140 |
| Chaotic | GS | Chebfun | ETDRK4 | 1000 |
| | KS | Chebfun | ETDRK4 | 1000 |
| High dim | all | | Analytical | |
| Inverse problems | all | | Analytical | |

Table 5: Details of the solver for reference data.

## B.3 Discussion about the reference solutions and Mesh Convergence Study.

Given that our benchmark includes various types of PDEs, we generated the reference data using different types of numerical solvers, including the FEM solvers in COMSOL [39], Chebfun [12], among others. For different PDEs, selecting the appropriate numerical solver allows for higher precision results. The types of solvers used, mesh sizes, parameters, and convergence accuracies are detailed in Table 5. For these problems, highly optimized numerical solvers can achieve solutions with very high accuracy and theoretical guarantees. However, the choice of mesh discretization and solver type is highly dependent on the PDE type and parameters. Although PINNs have limitations in terms of accuracy and efficiency, their flexibility and ability to generalize by incorporating data are advantages over traditional numerical solvers, which is one of the motivations behind the development of PINNs.

We conducted a mesh convergence study for the Poisson3d and NS2d-CG equations using grids with varying spacing. To estimate the error bound, we employed Richardson extrapolation [65], a technique that leverages the solutions from different grid sizes to predict the solution's behavior as the grid is further refined. The principle behind Richardson extrapolation is that the error in the numerical solution decreases predictably with grid refinement. If the error reduces as a power of the grid size $h$, the extrapolated solution $u_{\text{extrapolated}}$ can be calculated as:

$$u_{\text{extrapolated}} = \frac{h_2^p u_{h_1} - h_1^p u_{h_2}}{h_2^p - h_1^p}, \tag{117}$$

where $u_{h_1}$ and $u_{h_2}$ are the solutions obtained on grids with sizes $h_1$ and $h_2$ respectively, and $p$ is the theoretical convergence rate. The error bound can then be estimated by comparing the extrapolated solution with the finer grid solution $u_{h_2}$ as follows:

$$\text{Error Bound} = \frac{||u_{\text{extrapolated}} - u_{h_2}||}{||u_{\text{extrapolated}}||}. \tag{118}$$

We obtained the following Figure 18 showing the error bound as a function of grid size. As we refined the grid, the differences between the solutions decreased, and the error bound rapidly decreased with smaller grid sizes. The error for the reference data we used was below 0.1%, indicating that it is highly reliable and can serve as a valid reference for the PINN solutions.

## B.4 Relationship with existing PDE benchmarks

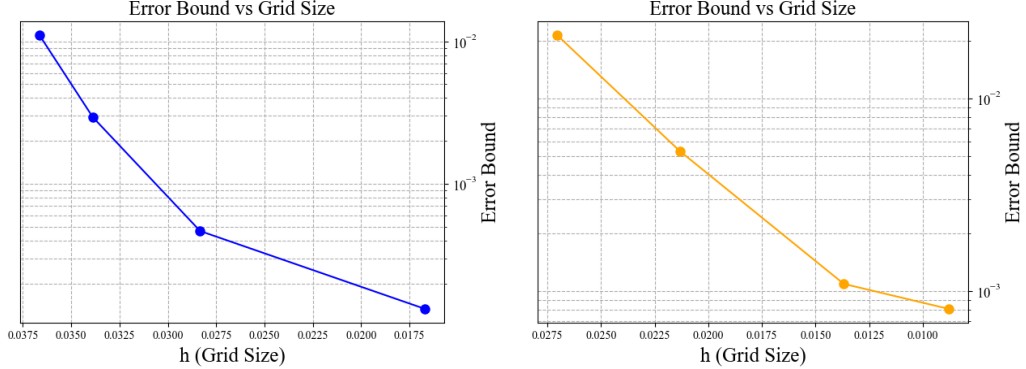

Figure 18: Richardson extrapolation and error bound analysis for Poisson3d-CG(left) and NS2d-CG(right).

| PDE type/Number of PDEs | PINNacle | PDEBench | PDEArena | Wang et. al[56] |
|---|---|---|---|---|
| Burgers | 2 | 1 | 0 | 0 |
| Poisson | 4 | 1 | 0 | 0 |
| Convection-Diffusion | 4 | 3 | 0 | 2 |
| Shallow water | 0 | 1 | 1 | 0 |
| Naiver-Stokes | 3 | 4 | 2 | 4 |
| Wave | 2 | 0 | 0 | 0 |
| Chaotic | 2 | 0 | 0 | 1 |
| High dim | 2 | 0 | 0 | 0 |
| Inverse problems | 2 | 0 | 0 | 0 |

Table 6: A comparison between our work and several existing PDE benchmarks. We list the PDEs used in the experiments for each paper.

Here we compare the PDEs we used with PDEs in PDEBench [52], PDEArena [17], and Wang et. al[56]. We list the number of different PDEs used in the experiments in Table 6. The selection of PDEs for our study was carefully curated to align with the objectives of comparing PINN methods, which differs from the approach taken in PDEBench or PDEArena. While PDEBench and PDEArena are oriented towards time-dependent PDEs, such as the compressible Naiver-Stokes and Diffusion Reaction equations, and provide extensive datasets for neural operator research, our focus was distinct. For [56], we compared more PINN variants and selected a wider range of equations We chose a range of PDEs. specifically for their relevance to PINN research, where datasets are not typically provided, emphasizing the direct application of PINNs to the PDEs themselves. We select a diverse range of PDE types and complexities from existing PINN literature. Among these, we included widely applicable and representative PDEs like the incompressible Naiver-Stokes equation and the Poisson equation (Darcy flow), which are fundamental to a multitude of disciplines. Our choice thus facilitates a more targeted and appropriate comparison of PINN methodologies, underscoring the unique aspects of our research approach.

## B.5 Overview of methods

The baselines we selected could be roughly divided into several categories, i.e., loss reweighting/re-sampling, novel optimizer, novel loss functions, and novel activation/architectures. As shown in Eq 119, the general formulation of PINNs is to optimize a mixture of PDE residual loss, boundary loss, and available data loss,

$$\mathcal{L}(\theta) = \frac{w_c}{N_c} \sum_{i=1}^{N_c} ||\mathcal{F}(u_\theta(x_c^i); x_c^i)||^2 + \frac{w_b}{N_b} \sum_{i=1}^{N_b} ||\mathcal{B}(u_\theta(x_b^i); x_b^i)||^2 + \frac{w_d}{N_d} \sum_{i=1}^{N_d} ||u_\theta(x_d^i) - u(x_d^i)||^2.$$
(119)

Under this formulation, we could explain different variants of PINNs.

- Loss reweighting methods dynamically modify the weights $w_c, w_b, w_d$ to enable a better convergence rate. Resampling methods allocate new collocation points $x_c, x_b$ or adjust their sampling probability. These methods alleviate the imbalance between PINN optimization. Results show that they achieve remarkable results on many cases of Poisson, Heat, and Wave equations.

- Novel loss functions. It modifies the form of $\mathcal{L}(\theta)$ or adds new regularization terms for higher convergence accuracy. Results show that vPINNs are excellent at solving inverse problems.

- Novel optimizer. An example of novel optimizer is Multi-Adam which is more suitable for dealing with multiple conflict loss terms especially when they have a different scale. Results show that it works for several problems with multi-scale problems.

- Novel activations/architectures. It modifies the form of surrogate neural networks $u_\theta$ for better model capacity. We see that these modifications are effective for some problems with complex geometries and nonlinear NS equations.

## C  Model Configuration and Hyperparameters

### C.1  Model architecture

Our research employs a specific model structure: a Multilayer Perceptron (MLP) with 5 layers, each of which has a width of 100 neurons.

The model was trained for a total of 20,000 iterations or epochs. This number of training rounds was found to be sufficient for the model to learn the underlying patterns in the data, while also avoiding potential overfitting that might occur with too many epochs.

As for the number of collocation points, for 2-dimensional problems, we used 8192 points. These collocation points provide dense coverage of the problem space while it does not consume too much GPU memory. In addition to these, we utilized 2048 boundary/initial points.

For 3-dimensional problems, the number of collocation points and boundary/initial points were increased to 32768 and 8192, respectively. This increase corresponds to the added complexity of 3-dimensional problems, requiring a more comprehensive representation of the problem space to achieve reliable and accurate results.

### C.2  Optimization hyperparameters

In our primary experiment, we use Adam optimizer with momentum $(0.9, 0.999)$. We set the learning rate at 1e-3. This learning rate was selected after carefully considering the trade-off between the speed of convergence and the stability of learning, which we discussed previously. We found that this learning rate provides a good balance, enabling robust learning without the issues associated with excessively high or low rates. For vanilla PINNs, the loss weights are set to 1.

In summary, our model structure and parameters were carefully selected to balance the need for accuracy and computational efficiency, providing a fair and effective comparison in our study. Detailed ablation studies about these hyperparameters are reported in Appendix E.

### C.3  Other method-specific hyperparameters

Here we present the hyperparameters of the methods we tested.

- **PINN.** There are no special hyperparameters for the baseline PINN. Please refer to the section above for the network structure and optimization hyperparameters.

- **PINN-w.** We assign larger weights to boundary conditions for PINN-w. Specifically, the weight for PDE loss is set at 1, while those for initial and boundary conditions are increased to 100. These losses are then aggregated as the target loss.

- **PINN-LRA.** We set $\alpha = 0.1$ for updating loss weights, which is the recommended value in the original paper.

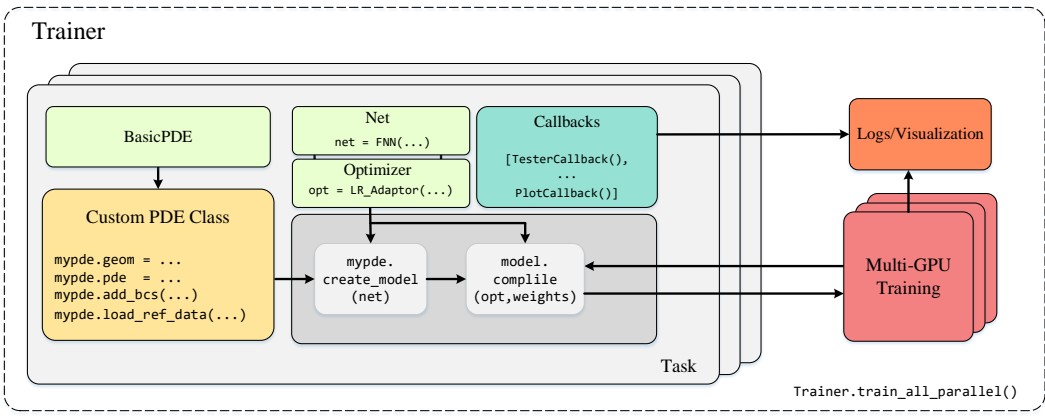

Figure 19: A high-level illustration of PINNacle code structure.

- **PINN-NTK.** No special hyperparameter is needed for this method.

- **RAR.** For residual-based adaptive refinement, we add new points where the residual is greatest into the training set every 2000 epochs.

- **MultiAdam.** Although there is no manual weighting for MultiAdam, the loss grouping criteria can affect its performance. Due to time constraints, we only tuned the grouping criteria for the Wave1d-C case, where losses were divided into Dirichlet boundary losses and non-Dirichlet losses and trained for 10,000 epochs. For all other cases, we simply categorize the losses into PDE and boundary losses.

- **gPINN.** For simplicity, we assign a weight of $0.01$ to the gradient terms and a weight of $1$ to all others. However, these weights are delicate and require further fine-tuning.

## D High-level Structure of Toolbox

In Figure 19, we provide a high-level overview of the usage and modules of the benchmark. We provide several encapsulated classes upon DeepXDE. Specifically, we have a PDE class for building PDE problems conveniently. Then we warp the model class by passing neural network architecture, optimizer, and custom callbacks. After that, the model is compiled by DeepXDE. Finally, we invoke the multi-GPU parallel training and evaluation framework to allocate the training tasks to different GPUs. We support convenient one-button parallel training and testing on all PDE cases using all methods. An example code snippet is shown here.

```python
import deepxde as dde
from trainer import Trainer
from src.pde import PDE1, ..., PDEn
from src.utils.callbacks import TesterCallback

trainer = Trainer('experiment-name',device)
for pde_class in [PDE1, ..., PDEn]:
    def get_model():
        pde = pde_class()
        net = dde.nn.FNN([pde.input_dim] + n_layers * [n_hidden] + [pde.output_dim])
        opt = torch.optim.Adam(net.parameters(), lr=learning_rate)
        model = pde.create_model(net)
        model.compile(opt)
        return model

    trainer.add_task(
        get_model, {'iterations': num_iterations,'callbacks': [TesterCallback()]}
    )
trainer.train_all_parallel()
```

# E  Detailed Experimental Results

## E.1  Detailed results of main experiments.

The detailed results of the main experiments in listed in the subsection. In Table 9, we provide the mean and std of L2RE for all baselines on all PDEs. In Table 8, we provide the mean and std of L1RE for all baselines on all PDEs. In Table 11, Table 12, and Table 13, we provide the low-frequency, medium-frequency, and high-frequency Fourier errors, respectively. In Table 10, we provide the mean and std of MSE for all baselines on all PDEs. In Table 14, we provide the average runtime (seconds) for all baselines trained with 20000 epochs on all PDEs averaged by three runs. In Table E.2, we show the results of all baselines on parametric PDEs.

Here we provide an analysis of these results. Since the results of the main experiments have been described in the main text, we won't go over them again. For different metrics of the same PDE, the best-performing methods often differ. This is because different errors reflect different mismatches between the predicted solution and the true solution.

- From the results, we can see that for most cases, methods that perform well in L2RE error also perform well in L1RE. This shows that L1RE and L2RE are generally similar. Although the absolute values differ, they can mostly be used interchangeably, or one can be chosen for calculation.

- Max error measures the worst-case error, significantly different from the average loss measured by L1RE/L2RE. From the results, we can see that hp-VPINN performs very well on this metric, followed by the adaptive activation function LAAF. PINN-LRA and PINN-NTK are optimal for some equations, but their effects are not as stable.

- Fourier error allows for the convergence of different frequency components, so it's an essential reference indicator. Since functions defined in irregular geometric areas are not suitable for calculating Fourier error, we ignored these equations. Looking at **Table 9, Table 10, and Table 11** comprehensively, for mid-low frequency functions, FBPINN is the best performing in most instances. Loss reweighting methods like PINN-LRA and ordinary PINN are better for low and high-frequency components, respectively. We speculate that reweighting the loss to some extent changes the convergence order of different function components.

- Regarding the runtime metric, hp-VPINN is the fastest in most problems. This might be due to the optimization inherent in hp-VPINN's implementation and its fewer required differentiations than vanilla PINN. All other methods introduced varying degrees of additional computational overhead compared to vanilla PINN, with some methods like gPINN even requiring about twice the computational time. We list all training and inference Flops in Table 15 and Table 18. The flops metric also shows that vanilla PINNs and hp-VPINNs are the most efficient PINN variants.

| L2RE | Name | Vanilla | | Loss Reweighting/Sampling | | | Optimizer | Loss functions | | Architecture | | |
|---|---|---|---|---|---|---|---|---|---|---|---|---|
| – | | PINN | PINN-w | LRA | NTK | RAR | MultiAdam | gPINN | vPINN | LAAF | GAAF | FBPINN |
| Burgers | 1d-C | 1.45E-2(1.59E-3) | 2.63E-2(4.68E-3) | 2.61E-2(1.18E-2) | 1.84E-2(3.66E-3) | 3.32E-2(2.14E-2) | 4.85E-2(1.61E-2) | 2.16E-1(3.34E-2) | 3.47E-1(3.49E-2) | **1.43E-2(1.44E-3)** | 5.20E-2(2.08E-2) | 2.32E-1(9.14E-2) |
| | 2d-C | 3.24E-1(7.54E-4) | 2.70E-1(3.93E-3) | **2.60E-1(5.78E-3)** | 2.75E-1(4.78E-3) | 3.45E-1(4.56E-5) | 3.33E-1(8.65E-3) | 3.27E-1(1.25E-4) | 6.38E-1(1.47E-2) | 2.77E-1(1.39E-2) | 2.95E-1(1.17E-2) | – |
| Poisson | 2d-C | 6.94E-1(8.78E-3) | 3.49E-2(6.91E-3) | 1.17E-1(1.26E-1) | **1.23E-2(7.37E-3)** | 6.99E-1(7.46E-3) | 2.63E-2(6.57E-3) | 6.87E-1(1.87E-2) | 4.91E-1(1.55E-2) | 7.68E-1(4.70E-2) | 6.04E-1(7.52E-2) | 4.49E-2(7.91E-3) |
| | 2d-CG | 6.36E-1(2.57E-3) | 6.08E-2(4.88E-3) | 4.34E-2(7.95E-3) | **1.43E-2(4.31E-3)** | 6.48E-1(7.87E-3) | 2.76E-1(1.03E-1) | 7.92E-1(4.56E-3) | 2.86E-1(2.00E-3) | 4.80E-1(1.43E-2) | 8.71E-1(2.67E-1) | 2.90E-2(3.92E-3) |
| | 3d-CG | 5.60E-1(2.84E-2) | 3.74E-1(3.23E-2) | **1.02E-1(3.16E-2)** | 9.47E-1(4.94E-4) | 5.76E-1(5.40E-2) | 3.63E-1(7.81E-2) | 4.85E-1(5.70E-2) | 7.38E-1(6.47E-4) | 5.79E-1(2.65E-2) | 5.02E-1(7.47E-2) | 7.39E-1(7.24E-2) |
| | 2d-MS | 6.30E-1(1.07E-2) | 7.60E-1(6.96E-3) | 7.94E-1(6.51E-2) | 7.48E-1(9.94E-3) | 6.44E-1(2.13E-2) | **5.90E-1(4.06E-2)** | 6.16E-1(1.74E-2) | 9.72E-1(2.23E-2) | 5.93E-1(1.18E-1) | 9.31E-1(7.12E-2) | 1.04E+0(6.13E-5) |
| Heat | 2d-VC | 1.01E+0(6.34E-2) | 2.35E-1(1.70E-2) | **2.12E-1(8.61E-4)** | 2.14E-1(5.82E-3) | 9.66E-1(1.86E-2) | 4.75E-1(8.44E-2) | 2.12E+0(5.51E-1) | 9.40E-1(1.73E-1) | 6.42E-1(6.32E-2) | 8.49E-1(1.06E-1) | 9.52E-1(2.29E-3) |
| | 2d-MS | 6.21E-2(1.38E-2) | 2.42E-1(2.67E-2) | 8.79E-2(2.56E-2) | **4.40E-2(4.81E-3)** | 7.49E-2(1.05E-2) | 2.18E-1(9.26E-2) | 1.13E-1(3.08E-3) | 9.30E-1(2.06E-2) | 7.40E-2(1.92E-2) | 9.85E-1(1.04E-1) | 8.20E-2(4.87E-3) |
| | 2d-CG | 3.64E-2(8.82E-3) | 1.45E-1(4.77E-3) | 1.25E-1(4.30E-3) | 1.16E-1(1.21E-2) | 2.72E-2(3.22E-3) | 7.12E-2(1.30E-2) | 9.38E-2(1.45E-2) | 1.67E+0(3.62E-3) | **2.39E-2(1.39E-3)** | 4.61E-1(2.63E-1) | 9.16E-2(3.29E-2) |
| | 2d-LT | 9.99E-1(1.05E-5) | 9.99E-1(8.01E-5) | 9.99E-1(7.37E-5) | 1.00E+0(2.82E-4) | 9.99E-1(1.56E-4) | 1.00E+0(3.85E-5) | 1.00E+0(9.82E-5) | 1.00E+0(0.00E+0) | 9.99E-1(4.49E-4) | 9.99E-1(2.20E-4) | 1.01E+0(1.23E-4) |
| NS | 2d-C | 4.70E-2(1.12E-3) | 1.45E-1(1.21E-2) | NaN(NaN) | 1.98E-2(2.60E-2) | 4.69E-1(1.16E-2) | 7.27E-1(1.95E-1) | 7.70E-2(2.99E-3) | 2.92E-1(8.24E-2) | **3.60E-2(3.87E-3)** | 3.79E-2(4.32E-3) | 8.45E-2(2.26E-2) |
| | 2d-CG | 1.19E-1(5.46E-3) | 3.26E-1(7.69E-3) | 3.32E-1(7.60E-3) | 2.93E-1(2.02E-2) | 3.34E-1(6.52E-4) | 4.31E-1(6.95E-2) | 1.54E-1(5.89E-3) | 9.94E-1(3.80E-3) | **8.24E-2(8.21E-3)** | 1.74E-1(7.00E-2) | 8.27E+0(3.68E-5) |
| | 2d-LT | 9.96E-1(1.19E-3) | 1.00E+0(3.34E-4) | 1.00E+0(4.05E-4) | 9.99E-1(6.04E-4) | 1.00E+0(3.35E-4) | 1.00E+0(2.19E-4) | 9.95E-1(7.19E-4) | 1.73E+0(1.00E-5) | 9.98E-1(3.42E-3) | 9.99E-1(1.10E-3) | 1.00E+0(2.07E-3) |
| Wave | 1d-C | 5.88E-1(9.63E-2) | 2.85E-1(8.97E-3) | 3.61E-1(1.95E-2) | **9.79E-2(7.72E-3)** | 5.39E-1(1.77E-2) | 1.21E-1(1.76E-2) | 5.56E-1(1.67E-2) | 8.39E-1(5.94E-2) | 4.54E-1(1.08E-2) | 6.77E-1(1.05E-1) | 5.91E-1(4.74E-2) |
| | 2d-CG | 1.84E+0(3.40E-1) | 1.66E+0(7.39E-2) | 1.48E+0(1.03E-1) | 2.16E+0(1.01E-1) | 1.15E+0(1.06E-1) | 1.09E+0(1.24E-1) | 8.14E-1(1.18E-2) | 7.99E-1(4.31E-2) | 8.19E-1(2.67E-2) | **7.94E-1(9.33E-3)** | 1.06E+0(7.54E-2) |
| | 2d-MS | 1.34E+0(2.34E-1) | 1.02E+0(1.16E-2) | 1.02E+0(1.36E-2) | 1.04E+0(3.11E-2) | 1.35E+0(2.43E-1) | 1.01E+0(5.64E-3) | 1.02E+0(4.00E-3) | 9.82E-1(1.23E-3) | 1.06E+0(1.71E-2) | 1.06E+0(5.35E-2) | 1.03E+0(6.68E-3) |
| Chaotic | GS | 3.19E-1(3.18E-1) | 1.58E-1(9.10E-2) | 9.37E-2(4.42E-5) | 2.16E-1(7.73E-2) | 9.46E-2(9.46E-4) | 9.37E-2(1.21E-5) | 2.48E-1(1.10E-1) | 1.16E+0(1.43E-1) | 9.47E-2(7.07E-5) | 9.46E-2(1.15E-4) | **7.99E-2(1.69E-2)** |
| | KS | 1.01E+0(1.28E-3) | 9.86E-1(2.24E-2) | 9.57E-2(2.85E-3) | 9.64E-1(4.94E-3) | 1.01E+0(8.63E-4) | 9.61E-1(4.77E-3) | 9.94E-1(3.83E-3) | 9.72E-1(5.80E-4) | 1.01E+0(2.12E-3) | 1.00E+0(1.24E-2) | 1.02E+0(2.31E-2) |
| High dim | PNd | 3.04E-3(5.62E-4) | 2.58E-3(1.31E-3) | **4.58E-4(1.89E-5)** | 4.64E-3(4.36E-3) | 3.59E-3(1.25E-3) | 3.98E-3(1.11E-3) | 5.05E-3(6.07E-4) | – | 4.14E-3(5.59E-4) | 7.75E-3(1.41E-3) | – |
| | HNd | 3.61E-1(4.40E-3) | 4.59E-1(4.34E-3) | 3.94E-1(1.28E-2) | 3.97E-1(1.26E-2) | 3.57E-1(3.69E-3) | **3.02E-1(4.07E-2)** | 3.17E-1(6.66E-3) | – | 5.22E-1(3.12E-3) | 5.21E-1(7.79E-4) | – |
| Inverse | PInv | 9.42E-2(1.58E-3) | 1.66E-1(5.45E-3) | 1.54E-1(3.32E-3) | 1.93E-1(1.39E-2) | 9.35E-2(1.12E-2) | 1.30E-1(1.55E-2) | 8.03E-2(2.79E-3) | **2.45E-2(1.03E-2)** | 1.30E-1(1.07E-2) | 2.54E-1(1.53E-1) | 8.44E-1(1.37E-1) |
| | HInv | 1.57E+0(7.21E-2) | 5.26E-2(3.31E-3) | **5.09E-2(4.34E-3)** | 7.52E-2(5.42E-3) | 1.52E+0(6.46E-2) | 8.04E-2(1.20E-2) | 4.84E+0(2.07E+0) | 4.56E-1(1.30E-2) | 5.59E-1(5.24E-1) | 2.12E-1(4.89E-2) | 9.27E-1(1.20E-1) |

Table 7: Mean (Std) of L2RE for main experiments.

| L1RE | Name | Vanilla | | Loss Reweighting/Sampling | | | Optimizer | Loss functions | | Architecture | | |
|---|---|---|---|---|---|---|---|---|---|---|---|---|
| – | | PINN | PINN-w | LRA | NTK | RAR | MultiAdam | gPINN | vPINN | LAAF | GAAF | FBPINN |
| Burgers | 1d-C | **9.55E-3(6.42E-4)** | 1.88E-2(4.05E-3) | 1.35E-2(2.57E-3) | 1.30E-2(1.73E-3) | 1.35E-2(4.66E-3) | 2.64E-2(5.69E-3) | 1.42E-1(1.98E-2) | 4.02E-2(6.41E-3) | 1.40E-2(3.68E-3) | 1.95E-2(8.30E-3) | 3.75E-2(9.70E-3) |
| | 2d-C | 2.96E-1(7.40E-4) | 2.43E-1(2.98E-3) | 2.31E-1(7.16E-3) | **2.48E-1(5.33E-3)** | 3.27E-1(3.73E-5) | 3.12E-1(1.15E-2) | 3.01E-1(3.55E-4) | 6.56E-1(3.01E-2) | 2.57E-1(2.06E-2) | 2.67E-1(1.22E-2) | – |
| Poisson | 2d-C | 7.40E-1(5.49E-3) | 3.08E-2(5.13E-3) | 7.82E-2(7.47E-2) | **1.30E-2(8.23E-3)** | 7.48E-1(1.01E-2) | 2.47E-1(6.38E-3) | 7.35E-1(2.08E-2) | 4.60E-1(1.39E-2) | 7.67E-1(1.36E-2) | 6.57E-1(3.99E-2) | 5.01E-2(4.71E-3) |
| | 2d-CG | 5.45E-1(4.71E-3) | 4.54E-2(6.42E-3) | 2.63E-2(5.50E-3) | **1.33E-2(4.96E-3)** | 5.60E-1(8.19E-3) | 2.46E-1(1.07E-1) | 7.31E-1(2.77E-3) | 2.45E-1(5.14E-3) | 4.04E-1(1.03E-2) | 7.09E-1(2.12E-1) | 3.21E-2(6.23E-3) |
| | 3d-CG | 4.51E-1(3.35E-2) | 3.33E-1(2.64E-2) | **7.76E-2(1.63E-2)** | 9.93E-1(2.91E-4) | 4.61E-1(4.46E-2) | 3.55E-1(7.75E-2) | 4.57E-1(5.07E-2)) | 7.96E-1(3.57E-4) | 4.60E-1(1.13E-2) | 3.82E-1(4.89E-2) | 6.91E-1(7.52E-2) |
| | 2d-MS | 7.60E-1(1.06E-2) | 7.49E-1(1.12E-2) | 7.93E-1(7.62E-2) | 7.26E-1(1.46E-2) | 7.84E-1(2.42E-2) | 6.94E-1(5.61E-2) | 7.41E-1(2.01E-2) | 9.61E-1(5.67E-2) | **6.31E-1(5.42E-2)** | 9.04E-1(1.01E-1) | 9.94E-1(9.67E-5) |
| Heat | 2d-VC | 1.12E+0(5.79E-2) | 2.41E-1(1.73E-2) | 2.07E-1(1.04E-3) | **2.03E-1(1.12E-2)** | 1.06E+0(5.13E-2) | 5.45E-1(1.07E-1) | 2.41E+0(5.27E-1) | 8.79E-1(2.57E-1) | 7.49E-1(8.54E-2) | 9.91E-1(1.37E-1) | 9.44E-1(1.75E-3) |
| | 2d-MS | 9.30E-2(2.27E-2) | 2.90E-1(2.43E-2) | 1.13E-1(3.57E-2) | 6.69E-2(8.24E-3) | 1.19E-1(2.16E-2) | 3.00E-1(1.14E-1) | 1.80E-1(1.12E-2) | 9.25E-1(3.90E-2) | 1.14E-1(4.98E-2) | 1.08E+0(2.02E-1) | **5.33E-2(3.92E-3)** |
| | 2d-CG | 3.05E-2(8.47E-3) | 1.37E-1(7.70E-3) | 1.12E-1(2.57E-3) | 1.07E-1(1.44E-2) | 2.21E-2(3.42E-3) | 5.88E-2(1.02E-2) | 8.20E-2(1.32E-2) | 3.09E+0(1.86E-2) | **1.94E-2(1.98E-3)** | 3.77E-1(2.17E-1) | 6.77E-3(3.93E-2) |
| | 2d-LT | 9.98E-1(6.00E-5) | 9.98E-1(1.42E-4) | 9.98E-1(1.47E-4) | 9.99E-1(1.01E-3) | 9.98E-1(2.28E-4) | 9.99E-1(5.69E-5) | 9.98E-1(8.62E-4) | 9.98E-1(0.00E+0) | 9.98E-1(1.27E-4) | 9.98E-1(8.58E-5) | 1.01E+0(7.75E-4) |
| NS | 2d-C | 5.08E-2(3.06E-3) | 1.84E-1(1.52E-2) | NaN | 2.44E-1(3.05E-2) | 5.54E-1(1.24E-2) | 9.86E-1(3.16E-1) | 9.43E-2(3.24E-3) | 1.98E-1(7.81E-2) | 4.42E-2(7.38E-3) | **3.78E-2(8.71E-3)** | 1.18E-1(3.10E-1) |
| | 2d-CG | 1.77E-1(1.00E-2) | 4.22E-1(8.72E-3) | 4.12E-1(6.93E-3) | 3.69E-1(2.46E-2) | 4.65E-1(4.44E-3) | 6.23E-1(8.86E-2) | 2.36E-1(1.15E-2) | 9.95E-1(3.50E-4) | **1.25E-1(1.42E-2)** | 2.40E-1(8.01E-2) | 5.92E+0(5.65E-4) |
| | 2d-LT | 9.88E-1(1.86E-3) | 9.98E-1(4.68E-4) | 9.97E-1(3.64E-4) | 9.95E-1(6.66E-4) | 1.00E+0(2.46E-4) | 9.99E-1(9.27E-4) | 9.90E-1(3.60E-4) | 1.00E+0(1.40E-4) | 9.90E-1(3.78E-3) | 9.96E-1(2.68E-3) | 1.00E+0(1.38E-3) |
| Wave | 1d-C | 5.87E-1(9.20E-2) | 2.78E-1(8.86E-3) | 3.49E-1(2.02E-2) | **9.42E-2(9.13E-3)** | 5.40E-1(1.74E-2) | 1.15E-1(1.91E-2) | 5.60E-1(1.69E-2) | 1.41E+0(1.30E-1) | 4.38E-1(1.40E-2) | 6.82E-1(1.08E-1) | 6.55E-1(4.86E-2) |
| | 2d-CG | 1.96E+0(3.83E-1) | 1.78E+0(8.89E-2) | 1.58E+0(1.15E-1) | 2.34E+0(1.14E-1) | 1.16E+0(1.16E-1) | 1.09E+0(1.54E-1) | 7.22E-1(1.63E-2) | 1.08E+0(1.25E-1) | 7.45E-1(2.15E-2) | **7.08E-1(9.13E-3)** | 1.15E+0(1.03E-1) |
| | 2d-MS | 2.04E+0(7.38E-1) | 1.10E+0(4.25E-2) | 1.08E+0(6.01E-2) | 1.13E+0(4.91E-2) | 2.08E+0(7.45E-1) | 1.07E+0(1.40E-2) | 1.11E+0(1.91E-2) | 1.05E+0(1.00E-2) | 1.17E+0(4.66E-2) | 1.12E+0(8.62E-2) | 1.29E+0(2.81E-1) |
| Chaotic | GS | 3.45E-1(4.57E-1) | 1.29E-1(1.54E-1) | 2.01E-2(5.99E-5) | 1.11E-1(4.79E-2) | 2.98E-2(6.44E-3) | **2.00E-2(6.12E-5)** | 2.72E-1(1.79E-1) | 1.04E+0(3.04E-1) | 2.07E-2(9.19E-4) | 1.16E-1(1.31E-1) | 5.06E-2(1.87E-2) |
| | KS | 9.44E-1(8.57E-4) | 8.95E-1(2.99E-2) | **8.60E-1(3.48E-3)** | 8.64E-1(3.31E-3) | 9.42E-1(8.75E-4) | 8.73E-1(8.40E-3) | 9.36E-1(6.12E-3) | 8.88E-1(9.92E-3) | 9.39E-1(3.25E-3) | 9.44E-1(9.86E-3) | 9.85E-1(3.35E-2) |
| High dim | PNd | 2.40E-3(3.44E-4) | 2.34E-3(1.27E-3) | **3.17E-4(9.16E-6)** | 4.58E-3(4.56E-3) | 2.98E-3(1.24E-3) | 3.40E-3(8.71E-4) | 4.43E-3(8.45E-4) | – | 4.33E-3(1.88E-3) | 5.72E-3(1.57E-3) | – |
| | HNd | 2.25E-1(3.87E-3) | 3.27E-1(5.13E-3) | 2.63E-1(1.30E-2) | 2.64E-1(1.59E-2) | 2.24E-1(2.56E-3) | **1.58E-1(2.71E-2)** | 1.83E-1(5.99E-3) | – | 3.42E-1(3.32E-3) | 3.40E-1(5.24E-3) | – |
| Inverse | PInv | 8.30E-2(6.88E-4) | 1.14E-1(3.56E-3) | 1.14E-1(6.95E-3) | 1.33E-1(1.01E-2) | 8.35E-2(9.53E-3) | 1.13E-1(1.64E-2) | 7.33E-2(2.49E-3) | **1.96E-2(7.75E-3)** | 8.12E-1(1.01E+0) | 2.18E-1(1.20E-1) | 8.39E-1(1.39E-1) |
| | HInv | 1.06E+0(5.39E-2) | 4.16E-2(3.18E-3) | **3.94E-2(1.52E-3)** | 5.96E-2(2.54E-3) | 1.01E+0(5.68E-2) | 6.29E-2(8.58E-3) | 3.51E+0(1.59E+0) | 4.59E-1(1.22E-3) | 3.93E-1(3.32E-1) | 1.89E-1(6.30E-2) | 8.46E-1(7.18E-2) |

Table 8: Mean (Std) of L1RE for main experiments.

| mERR | Name | Vanilla | Loss Reweighting/Sampling | | | Optimizer | Loss functions | | Architecture | | |
|---|---|---|---|---|---|---|---|---|---|---|---|
| – | | PINN | LRA | NTK | RAR | MultiAdam | gPINN | vPINN | LAAF | GAAF | FBPINN |
| Burgers | 1d-C | 9.03E-02(6.76E-03) | 1.53E-01(5.64E-02) | **1.33E-01(7.85E-02)** | 3.38E-01(2.82E-01) | 4.02E-01(2.04E-01) | 9.47E-01(1.88E-02) | 1.84E+00(1.53E-02) | 2.58E-01(2.96E-01) | 1.88E-01(6.02E-02) | 1.34E+00(4.98E-1) |
| | 2d-C | 4.32E+00(8.01E-02) | 3.84E+00(1.96E-01) | 4.07E+00(6.99E-02) | 4.47E+00(1.32E-01) | 3.99E+00(1.33E-01) | **3.83E+00(6.03E-03)** | 9.12E+00(3.83E+00) | 4.11E+00(1.99E-01) | 4.14E+00(1.11E-01) | - |
| Poisson | 2d-C | 9.41E-01(9.40E-02) | 5.93E-01(3.87E-01) | **2.94E-02(1.37E-02)** | 9.27E-01(9.67E-02) | 3.99E-01(3.67E-01) | 7.99E-01(2.80E-02) | 2.09E-01(1.30E-02) | 8.26E-01(2.78E-02) | 5.02E-01(2.37E-03) | 1.71E-1(1.52E-2) |
| | 2d-CG | 1.63E+00(1.76E-02) | 3.49E-01(2.49E-01) | **6.81E-02(3.06E-02)** | 1.65E+00(1.32E-02) | 5.84E-01(7.86E-02) | 1.67E+00(5.90E-03) | 1.62E+00(3.37E-03) | 1.61E+00(2.51E-02) | 1.49E+00(1.45E-01) | 1.98E-1(3.14E-2) |
| | 3d-CG | 1.04E+00(5.37E-02) | 3.91E-01(1.40E-01) | 1.12E+00(7.37E-04) | 1.09E+00(1.11E-01) | 6.88E-01(1.51E-01) | 7.87E-01(1.34E-01) | **1.59E-1(7.13E-5)** | 1.14E+00(3.77E-02) | 1.21E+00(2.49E-01) | 1.07E+00(2.63E-02) |
| | 2d-MS | 4.87E+00(2.10E-01) | 9.58E+00(2.05E-01) | 9.66E+00(1.86E-02) | 4.96E+00(2.90E-01) | 5.88E+00(6.69E-01) | 5.01E+00(2.61E-01) | 9.87E+00(1.20E-03) | **4.40E+00(4.58E-01)** | 8.77E+00(2.15E+00) | 9.87E+00(5.44E-4) |
| Heat | 2d-VC | 9.93E-01(7.20E-02) | **2.63E-01(8.90E-03)** | 2.67E-01(1.74E-02) | 1.03E+00(7.73E-02) | 4.73E-01(1.07E-01) | 4.46E+00(1.05E+00) | 8.83E-01(3.35E-01) | 7.79E-01(8.19E-02) | 7.85E-01(2.12E-01) | 7.78E-1(1.11E-3) |
| | 2d-MS | 9.10E-02(3.20E-02) | 1.60E-01(5.65E-02) | 6.65E-02(2.50E-02) | **4.36E-02(1.28E-02)** | 1.58E-01(8.85E-02) | 7.10E-01(3.05E-01) | 3.69E-01(1.00E-03) | 5.53E-02(1.20E-02) | 8.35E-02(4.70E-02) | 1.81E-1(4.94E-3) |
| | 2d-CG | 9.40E-01(7.48E-02) | **6.40E-01(3.70E-02)** | 1.14E+00(1.21E-01) | 9.00E-01(1.47E-01) | 1.39E+00(2.32E-01) | 2.20E+00(2.95E-01) | 4.38E+00(3.48E-01) | 9.59E-01(5.39E-02) | 3.18E+00(4.99E-01) | 2.83E+00(3.63E-1) |
| | 2d-LT | 2.18E+00(6.95E-01) | 1.82E+00(1.60E-02) | 1.83E+00(1.40E-02) | 1.85E+00(1.16E-02) | 1.82E+00(2.40E-02) | 5.46E+00(6.13E+00) | 3.09E+00(3.46E-01) | 1.84E+00(1.51E-02) | **1.81E+00(7.94E-03)** | 3.32E+00(6.15E-2) |
| NS | 2d-C | 2.26E-01(6.33E-03) | nan(nan) | 2.65E-01(3.05E-02) | 2.22E-01(1.42E-02) | 5.67E-01(6.28E-02) | 4.73E-01(3.17E-02) | **1.80E-01(1.64E-02)** | 1.84E-01(5.41E-03) | 1.99E-01(9.09E-03) | 2.00E-1(4.73E-2) |
| | 2d-CG | 2.06E-01(6.69E-03) | 4.97E-01(9.10E-02) | 3.33E-01(3.92E-02) | 2.11E-01(4.38E-03) | 6.23E-01(1.87E-01) | 2.94E-01(9.84E-03) | 4.31E+00(1.47E-02) | **1.68E-01(2.34E-03)** | 1.80E-01(7.47E-03) | 8.00E+00(0.00E+00) |
| | 2d-LT | 1.17E+02(5.00E-01) | 1.21E+02(2.00E-01) | 1.21E+02(6.51E-01) | 1.18E+02(7.69E-01) | 1.21E+02(2.40E-01) | 1.21E+02(5.69E-01) | 1.23E+02(5.54E-01) | **1.18E+02(6.76E-01)** | 1.19E+02(5.28E-01) | 1.24E+02(7.76E-1) |
| Wave | 1d-C | 9.34E-01(1.16E-01) | 5.17E-01(6.11E-02) | **2.75E-01(2.22E-02)** | 8.16E-01(6.80E-02) | 1.26E+00(1.89E-01) | 1.28E+00(6.21E-02) | 6.17E-01(5.41E-02) | 7.40E-01(7.71E-02) | 1.18E+00(3.23E-01) | 8.51E-1(1.11E-1) |
| | 2d-CG | 2.00E+00(9.89E-02) | 1.95E+00(1.26E-01) | 2.00E+00(1.80E-02) | 1.93E+00(8.80E-02) | 1.71E+00(5.74E-02) | 1.73E+00(2.81E-03) | 1.66E+00(2.19E-02) | 1.93E+00(1.48E-01) | 1.88E+00(1.13E-01) | **1.65E+00(2.44E-2)** |
| | 2d-MS | 1.44E+03(2.92E+02) | 1.95E+03(3.91E+02) | 1.74E+03(2.15E+02) | 1.30E+03(2.72E+02) | 1.05E+03(4.29E+01) | 1.09E+03(4.19E+01) | 4.43E+02(4.24E+00) | 1.80E+03(8.80E+01) | 1.45E+03(4.66E+02) | 5.59E+03(1.55E+02) |
| Chaotic | GS | 3.66E+00(1.00E-01) | 3.48E+00(8.97E-02) | 3.61E+00(6.38E-02) | 3.60E+00(6.85E-02) | 3.41E+00(1.27E-01) | 3.41E+00(3.54E-02) | 8.93E-01(6.51E-02) | 3.76E+00(5.27E-02) | 3.41E+00(1.28E-01) | **8.36E-1(8.16E-2)** |
| | KS | 9.84E-01(1.64E-03) | 9.83E-01(3.76E-04) | **8.76E-01(1.72E-01)** | 9.83E-01(7.11E-04) | 9.82E-01(1.42E-04) | 9.84E-01(4.09E-03) | 3.33E+00(7.80E-02) | 9.83E-01(6.72E-04) | 9.83E-01(3.76E-04) | 3.30E+00(4.74E-2) |
| High dim | PNd | 2.96E-02(1.57E-02) | **4.05E-03(9.49E-04)** | 4.99E-03(4.48E-03) | 2.72E-02(1.17E-02) | 3.96E-02(2.29E-02) | 3.16E-02(1.21E-02) | – | 5.90E-02(4.88E-02) | 1.76E+00(8.43E-01) | – |
| | HNd | 5.18E-02(2.21E-02) | 1.29E-01(1.94E-01) | 6.32E-02(3.49E-02) | 4.64E-02(1.59E-02) | 7.92E-03(3.01E-03) | 5.02E-02(5.95E-03) | – | **2.04E-02(1.22E-02)** | 1.27E+00(1.45E+00) | – |

Table 9: Mean (Std) of max error for main experiments.

| MSE | Name | Vanilla | | Loss Reweighting/Sampling | | | Optimizer | Loss functions | | Architecture | | |
|---|---|---|---|---|---|---|---|---|---|---|---|---|
| – | | PINN | PINN-w | LRA | NTK | RAR | MultiAdam | gPINN | vPINN | LAAF | GAAF | FBPINN |
| Burgers | 1d-C | **7.90E-5(1.78E-5)** | 2.64E-4(8.69E-5) | 3.03E-4(2.62E-4) | 1.30E-4(5.19E-5) | 5.78E-4(6.31E-4) | 9.68E-4(5.51E-4) | 1.77E-2(5.58E-3) | 5.13E-3(1.90E-3) | 1.80E-4(1.35E-4) | 3.00E-4(1.56E-4) | 1.53E-2(1.03E-2) |
| | 2d-C | 1.69E-1(7.86E-4) | 1.17E-1(3.41E-3) | **1.09E-1(4.84E-3)** | 1.22E-1(4.22E-3) | 1.92E-1(5.07E-5) | 1.79E-1(9.36E-3) | 1.72E-1(1.31E-4) | 7.08E-1(5.16E-2) | 1.26E-1(1.54E-2) | 1.41E-1(1.12E-2) | – |
| Poisson | 2d-C | 1.17E-1(2.98E-3) | 3.09E-4(1.25E-4) | 7.24E-3(9.95E-3) | **5.00E-5(5.33E-5)** | 1.19E-1(2.55E-3) | 1.79E-4(8.84E-5) | 1.15E-1(6.22E-3) | 4.86E-2(4.43E-3) | 1.39E-1(5.67E-3) | 9.38E-2(1.91E-2) | 7.89E-4(2.17E-4) |
| | 2d-CG | 1.28E-1(1.03E-3) | 1.17E-3(1.83E-4) | 6.13E-4(2.31E-4) | **6.99E-5(3.50E-5)** | 1.32E-1(3.23E-3) | 2.73E-2(1.92E-2) | 1.98E-1(2.28E-3) | 2.50E-2(3.80E-4) | 7.67E-2(2.73E-3) | 1.77E-1(8.70E-2) | 4.84E-4(9.87E-5) |
| | 3d-CG | 2.64E-2(2.67E-3) | 1.18E-2(1.97E-3) | **9.51E-4(6.51E-4)** | 7.54E-2(7.86E-5) | 2.81E-2(5.15E-3) | 1.16E-2(4.42E-3) | 2.01E-2(4.93E-3) | 4.58E-2(8.04E-5) | 2.82E-2(2.62E-3) | 2.16E-2(5.87E-3) | 4.63E-2(9.28E-3) |
| | 2d-MS | 2.67E+0(9.04E-2) | 3.90E+0(7.16E-2) | 4.28E+0(6.83E-1) | 3.77E+0(9.98E-2) | 2.80E+0(1.87E-1) | 2.36E+0(3.15E-1) | 2.56E+0(1.43E-1) | 6.09E+0(5.46E-1) | **1.83E+0(3.00E-1)** | 5.87E+0(8.72E-1) | 6.68E+0(8.23E-4) |
| Heat | 2d-VC | 4.00E-2(4.94E-3) | 2.19E-3(3.21E-4) | **1.76E-3(1.43E-5)** | 1.79E-3(9.80E-5) | 3.67E-2(1.42E-3) | 9.14E-3(3.13E-3) | 1.89E-1(9.44E-2) | 3.23E-2(2.26E-2) | 1.74E-2(4.35E-3) | 2.93E-2(7.12E-3) | 3.56E-2(1.71E-4) |
| | 2d-MS | 1.09E-4(4.94E-5) | 1.60E-3(3.35E-4) | 2.25E-4(1.22E-4) | **5.27E-5(1.18E-5)** | 1.54E-4(4.17E-5) | 1.51E-3(1.25E-3) | 3.43E-4(1.87E-5) | 2.57E-2(2.22E-3) | 1.57E-4(8.06E-5) | 3.10E-2(1.15E-2) | 2.17E-4(2.47E-5) |
| | 2d-CG | 2.09E-3(9.69E-4) | 3.15E-2(2.08E-3) | 2.32E-2(1.59E-3) | 2.02E-2(4.15E-3) | 1.12E-3(2.65E-4) | 7.79E-3(2.63E-3) | 1.34E-2(4.13E-3) | 1.16E+1(9.04E-2) | **8.53E-4(9.74E-5)** | 3.94E-1(2.71E-1) | 5.61E-1(5.96E-2) |
| | 2d-LT | 1.14E+0(2.38E-5) | **1.13E+0(1.82E-4)** | 1.14E+0(1.67E-4) | 1.14E+0(6.41E-4) | 1.14E+0(3.55E-4) | 1.14E+0(8.74E-5) | 1.14E+0(2.23E-4) | 1.14E+0(0.00E+0) | 1.14E+0(2.20E-4) | 1.14E+0(3.27E-4) | 1.16E+0(2.83E-4) |
| NS | 2d-C | 4.19E-5(2.00E-6) | 4.03E-4(6.45E-5) | NaN | 7.56E-4(1.90E-4) | 4.18E-3(2.05E-4) | 1.07E-2(5.67E-3) | 1.13E-4(8.77E-6) | 5.30E-4(3.50E-4) | **2.33E-5(4.71E-6)** | 2.67E-5(4.71E-6) | 1.37E-4(7.24E-5) |
| | 2d-CG | 6.94E-4(6.45E-5) | 5.19E-3(2.43E-4) | 5.40E-3(2.49E-4) | 4.22E-3(5.82E-4) | 5.45E-3(2.13E-5) | 9.32E-3(3.09E-3) | 1.16E-3(8.97E-5) | 1.06E+0(1.61E-2) | **3.37E-4(6.60E-5)** | 1.72E-3(1.33E-3) | 3.34E+0(2.97E-5) |
| | 2d-LT | 5.06E+2(1.21E+0) | 5.10E+2(3.40E-1) | 5.10E+2(4.13E-1) | 5.09E+2(6.15E-1) | 5.10E+2(3.42E-1) | 5.10E+2(2.23E-1) | **5.05E+2(7.30E-1)** | 5.11E+2(1.76E-2) | 5.06E+2(1.82E-2) | 5.11E+2(2.99E+0) | 5.15E+2(1.77E+0) |
| Wave | 1d-C | 1.11E-1(3.66E-2) | 2.54E-2(1.61E-3) | 4.08E-2(4.31E-3) | **3.01E-3(4.82E-4)** | 9.07E-2(6.02E-3) | 4.68E-3(1.28E-3) | 9.66E-2(5.85E-3) | 6.17E-1(1.19E-1) | 6.03E-2(2.87E-3) | 1.48E-1(4.44E-2) | 1.39E-1(1.97E-2) |
| | 2d-CG | 1.64E-1(6.13E-2) | 1.28E-1(1.13E-2) | 1.03E-1(1.46E-2) | 2.17E-1(2.05E-2) | 6.25E-2(1.17E-2) | 5.59E-2(1.29E-2) | 3.09E-2(8.98E-4) | 5.24E-2(9.01E-3) | 3.49E-2(3.38E-3) | **2.99E-2(4.68E-4)** | 5.78E-2(7.99E-3) |
| | 2d-MS | 1.30E+5(4.25E+4) | 7.35E+4(1.68E+3) | 7.34E+4(1.97E+3) | 7.69E+4(4.55E+3) | 1.33E+5(4.47E+4) | 7.15E+4(8.04E+2) | 7.27E+4(5.47E+2) | 1.13E+2(1.46E+2) | 7.91E+4(2.55E+3) | 7.98E+4(8.00E+3) | 8.95E+5(1.15E+4) |
| Chaotic | GS | 1.00E-1(1.35E-1) | 1.64E-2(1.70E-2) | **4.32E-3(4.07E-6)** | 2.59E-2(1.44E-2) | 4.40E-3(8.83E-5) | 4.32E-3(1.11E-6) | 3.62E-2(2.28E-2) | 4.00E-1(2.33E-1) | 4.32E-3(4.71E-6) | 1.69E-2(1.79E-2) | 5.16E-3(1.64E-3) |
| | KS | 1.16E+0(2.95E-3) | 1.11E+0(5.07E-2) | **1.04E+0(6.20E-3)** | 1.06E+0(1.09E-2) | 1.16E+0(1.98E-3) | 1.05E+0(1.04E-2) | 1.12E+0(8.67E-3) | 1.05E+0(2.50E-3) | 1.16E+0(4.50E-3) | 1.14E+0(2.33E-2) | 1.16E+0(5.28E-2) |
| High dim | PNd | 9.47E-5(3.47E-5) | 8.30E-5(5.53E-5) | **2.09E-6(1.69E-7)** | 4.02E-4(5.23E-4) | 1.43E-4(9.92E-5) | 1.70E-4(9.61E-5) | 2.57E-4(6.31E-5) | – | 3.03E-4(2.25E-4) | 4.80E-4(2.81E-4) | – |
| | HNd | 1.19E+1(2.92E-1) | 1.93E+1(3.65E-1) | 1.42E+1(9.23E-1) | 1.44E+1(9.14E-1) | 1.17E+1(2.41E-1) | **8.52E+0(2.34E+0)** | 9.21E+0(3.90E-1) | – | 2.49E+1(2.99E-1) | 2.50E+1(2.76E-1) | – |
| Inverse | PInv | 1.89E-3(6.31E-5) | 5.89E-3(3.88E-4) | 5.08E-3(2.18E-4) | 7.94E-3(1.16E-3) | 1.89E-3(4.49E-4) | 3.64E-3(8.28E-4) | 1.37E-3(9.45E-5) | **1.23E-4(9.50E-5)** | 6.25E-1(8.80E-1) | 1.87E-2(1.98E-2) | 3.98E+0(1.33E+0) |
| | HInv | 5.36E+0(4.86E-1) | 6.02E-3(7.71E-4) | **5.66E-3(9.88E-4)** | 1.23E-2(1.75E-3) | 5.01E+0(4.22E-1) | 1.43E-2(4.35E-3) | 6.01E+1(3.72E+1) | 8.83E-1(6.52E-2) | 1.27E+0(1.69E+0) | 1.03E-1(4.73E-2) | 2.23E+2(5.54E+1) |

Table 10: Mean (Std) of MSE for main experiments.

| fMSE-L | Name | Vanilla | Loss Reweighting/Sampling | | | Optimizer | Loss functions | | Architecture | | |
|---|---|---|---|---|---|---|---|---|---|---|---|
| – | | PINN | LRA | NTK | RAR | MultiAdam | gPINN | vPINN | LAAF | GAAF | FBPINN |
| Burgers | 1d-C | 2.21E-02(1.02E-02) | **1.46E-02(1.77E-02)** | 1.75E-01(2.76E-01) | 5.02E-01(6.21E-01) | 1.19E-01(2.00E-01) | 1.79E+00(1.99E+00) | 1.40E+01(1.06E+00) | 1.32E-01(2.50E-01) | 9.38E-02(1.47E-01) | 2.10E+00(1.50E+00) |
| | 2d-C | 4.85E+01(9.27E+00) | 8.18E+01(1.24E+01) | 8.36E+01(1.07E+01) | **4.77E+01(6.85E+00)** | 1.39E+02(5.63E+01) | 8.89E+01(3.98E+00) | 4.54E+03(5.57E+03) | 8.34E+01(7.63E+00) | 9.27E+01(7.53E+00) | – |
| Poisson | 2d-C | – | – | – | – | – | – | – | – | – | – |
| | 2d-CG | – | – | – | – | – | – | – | – | – | – |
| | 3d-CG | – | – | – | – | – | – | – | – | – | – |
| | 2d-MS | **1.74E+03(6.29E+01)** | 8.62E+03(1.10E+03) | 8.62E+03(6.08E+02) | 2.99E+03(2.59E+02) | 3.46E+03(1.93E+03) | 7.41E+03(5.99E+02) | 1.13E+04(4.70E+01) | 2.61E+03(5.60E+02) | 1.24E+04(5.71E+03) | 5.90E+03(6.03E+00) |
| Heat | 2d-VC | 4.78E+00(5.53E-01) | **3.66E-02(8.92E-03)** | 3.58E-01(2.81E-01) | 2.00E+00(1.49E+00) | 2.78E+00(3.95E+00) | 2.91E+03(1.84E+03) | 1.74E+00(1.04E+00) | 1.43E+00(1.87E+00) | 1.28E+01(2.08E+01) | 4.34E+00(2.13E-2) |
| | 2d-MS | 1.56E-01(2.33E-01) | **1.12E-01(1.76E-01)** | 1.46E+00(1.60E+00) | 3.55E-01(3.96E-01) | 3.48E-01(3.43E-01) | 1.37E+01(1.38E+01) | – | 3.50E-01(2.74E-01) | 1.11E+00(1.01E+00) | – |
| | 2d-CG | – | – | – | – | – | – | – | – | – | – |
| | 2d-LT | 3.90E+02(6.18E+02) | 2.71E+01(3.42E-02) | 2.75E+01(5.86E-01) | 2.70E+01(1.36E-01) | **2.70E+01(9.84E-02)** | 3.34E+05(6.48E+05) | 2.63E+01(7.51E-01) | 2.70E+01(1.56E-01) | 2.71E+01(3.78E-02) | 8.47E+01(7.99E-1) |
| NS | 2d-C | 4.29E-02(3.76E-02) | – | 3.74E-01(1.26E-01) | 2.35E-02(1.24E-02) | 2.20E+01(2.96E+01) | 6.97E-01(4.72E-01) | 5.38E-01(1.29E-01) | **1.34E-02(1.03E-02)** | 1.73E-02(8.42E-03) | 2.47E-2(2.70E-3) |
| | 2d-CG | – | – | – | – | – | – | – | – | – | – |
| | 2d-LT | 2.07E+05(9.61E+02) | 2.05E+05(2.37E+02) | 2.07E+05(5.52E+02) | 2.06E+05(4.53E+02) | **2.05E+05(2.78E+02)** | 2.05E+05(2.96E+02) | 4.87E+04(1.64E+02) | 2.07E+05(7.29E+02) | 2.06E+05(7.01E+02) | – |
| Wave | 1d-C | 5.38E+01(1.52E+01) | 4.81E-01(5.90E-01) | **4.65E-01(4.37E-01)** | 1.10E+02(7.79E+01) | 3.57E+02(1.97E+02) | 3.00E+02(8.28E+01) | 2.85E+01(8.07E+00) | 1.98E+01(1.48E+01) | 3.89E+02(3.79E+02) | 6.01E+01(1.46E+01) |
| | 2d-CG | 2.42E+01(1.08E+01) | 3.47E+02(2.09E+02) | 5.26E+02(5.40E+01) | 1.75E+02(1.90E+02) | 1.25E+02(9.42E+01) | 8.10E+01(6.42E+00) | 1.25E+01(4.98E+00) | 4.36E+02(4.69E+02) | 7.19E+01(6.05E+01) | **1.49E+01(3.22E+00)** |
| | 2d-MS | 3.72E+08(3.03E+08) | 8.91E+05(1.18E+06) | 7.01E+05(3.95E+05) | 3.93E+08(3.18E+08) | 2.39E+06(2.83E+06) | 1.85E+06(1.89E+06) | **1.13E+02(9.57E-01)** | 3.08E+06(2.04E+06) | 4.33E+06(8.23E+06) | 1.10E+07(1.93E+06) |
| Chaotic | GS | 1.45E+02(4.99E+00) | 1.44E+01(1.09E+01) | 7.79E+00(4.79E+00) | 2.96E+02(7.67E+01) | 6.51E+00(9.23E+00) | 5.26E+01(2.87E+01) | 2.79E+01(2.09E+01) | 2.69E+02(1.34E+01) | 4.89E+01(5.30E+01) | **3.49E+00(2.45E+00)** |
| | KS | 1.65E+01(3.09E+01) | 1.06E+00(5.77E-03) | 3.81E+02(2.09E+02) | **1.03E+00(6.13E-02)** | 1.07E+00(6.48E-03) | 1.38E+02(6.24E+00) | 1.24E+02(1.76E+01) | 1.08E+00(1.41E-02) | 1.04E+00(2.96E-02) | – |
| High dim | PNd | – | – | – | – | – | – | – | – | – | – |
| | HNd | – | – | – | – | – | – | – | – | – | – |

Table 11: Mean (Std) of low-frequency Fourier error for main experiments.

| fMSE-M | Name | Vanilla | Loss Reweighting/Sampling | | | Optimizer | Loss functions | | Architecture | | |
|---|---|---|---|---|---|---|---|---|---|---|---|
| – | | PINN | LRA | NTK | RAR | MultiAdam | gPINN | vPINN | LAAF | GAAF | FBPINN |
| Burgers | 1d-C | 1.43E-03(1.93E-04) | 2.94E-05(2.03E-05) | 9.34E-05(7.38E-05) | 7.51E-05(6.01E-05) | 6.18E-04(5.93E-04) | 6.69E-03(1.40E-03) | 3.00E+00(2.22E-01) | 3.53E-05(5.77E-05) | **1.72E-05(1.72E-05)** | 4.88E-1(3.65E-1) |
| | 2d-C | 3.28E-01(4.31E-03) | 2.23E-01(3.59E-02) | **2.11E-01(2.04E-02)** | 3.32E-01(4.83E-03) | 3.25E-01(1.15E-03) | 3.03E+00(3.63E+00) | 3.23E-01(2.30E-04) | 3.25E-01(2.89E-02) | 2.95E-01(1.71E-02) | – |
| Poisson | 2d-C | – | – | – | – | – | – | – | – | – | – |
| | 2d-CG | – | – | – | – | – | – | – | – | – | – |
| | 3d-CG | – | – | – | – | – | – | – | – | – | – |
| | 2d-MS | 6.57E+00(3.70E-02) | 1.70E+01(4.14E+00) | 1.47E+01(1.54E+00) | 1.18E+01(2.32E+00) | **5.61E+00(3.18E+00)** | 8.26E+00(5.99E-01) | 2.88E+01(2.37E-01) | 1.26E+01(3.53E+00) | 9.87E+00(2.25E+00) | 2.03E+01(2.50E-3) |
| Heat | 2d-VC | 2.75E-02(2.11E-03) | **1.46E-03(4.26E-04)** | 7.99E-03(1.17E-02) | 1.01E-01(7.00E-02) | 1.54E-02(1.23E-02) | 6.91E+01(1.07E+02) | 9.26E-03(3.06E-03) | 3.58E-02(2.86E-02) | 8.90E-01(1.27E+00) | 1.56E-2(2.62E-4) |
| | 2d-MS | 7.35E-04(7.70E-04) | 4.55E-05(5.24E-05) | 1.26E-04(3.07E-05) | **2.07E-05(1.15E-05)** | 3.13E-03(3.56E-03) | 2.65E-03(1.55E-03) | 7.57E-02(2.40E-03) | 6.19E-05(1.89E-05) | 1.05E-04(1.25E-04) | Nan |
| | 2d-CG | – | – | – | – | – | – | – | – | – | – |
| | 2d-LT | 2.11E+00(3.00E+00) | 1.59E+02(1.37E-01) | 1.59E+02(1.97E-01) | 1.59E+02(2.12E-01) | 1.59E+02(3.85E-02) | 1.59E+02(1.81E-01) | **4.57E-01(2.67E-02)** | 1.59E+02(2.20E-01) | 1.59E+02(1.36E-01) | 8.91E-1(4.84E-2) |
| NS | 2d-C | 1.72E-04(9.81E-05) | – | 3.96E-03(3.45E-03) | 1.48E-04(6.28E-05) | 1.48E-01(2.31E-01) | 5.84E-03(9.82E-04) | 2.86E-03(1.58E-03) | 1.05E-04(8.46E-05) | 4.46E-05(3.38E-05) | **2.18E-5(2.47E-6)** |
| | 2d-CG | – | – | – | – | – | – | – | – | – | – |
| | 2d-LT | **1.00E-02(9.51E-04)** | 2.63E-02(2.55E-03) | 1.49E-02(3.02E-03) | 1.06E-02(9.30E-04) | 2.52E-02(4.51E-03) | 2.12E-02(2.08E-03) | – | 1.05E-02(1.37E-03) | 1.26E-02(2.12E-03) | 4.53E+00(1.83E-2) |
| Wave | 1d-C | 1.61E-01(3.55E-02) | **7.58E-03(8.03E-03)** | 3.62E-02(8.80E-03) | 8.55E-01(3.57E-01) | 2.63E+00(2.04E+00) | 2.50E+00(9.12E-01) | 5.93E-01(6.18E-02) | 1.96E-01(1.46E-01) | 1.48E+00(9.12E-01) | 8.48E-2(2.36E-2) |
| | 2d-CG | 8.29E-02(6.50E-03) | 1.06E-03(1.01E-03) | **8.18E-04(2.57E-04)** | 8.27E-04(3.83E-04) | 3.12E-03(2.43E-03) | 1.36E-03(3.20E-04) | 4.73E-02(3.85E-03) | 4.73E-02(3.85E-03) | 1.53E-03(5.15E-04) | 1.49E-03(5.58E-04) |
| | 2d-MS | 1.47E+04(1.73E+04) | 1.31E+05(1.66E+05) | 1.78E+05(9.50E+04) | 2.39E+04(4.63E+04) | 2.39E+05(1.80E+05) | 1.82E+04(2.13E+04) | **4.75E+01(2.01E+00)** | 1.62E+05(1.72E+05) | 3.15E+05(4.71E+05) | 6.18E+04(3.16E+04) |
| Chaotic | GS | 5.39E+01(2.19E-01) | 7.94E-02(5.38E-02) | 3.37E-02(9.80E-03) | 6.27E-02(1.76E-02) | 8.88E-02(8.20E-02) | 1.72E-01(3.91E-02) | **2.36E-02(1.40E-02)** | 4.62E-02(6.99E-03) | 1.35E-01(1.17E-01) | 5.08E-2(3.92E-2) |
| | KS | 5.54E-01(1.46E-02) | **5.45E-01(2.42E-03)** | 5.60E-01(2.11E-02) | 5.47E-01(3.82E-03) | 5.46E-01(6.98E-05) | 5.48E-01(1.06E-02) | – | 5.46E-01(3.93E-04) | 5.46E-01(1.23E-03) | – |
| High dim | PNd | – | – | – | – | – | – | – | – | – | – |
| | HNd | – | – | – | – | – | – | – | – | – | – |

Table 12: Mean (Std) of medium-frequency Fourier error for main experiments.

| fMSE-H | Name | Vanilla | Loss Reweighting/Sampling | | | Optimizer | Loss functions | | Architecture | | |
|---|---|---|---|---|---|---|---|---|---|---|---|
| – | | PINN | LRA | NTK | RAR | MultiAdam | gPINN | vPINN | LAAF | GAAF | FBPINN |
| Burgers | 1d-C | **2.96E-05(9.65E-06)** | 1.85E-04(8.34E-05) | 1.48E-04(1.36E-04) | 2.39E-03(2.55E-03) | 6.16E-04(4.21E-04) | 1.09E-01(8.12E-03) | 1.40E-02(1.26E-03) | 7.15E-04(1.24E-03) | 1.93E-04(7.30E-05) | 6.87E-3(4.12E-3) |
| | 2d-C | 6.78E-02(1.23E-03) | **5.33E-02(1.37E-03)** | 5.43E-02(1.56E-03) | 6.78E-02(1.08E-03) | 7.07E-02(7.45E-04) | 6.80E-02(2.95E-04) | 2.31E-01(2.46E-01) | 5.84E-02(1.19E-03) | 5.98E-02(1.01E-03) | – |
| Poisson | 2d-C | – | – | – | – | – | – | – | – | – | – |
| | 2d-CG | – | – | – | – | – | – | – | – | – | – |
| | 3d-CG | – | – | – | – | – | – | – | – | – | – |
| | 2d-MS | **1.68E-02(4.05E-04)** | 2.07E+00(3.81E-01) | 2.23E+00(1.14E-01) | 1.89E+00(2.11E-01) | 2.30E+00(5.97E-01) | 8.62E-01(3.40E-02) | 5.44E-02(4.98E-03) | 1.27E+00(2.34E-01) | 3.07E+00(1.14E+00) | 7.17E-2(8.16E-6) |
| Heat | 2d-VC | 4.22E-04(2.39E-04) | 1.88E-03(1.07E-04) | 1.90E-03(7.60E-05) | 3.02E-02(3.49E-03) | 1.15E-02(8.98E-03) | 1.99E+00(9.14E-01) | 5.11E-04(3.47E-04) | 2.43E-02(2.66E-03) | 2.63E-02(1.52E-02) | **6.39E-5(3.77E-6)** |
| | 2d-MS | **6.81E-06(5.66E-06)** | 7.12E-05(4.00E-05) | 9.24E-05(6.51E-05) | 9.91E-05(1.35E-04) | 5.69E-04(3.34E-04) | 1.03E-02(3.68E-03) | 1.99E-03(1.45E-04) | 8.63E-05(3.67E-05) | 1.62E-04(1.36E-04) | – |
| | 2d-CG | – | – | – | – | – | – | – | – | – | – |
| | 2d-LT | 2.10E-01(1.45E-02) | 7.73E-01(2.31E-04) | 7.72E-01(2.41E-04) | 7.72E-01(9.35E-05) | 7.73E-01(2.03E-04) | 7.95E-01(4.27E-02) | 2.70E-01(2.79E-02) | 7.73E-01(1.59E-04) | 7.72E-01(8.87E-05) | **2.05E-1(4.46E-4)** |
| NS | 2d-C | 4.89E-06(1.01E-06) | – | 2.05E-04(4.41E-05) | 3.80E-06(3.71E-07) | 2.16E-03(5.85E-04) | 1.32E-03(3.12E-04) | 6.98E-06(4.41E-06) | 1.18E-06(2.56E-07) | 2.05E-06(7.23E-07) | **6.48E-8(1.75E-8)** |
| | 2d-CG | – | – | – | – | – | – | – | – | – | – |
| | 2d-LT | 1.09E+02(1.92E-01) | 1.11E+02(3.51E-02) | 1.10E+02(1.61E-01) | 1.09E+02(1.16E-01) | 1.10E+02(2.72E-01) | 1.10E+02(1.00E-01) | 4.51E+02(3.00E+00) | 1.09E+02(1.30E-01) | 1.09E+02(4.04E-01) | – |
| Wave | 1d-C | **1.25E-03(3.43E-04)** | 3.43E-02(9.35E-03) | 4.28E-03(6.00E-04) | 7.06E-02(7.51E-03) | 8.96E-02(1.06E-02) | 8.64E-02(5.08E-03) | 6.15E-04(7.67E-05) | 5.34E-02(2.92E-03) | 8.45E-02(1.85E-02) | – |
| | 2d-CG | 6.39E-03(1.09E-03) | 3.93E-02(1.01E-02) | 4.80E-02(3.17E-03) | 3.31E-02(9.96E-03) | 2.70E-02(4.96E-03) | 3.09E-02(5.50E-04) | 3.03E-03(2.29E-04) | 4.91E-02(3.19E-02) | 2.54E-02(2.53E-03) | **5.11E-3(1.86E-4)** |
| | 2d-MS | 7.61E+04(4.06E+03) | 7.51E+04(1.67E+03) | 7.90E+04(4.65E+03) | 7.62E+04(5.81E+03) | **7.35E+04(3.05E+02)** | 7.49E+04(5.94E+02) | – | 8.09E+04(2.45E+03) | 8.09E+04(6.69E+03) | – |
| Chaotic | GS | 5.30E-01(1.48E-03) | 1.04E+00(7.44E-03) | 1.05E+00(4.11E-03) | 1.12E+00(2.99E-03) | 1.04E+00(5.34E-03) | 1.10E+00(2.14E-03) | **1.70E-03(1.01E-03)** | 1.12E+00(2.14E-03) | 1.09E+00(7.71E-03) | – |
| | KS | 1.27E-03(2.94E-04) | 1.11E-03(1.32E-06) | 2.17E-03(2.09E-03) | 1.12E-03(2.36E-05) | **1.11E-03(7.08E-09)** | 7.28E-03(1.16E-03) | 4.48E-01(1.76E-03) | 1.11E-03(1.45E-07) | 1.11E-03(7.82E-07) | – |
| High dim | PNd | – | – | – | – | – | – | – | – | – | – |
| | HNd | – | – | – | – | – | – | – | – | – | – |

Table 13: Mean (Std) of high-frequency Fourier error for main experiments.

| Avg Runtime | Name | Vanilla | | | Loss Reweighting/Sampling | | Optimizer | Loss functions | | Architecture | | |
|---|---|---|---|---|---|---|---|---|---|---|---|---|
| – | | PINN | PINN-w | L-BFGS | LRA | NTK | MultiAdam | gPINN | vPINN | LAAF | GAAF | FBPINN |
| Burgers | 1d-C | **2.84E+2** | 2.78E+2 | 3.76E+3 | 7.64E+2 | 6.70E+2 | 5.06E+2 | 6.28E+2 | 2.85E+2 | 3.61E+2 | 3.56E+2 | 1.11E+3 |
| | 2d-C | 3.11E+3 | 3.11E+3 | 4.98E+4 | 1.84E+4 | 4.35E+3 | 2.72E+3 | 4.03E+3 | **8.95E+2** | 4.08E+3 | 4.07E+3 | – |
| Poisson | 2d-C | 3.39E+2 | 3.33E+2 | 1.65E+3 | 9.01E+2 | 8.09E+2 | 6.13E+2 | 7.66E+2 | **3.29E+2** | 5.72E+2 | 4.20E+2 | 4.12E+3 |
| | 2d-CG | 3.69E+2 | 3.59E+2 | 3.13E+4 | 9.36E+2 | 8.80E+2 | 6.57E+2 | 8.06E+2 | **3.55E+2** | 6.05E+2 | 4.34E+2 | 4.17E+3 |
| | 3d-CG | **1.45E+3** | 2.32E+3 | 8.55E+4 | 4.06E+3 | 4.40E+3 | 2.41E+3 | 5.01E+3 | 1.94E+3 | 2.01E+3 | 1.68E+3 | 2.18E+3 |
| | 2d-MS | 3.83E+2 | **3.74E+2** | 2.00E+4 | 7.47E+2 | 8.74E+2 | 6.67E+2 | 7.92E+2 | 1.81E+3 | 6.62E+2 | 4.57E+2 | 4.22E+3 |
| Heat | 2d-VC | **1.16E+3** | **1.16E+3** | 8.21E+4 | 3.52E+3 | 1.69E+3 | 1.91E+3 | 1.34E+3 | 3.03E+3 | 1.52E+3 | 1.52E+3 | 3.92E+3 |
| | 2d-MS | **1.13E+3** | 1.14E+3 | 5.72E+3 | 3.48E+3 | 1.61E+3 | 1.89E+3 | 1.30E+3 | 1.69E+3 | 1.51E+3 | 1.50E+3 | 5.84E+3 |
| | 2d-CG | **1.16E+3** | 1.17E+3 | 1.28E+4 | 5.14E+3 | 1.64E+3 | 1.90E+3 | 1.31E+3 | 3.05E+3 | 1.52E+3 | 1.51E+3 | 5.28E+3 |
| | 2d-LT | **1.15E+3** | 1.18E+3 | 7.78E+3 | 3.52E+3 | 1.65E+3 | 1.90E+3 | 1.32E+3 | 2.12E+3 | 1.51E+3 | 1.50E+3 | 3.93E+3 |
| NS | 2d-C | 7.52E+2 | 7.64E+2 | 1.24E+3 | 2.24E+3 | 1.84E+3 | 1.25E+3 | 2.03E+3 | **5.68E+2** | 9.49E+2 | 9.43E+2 | 7.16E+3 |
| | 2d-CG | 7.56E+2 | 7.58E+2 | 2.78E+3 | 3.26E+3 | 1.84E+3 | 1.22E+3 | 1.97E+3 | **6.79E+2** | 9.35E+2 | 9.31E+2 | 5.48E+3 |
| | 2d-LT | 3.05E+3 | 3.05E+3 | 4.54E+4 | 2.25E+4 | 4.29E+3 | 3.73E+3 | 4.42E+3 | **1.38E+3** | 3.99E+3 | 3.99E+3 | 4.10E+3 |
| Wave | 1d-C | 3.50E+2 | 3.52E+2 | 2.98E+4 | 1.12E+3 | 8.40E+2 | 2.72E+2 | 7.75E+2 | **2.22E+2** | 6.01E+2 | 4.36E+2 | 3.09E+3 |
| | 2d-CG | 1.21E+3 | 1.24E+3 | 2.62E+4 | 4.50E+3 | 1.77E+3 | 2.01E+3 | 1.27E+3 | **5.99E+2** | 2.35E+3 | 1.57E+3 | 3.01E+3 |
| | 2d-MS | **2.19E+3** | **2.19E+3** | 1.23E+4 | 6.76E+3 | 5.02E+3 | 4.12E+3 | 6.18E+3 | **2.11E+3** | 2.63E+3 | 2.25E+3 | 3.67E+3 |
| Chaotic | GS | 2.55E+3 | 2.55E+3 | 2.76E+3 | 7.57E+3 | 3.17E+3 | 4.22E+3 | 2.59E+3 | **6.12E+2** | 3.23E+3 | 3.22E+3 | 5.47E+3 |
| | KS | 1.40E+3 | 1.40E+3 | 2.97E+3 | 3.17E+3 | 3.59E+3 | 2.29E+3 | 3.83E+3 | **7.14E+2** | 1.62E+3 | 1.63E+3 | 8.83E+3 |
| High dim | PNd | **1.78E+3** | 1.83E+3 | 2.19E+3 | 4.30E+3 | 4.75E+3 | 3.02E+3 | 1.91E+3 | – | 3.50E+3 | 2.33E+3 | – |
| | HNd | **2.35E+3** | 2.45E+3 | 2.97E+3 | 7.42E+3 | 6.28E+3 | 4.00E+3 | 2.74E+3 | – | 3.09E+3 | 3.08E+3 | – |
| Inverse | PInv | 4.53E+2 | 4.88E+2 | 2.76E+3 | 1.25E+3 | 1.71E+3 | 7.46E+2 | 1.50E+3 | **4.90E+2** | 5.75E+2 | 5.88E+2 | 3.63E+3 |
| | HInv | **1.09E+3** | 1.12E+3 | 2.78E+3 | 3.39E+3 | 1.68E+3 | 1.77E+3 | 1.56E+3 | 1.86E+3 | 1.44E+3 | 1.44E+3 | 3.93E+3 |

Table 14: Average running time (seconds) for main experiments, we run all methods three times with 20000 epochs.

| Training Flops | Name | Vanilla | | | Loss Reweighting/Sampling | | Optimizer | Loss functions | | Architecture | | |
|---|---|---|---|---|---|---|---|---|---|---|---|---|
| – | | PINN | PINN-w | L-BFGS | LRA | NTK | MultiAdam | gPINN | vPINN | LAAF | GAAF | FBPINN |
| Burgers | 1d-C | **1.87E+11** | 1.87E+11 | 2.32E+12 | 5.12E+11 | 4.29E+11 | 3.39E+11 | 4.11E+11 | 1.81E+11 | 2.22E+11 | 2.29E+11 | 7.34E+11 |
| | 2d-C | 2.72E+12 | 2.72E+12 | 4.17E+13 | 1.23E+13 | 2.61E+12 | 1.82E+12 | 2.79E+12 | **6.23E+11** | 2.53E+12 | 2.73E+12 | – |
| Poisson | 2d-C | 2.55E+11 | 2.55E+11 | 1.34E+12 | 6.04E+11 | 5.32E+11 | 4.15E+11 | 5.03E+11 | **2.21E+11** | 3.83E+11 | 2.81E+11 | 2.46E+12 |
| | 2d-CG | 2.37E+11 | 2.37E+11 | 2.08E+13 | 6.17E+11 | 5.82E+11 | 4.4E+11 | 5.29E+11 | **2.38E+11** | 4.05E+11 | 2.91E+11 | 2.79E+12 |
| | 3d-CG | **9.03E+11** | 9.03E+11 | 5.32E+13 | 2.72E+12 | 2.95E+12 | 1.61E+12 | 3.36E+12 | 1.3E+12 | 1.35E+12 | 1.13E+12 | 1.46E+12 |
| | 2d-MS | **2.75E+11** | 2.75E+11 | 5.02E+11 | 1.43E+13 | 5.66E+11 | 4.47E+11 | 5.31E+11 | 1.21E+12 | 4.44E+11 | 3.06E+11 | 2.83E+12 |
| Heat | 2d-VC | **7.10E+11** | 7.10E+11 | 4.87E+13 | 2.26E+12 | 1.03E+12 | 1.28E+12 | 8.98E+11 | 2.03E+12 | 1.02E+12 | 1.02E+12 | 2.63E+12 |
| | 2d-MS | **7.15E+11** | 7.15E+11 | 2.23E+12 | 3.35E+13 | 1.01E+12 | 1.27E+12 | 8.71E+11 | 1.13E+12 | 1.05E+12 | 1.01E+12 | 3.71E+12 |
| | 2d-CG | **6.91E+11** | 6.91E+11 | 7.52E+12 | 3.34E+12 | 1.07E+12 | 1.27E+12 | 8.78E+11 | 2.04E+12 | 1.08E+12 | 1.01E+12 | 3.54E+12 |
| | 2d-LT | **7.62E+11** | 7.62E+11 | 3.84E+12 | 2.26E+12 | 1.11E+12 | 1.27E+12 | 8.84E+11 | 1.42E+12 | 1.01E+12 | 1.01E+12 | 2.63E+12 |
| NS | 2d-C | 5.05E+11 | 5.05E+11 | 9.82E+11 | 1.39E+12 | 1.23E+12 | 8.38E+11 | 1.36E+12 | **3.81E+11** | 6.36E+11 | 6.32E+11 | 4.85E+12 |
| | 2d-CG | 4.85E+11 | 4.85E+11 | 1.80E+12 | 2.14E+12 | 1.23E+12 | 8.17E+11 | 1.32E+12 | **4.55E+11** | 6.26E+11 | 6.24E+11 | 3.67E+12 |
| | 2d-LT | 1.87E+12 | 1.87E+12 | 2.57E+13 | 1.51E+13 | 2.87E+12 | 2.5E+12 | 2.96E+12 | **9.25E+11** | 2.77E+12 | 2.67E+12 | 2.75E+12 |
| Wave | 1d-C | 2.15E+11 | 2.15E+11 | 1.44E+13 | 7.51E+11 | 5.63E+11 | 1.82E+11 | 5.19E+11 | **1.49E+11** | 4.13E+11 | 2.92E+11 | 2.07E+12 |
| | 2d-CG | 7.11E+11 | 7.11E+11 | 1.57E+13 | 3.02E+12 | 1.19E+12 | 1.35E+12 | 8.51E+11 | **4.01E+11** | 1.75E+12 | 1.08E+12 | 2.02E+12 |
| | 2d-MS | 1.47E+12 | 1.47E+12 | 8.77E+12 | 4.53E+12 | 3.36E+12 | 2.76E+12 | 4.14E+12 | **1.41E+12** | 1.74E+12 | 1.51E+12 | 2.46E+12 |
| Chaotic | GS | 1.68E+12 | 1.68E+12 | 1.79E+12 | 5.07E+12 | 2.12E+12 | 2.83E+12 | 1.74E+12 | **4.07E+11** | 2.16E+12 | 2.16E+12 | 3.66E+12 |
| | KS | 9.12E+11 | 9.12E+11 | 1.96E+12 | 2.12E+12 | 2.41E+12 | 1.53E+12 | 2.57E+12 | **4.78E+11** | 1.09E+12 | 1.09E+12 | 5.92E+12 |
| High dim | PNd | **1.19E+12** | 1.19E+12 | 1.40E+12 | 2.88E+12 | 3.18E+12 | 2.02E+12 | 1.28E+12 | – | 2.35E+12 | 1.56E+12 | – |
| | HNd | **1.57E+12** | 1.57E+12 | 4.97E+12 | 2.02E+12 | 4.21E+12 | 2.68E+12 | 1.84E+12 | – | 2.07E+12 | 2.06E+12 | – |
| Inverse | PInv | **3.04E+11** | 3.27E+11 | 1.68E+12 | 8.38E+11 | 1.15E+12 | 5.24E+11 | 1.01E+12 | 3.28E+11 | 3.85E+11 | 3.94E+11 | 2.43E+12 |
| | HInv | **7.34E+11** | 7.34E+11 | 1.81E+12 | 2.27E+12 | 1.13E+12 | 1.19E+12 | 1.05E+12 | 1.25E+12 | 9.65E+11 | 9.65E+11 | 2.63E+12 |

Table 15: Average Flops every epoch for main experiments, we run all methods three times.

| L2RE | | Burgers1d | GS | Heat2d-CG | Poisson2d-C |
|---|---|---|---|---|---|
| PINN | 1e-5 | 2.35E-2(1.90E-3) | 9.39E-2(3.60E-4) | 1.20E-1(2.40E-3) | 1.08E+0(1.08E-1) |
| | 1e-4 | 1.99E-2(4.30E-3) | 1.79E-1(1.20E-1) | 1.35E-1(2.00E-2) | 2.81E-2(2.44E-3) |
| | 1e-3 | 1.93E-2(4.00E-3) | **9.35E-2(2.30E-4)** | **8.51E-2(8.90E-3)** | **2.32E-2(1.52E-3)** |
| | 1e-2 | 3.79E-1(1.40E-1) | 1.91E-1(1.30E-1) | 1.73E-1(7.10E-2) | 3.26E-2(1.51E-3) |
| | decay | **1.69E-2(4.10E-3)** | 1.81E-1(1.20E-1) | 1.59E-1(2.00E-2) | 2.41E-2(9.33E-4) |
| PINN-LRA | 1e-5 | 3.44E-2(1.40E-2) | 1.79E-1(1.20E-1) | 1.18E-1(7.60E-4) | 2.91E-2(3.19E-3) |
| | 1e-4 | 2.12E-2(5.30E-3) | **9.36E-2(4.50E-4)** | 1.37E-1(8.50E-3) | 2.49E-2(3.88E-3) |
| | 1e-3 | 1.49E-2(9.60E-4) | 9.37E-2(3.63E-5) | 1.31E-1(9.60E-3) | **2.26E-2(1.93E-3)** |
| | 1e-2 | 6.23E-1(7.40E-2) | 1.29E-1(5.10E-2) | **8.99E-2(7.00E-3)** | 1.00E+0(5.62E-7) |
| | decay | **1.37E-2(5.00E-4)** | 1.81E-1(1.20E-1) | 1.19E-1(1.30E-2) | 2.61E-2(7.64E-4) |
| PINN-NTK | 1e-5 | 1.08E-1(2.70E-2) | 4.09E-1(1.20E-3) | **1.21E-1(2.80E-3)** | 1.86E-3(1.26E-4) |
| | 1e-4 | 4.72E-2(8.70E-3) | **1.96E-1(1.40E-1)** | 1.27E-1(5.30E-3) | 2.30E-3(9.48E-4) |
| | 1e-3 | 2.91E-2(7.40E-3) | 2.99E-1(1.50E-1) | 1.21E-1(9.50E-3) | 5.34E-3(1.22E-4) |
| | 1e-2 | NaN | 1.90E+0(1.63E+0) | NaN | 2.39E-1(2.00E-1) |
| | decay | **1.74E-2(2.30E-3)** | 3.06E-1(1.50E-1) | 1.48E-1(9.60E-3) | **8.24E-4(1.32E-4)** |

Table 16: Results of PINN, PINN-NTK, PINN-LRA under different learning rates or learning rate schedules.

| L2RE | | Burgers1d | GS | Heat2d-CG | Poisson2d-C |
|---|---|---|---|---|---|
| PINN | 512 | 4.59E-1(8.36E-2) | 2.46E-1(1.09E-1) | 4.31E-1(6.57E-2) | 3.15E-2(4.04E-3) |
| | 2048 | 2.60E-1(2.43E-1) | 9.37E-2(2.60E-4) | 2.02E-1(1.92E-2) | 2.62E-2(2.31E-3) |
| | 8192 | 2.14E-2(1.76E-3) | 9.41E-2(6.05E-4) | 1.35E-1(1.71E-2) | **2.58E-2(6.51E-4)** |
| | 32768 | **1.44E-2(4.91E-4)** | **9.37E-2(3.89E-5)** | 3.73E-2(3.23E-3) | 2.63E-2(2.32E-3) |
| PINN-LRA | 512 | 2.80E-1(2.02E-1) | 9.39E-2(1.66E-4) | 3.66E-1(3.86E-2) | 3.00E-2(3.16E-3) |
| | 2048 | 1.82E-1(1.85E-1) | 1.33E-1(5.57E-2) | 2.07E-1(4.96E-3) | 2.57E-2(1.78E-3) |
| | 8192 | 1.88E-2(9.45E-4) | **9.36E-2(2.14E-4)** | 1.01E-1(2.18E-2) | 2.82E-2(8.12E-4) |
| | 32768 | **1.49E-2(1.51E-3)** | 1.17E-1(3.25E-2) | **4.44E-2(1.05E-2)** | **2.49E-2(6.32E-4)** |

Table 17: Comparison of PINN and PINN-LRA's performance under different batch sizes (number of collocation points).

### E.2 Ablation Experiments

**Influence of learning rates.** To understand the impact of learning rates We selected three methods, i.e., vanilla Physics-Informed Neural Networks (PINN), PINN-NTK, and PINN-LRA. We conduct experiments on four PDE problems, i.e., Burgers1d-C, GS, Heat2d-CG, and Poisson2d-C. The comparative analysis involved evaluating the performance of these methods using learning rates of 1e-5, 1e-4, 1e-3, and 1e-2, along with a step learning rate decay strategy implemented every 1000 epochs with a decay factor of 0.75. The results are shown in Table E.2. As stated in the main text, a moderate learning rate like 1e-3, 1e-4, or using a decay strategy is a good choice.

**Influence of batch size (Collocation points).** To further understand the impact of the number of collocation points on our model's performance, we conducted an ablation study. We used four different numbers of collocation points, specifically 512, 2048, 8192, and 32768. The cases tested in this study were burgers1d, GS, Heat2d-CG, and Poisson2d, which is the same as the ablation study on learning rates. We utilized two variants of Physics-Informed Neural Networks: the vanilla PINN and the PINN-LRA. We found that using more batch size leads to a continual improvement in performance. For some cases, 8192 is a enough large batch size and the performance saturates. The conclusions and plots of this experiment are shown in the main text.

**Influence of training epochs.** In this ablation study, we examine the impact of varying the number of training epochs on our model's performance. We selected four different values, specifically 5k, 20k, 80k, and 160k epochs. Similar to the previous study, the cases chosen for testing were burgers1d, GS, Heat2d-CG, and Poisson2d. The trend is that training more epochs leads to better performance. However, it is easier to saturate than a larger batch size.

**Influence of Adam hyperparameters.** Here we examine the impact of varying the momentum hyperparameters in the Adam optimizer. Despite the learning rate, Adam contains two momentum hyperparameters, i.e., $(\beta_1, \beta_2)$ for storing the approximate first and second-order momentum. In

| Name | | Inference Flops |
|---|---|---|
| Burgers | 1d-C | 5.03E+4 |
| | 2d-C | 5.05E+4 |
| Poisson | 2d-C | 5.03E+4 |
| | 2d-CG | 5.03E+4 |
| | 3d-CG | 5.03E+4 |
| | 2d-MS | 5.04E+4 |
| Heat | 2d-VC | 5.03E+4 |
| | 2d-MS | 5.04E+4 |
| | 2d-CG | 5.04E+4 |
| | 2d-LT | 5.04E+4 |
| NS | 2d-C | 5.04E+4 |
| | 2d-CG | 5.05E+4 |
| | 2d-LT | 5.05E+4 |
| Wave | 1d-C | 5.06E+4 |
| | 2d-CG | 5.04E+4 |
| | 2d-MS | 5.04E+4 |
| Chaotic | GS | 5.04E+4 |
| | KS | 5.02E+4 |
| High dim | PNd | 5.06E+4 |
| | HNd | 5.06E+4 |
| Inverse | PInv | 5.04E+4 |
| | HInv | 5.05E+4 |

Table 18: Inference flops on a single collocation point for PINNs using network parameters the same with main experiments.

| L2RE | | Burgers1d-C | GS | Heat2d-CG | Poisson2d-C |
|---|---|---|---|---|---|
| PINN | 5k | 3.71E-2(1.21E-2) | 2.40E-1(1.11E-1) | 1.23E-1(3.77E-3) | 3.84E-2(1.77E-3) |
| | 20k | 1.66E-2(1.87E-3) | 1.65E-1(1.01E-1) | 8.95E-2(2.29E-2) | 2.38E-2(1.43E-3) |
| | 80k | 1.42E-2(6.63E-4) | **9.36E-2(8.41E-5)** | 9.64E-2(1.85E-2) | 1.86E-2(3.26E-3) |
| | 160k | 1.**38E-2(5.45E-4)** | 9.38E-2(5.38E-5) | **8.21E-2(7.52E-3)** | **1.48E-2(1.55E-3)** |
| PINN-LRA | 5k | 3.60E-2(8.82E-3) | 1.64E-1(9.91E-2) | 1.18E-1(2.32E-3) | 3.87E-2(3.28E-3) |
| | 20k | 1.56E-2(8.87E-4) | 1.09E-1(2.19E-2) | 9.29E-2(1.97E-2) | 2.65E-2(1.92E-3) |
| | 80k | 1.42E-2(1.23E-3) | 9.38E-2(1.63E-4) | **1.05E-1(1.48E-2)** | 1.79E-2(4.19E-4) |
| | 160k | **1.35E-2(1.84E-4)** | **9.38E-2(5.48E-4)** | 1.19E-1(2.28E-2) | **1.66E-2(3.50E-3)** |

Table 19: Performance of PINNs and PINN-LRA with different numbers of training epochs on 4 cases.

experiments, we observe that the momentum parameters not only affect the convergence speed and stability but also influence the final error. Here we list the results in Table E.2. We observe that in average $(\beta_1, \beta_2) = (0.99, 0.99)$ achieves the best results compared with others.

**Other method-specific parameters**

We chose several different method-specific hyperparameters to study their influence.

| L2RE | | burgers | GS | HeatComplex | Poisson2d |
|---|---|---|---|---|---|
| PINN | (0.9,0.999) | 1.79E-2(2.20E-3) | 2.47E-1(1.09E-1) | 7.76E-2(8.27E-3) | 2.72E-2(2.40E-3) |
| | (0.9,0.99) | 1.52E-2(1.34E-4) | 9.38E-2(5.93E-5) | 5.10E-2(7.20E-3) | 3.00E-2(6.98E-3) |
| | (0.9,0.9) | 1.68E-2(2.45E-3) | 9.38E-2(1.98E-4) | 4.56E-2(2.55E-3) | 2.81E-2(3.95E-3) |
| | (0.99,0.99) | **1.35E-2(1.03E-4)** | **9.37E-2(1.38E-5)** | **2.98E-2(5.24E-3)** | **9.18E-3(4.90E-4)** |
| PINN-NTK | (0.9,0.999) | 1.60E-2(5.50E-4) | 1.79E-1(1.20E-1) | 7.37E-2(1.59E-2) | 1.40E-2(4.06E-3) |
| | (0.9,0.99) | 1.57E-2(1.34E-4) | 9.37E-2(5.93E-5) | 6.65E-2(7.20E-3) | 1.57E-2(3.03E-3) |
| | (0.9,0.9) | 1.74E-2(1.40E-3) | 9.37E-2(2.11E-4) | 8.12E-2(3.33E-2) | 2.45E-2(3.64E-3) |
| | (0.99,0.99) | **1.35E-2(2.27E-4)** | **9.37E-2(1.54E-5)** | **3.62E-2(1.60E-3)** | **2.85E-3(1.68E-4)** |

Table 20: Performance comparison of PINN and PINN-NTK under different momentum parameters of Adam optimizer.

| $\alpha$ | Burgers1d | GS | Heat2d-CG | Poisson2d-C |
|---|---|---|---|---|
| 0.01 | 2.45E-2(1.75E-3) | 9.37E-2(4.25E-5) | **1.18E-1(4.72E-3)** | **2.51E-2(8.40E-3)** |
| 0.05 | 5.20E-2(2.14E-2) | 9.37E-2(3.48E-5) | 1.25E-1(7.62E-3) | 2.63E-2(1.10E-2) |
| 0.1 | **1.99E-2(5.61E-3)** | **9.37E-2(1.70E-5)** | 1.28E-1(4.66E-3) | 2.62E-1(3.00E-1) |
| 0.2 | 2.04E-2(3.66E-3) | 9.37E-2(1.03E-5) | 1.55E-1(3.31E-2) | 4.69E-2(1.37E-2) |
| 0.4 | 3.53E-2(2.47E-2) | 1.75E-1(1.15E-1) | 1.35E-1(7.37E-3) | 1.14E-1(1.23E-1) |
| 0.7 | 2.00E-2(3.72E-3) | 9.37E-2(4.21E-5) | 1.90E-1(4.09E-2) | 3.50E-1(2.24E-1) |

Table 21: Performance comparison of PINN-LRA with different momentum parameters.

| weight $w$ | Burgers1d | GS | Heat2d-CG | Poisson2d-C |
|---|---|---|---|---|
| 0.001 | **6.12E-2(1.36E-2)** | 1.66E-1(1.01E-1) | **4.97E-2(7.10E-4)** | **6.74E-1(1.71E-2)** |
| 0.01 | 1.95E-1(2.47E-2) | 1.79E-1(1.21E-1) | 7.78E-2(1.47E-2) | 6.89E-1(2.47E-2) |
| 0.1 | 4.93E-1(1.59E-2) | 4.61E-1(1.99E-1) | 1.34E-1(1.37E-3) | 6.92E-1(7.72E-3) |
| 1 | 5.53E-1(7.49E-2) | **9.38E-2(1.79E-5)** | 2.19E-1(9.90E-2) | 6.96E-1(4.39E-3) |

Table 22: Performance comparison of gPINN with different weights.

**Influence of momentum parameters for loss reweighting.**   Here we choose the momentum update $\alpha$ from $\{0.01, 0.05, 0.2, 0.4, 0.7\}$. We see that the optimal value of $\alpha$ is problem-dependent. However, we observe that relatively small $\alpha$ achieves better performance.

**Influence of weight for gPINNs.**   Here we choose the weight of gPINNs $w$ from $\{0.001, 0.01, 0.1, 1\}$. We see that the optimal value of $w$ is also problem-dependent and the property is intriguing. We observe that the performance of gPINNs is bad on Poisson2d-C for all values of $w$. We suggest that adding higher-order PDE residuals might harm the training process in some situations.

**Influence of number of grids for hp-VPINNs.**   The number of points to compute integral within a domain $Q$ and number of grids $N_{\text{grid}}$ are two critical hyperparameters for hp-VPINN. Here we choose $Q$ from $\{5, 10, 15, 20\}$ for 2-dimensional problems and $\{6, 8, 10, 12\}$ for 3-dimensional problems to investigate their influence. We also take $N_{\text{grid}}$ into consideration, which varies in $\{4, 8, 16, 32\}$ for 2-dimensional problems and 3-dimensional problems. Different parameter selection is applied due to the limit of the VRAM. We can observe a consistent trend that as the $Q$ value rises, the accuracy of the model's predictions also enhanced. This is attributed to the fact that the $Q$ value dictates the number of integration points; hence, a higher value leads to more precise integration. However, for certain scenarios where hp-VPINN might not be the best fit, a surge in the $Q$ value doesn't significantly bolster the prediction accuracy. On the other hand, the choice of $N_{\text{grid}}$ exhibits a complex influence on accuracy. Generally, as the value of $N_{\text{grid}}$ increases, precision tends to improve. However, in regions where the solution has large gradients or discontinuities, a denser grid might amplify these anomalies, leading to larger errors during model training.

**Influence of the number of subdomains and overlap factors for FBPINNs.**   The number of subdomains for domain decomposition and the overlap ratio $\alpha$ are two important hyperparameters for FBPINNs. The overlap ratio is chosen from $\{0.2, 0.4, 0.6, 0.8\}$.

**Results on different domain scale**   Here we study the influence of domain scales. While numerical methods are usually resistant to domain scales, PINN methods are not invariant to domain scale changes. Moreover, normalizing the domain to $[0, 1]$ might be suboptimal for PINNs. Here we take the domain scale $L$ of Poisson2d-C as an example to study the performance under different

| Q | Burgers1d | Q | GS | Q | Heat2d-CG | Q | Poisson2d-C |
|---|---|---|---|---|---|---|---|
| 5 | 3.19E-01(2.91E-02) | 6 | 3.88E-01(9.73E-02) | 6 | 7.14E-01(7.14E-01) | 5 | 2.46E-01(1.62E-01) |
| 10 | 2.88E-01(6.03E-03) | 8 | 4.25E-01(1.51E-01) | 8 | 7.19E-01(4.89E-02) | 10 | 2.43E-01(1.57E-01) |
| 15 | 1.85E-01(6.97E-02) | 10 | 3.68E-01(2.04E-01) | 10 | 7.19E-01(4.75E-02) | 15 | 2.45E-01(1.61E-01) |
| 20 | 1.85E-01(4.65E-02) | 12 | 3.58E-01(2.06E-01) | 12 | 7.21E-01(4.95E-02) | 20 | 2.46E-01(2.46E-01) |

Table 23: Performance comparison of hp-VPINN with different $Q$.

| $N_{\text{grid}}$ | Burgers1d | $N_{\text{grid}}$ | GS | $N_{\text{grid}}$ | Heat2d-CG | $N_{\text{grid}}$ | Poisson2d-C |
|---|---|---|---|---|---|---|---|
| 4 | 3.67E-01(1.28E-02) | 3 | 1.93E-01(2.06E-02) | 3 | 6.91E-01(2.44E-02) | 4 | 4.95E-01(8.46E-02) |
| 8 | 2.43E-01(2.39E-03) | 4 | 3.68E-01(2.04E-01) | 4 | 7.19E-01(4.75E-02) | 8 | 4.95E-01(8.63E-02) |
| 16 | 3.66E-01(3.67E-02) | 5 | 3.59E-01(1.34E-01) | 5 | 7.22E-01(5.14E-02) | 16 | 2.86E-01(1.94E-02) |
| 32 | 4.59E-01(1.34E-02) | 6 | 2.81E-01(1.96E-01) | 6 | 7.23E-01(5.19E-02) | 32 | 2.43E-01(1.57E-01) |

Table 24: Performance comparison of hp-VPINN with different number of grids $N_{\text{grid}}$.

|  | Burgers1d |  | GS |
|---|---|---|---|
| (1,1) | 2.12E-1(1.19E-1) | (1,1,1) | 7.98E-2(3.59E-3) |
| (2,1) | 1.75E-1(7.97E-2) | (1,1,3) | 8.15E-2(1.73E-3) |
| (3,1) | 1.61E-1(9.77E-2) | (1,1,5) | 7.90E-2(1.28E-3) |
| (1,2) | 1.98E-1(7.34E-2) | (2,2,1) | 8.15E-2(3.56E-3) |
|  | Heat2d-CG |  | Poisson2d-C |
| (1,1,1) | 3.30E-1(1.04E-1) | (1,1) | 5.01E-2(2.80E-3) |
| (1,1,3) | 6.80E-1(1.18E-1) | (1,2) | 3.51E-1(1.26E-1) |
| (1,1,5) | 7.48E-1(3.39E-2) | (2,1) | 4.38E-1(5.30E-2) |
| (2,2,1) | 2.89E-1(2.30E-2) | (2,2) | 5.54E-2(1.23E-3) |

Table 25: Performance (L2RE) comparison of FBPINN with different domain decomposition types.

settings. We see that Multi-Adam is the most stable under domain scale changes and achieves the best performance when $L$ is small.

**Comparison between MultiAdam and L-BFGS** . Here we compare the new MUltiAdam optimizer for PINNs with L-BFGS, which is a frequently used optimizer in PINN variants. The L2Re result is listed in the Table E.2. We see that L-BFGS does not converge in many cases as it is unstable while MultiAdam has a better convergence property. However, L-BFGS achieves better accuracy on some of the problems like high dimensional PDEs.

**Temporal error analysis** For time-dependent problems, an important metric is the generalization ability along the time dimension. We selected Heat2d-CG, Heat2d-MS, and Wave1d-C with two different parameters (domain scale is 2 and 8) to observe how the error evolves over time. We found that the error accumulation over time varies depending on the specific PDE problem. For instance, in the case of Heat2d-CG, its final state is a relatively easy steady state, which results in a gradual reduction of error over time. On the other hand, for Heat2d-MS, the solution continuously oscillates, leading to an increasing error as time progresses. In the case of Wave1d-C, due to the periodic nature of the wave equation and the presence of a ground truth solution that is entirely zero, we observed the L2 Relative Error (L2RE) also increases with fluctuations. In summary, error accumulation in time-dependent problems remains challenging for PINNs, necessitating deeper analysis and improved optimization methods in future research.

**Runtime analysis** The runtime results for different methods are shown in Table 14. We have analyzed the results in the previous section.

# F    Other visualization results and analysis

Here we list some visualization results of these experiments. We see that Burgers1d, Poisson2d-C, Poisson2d-CG, and NS2d-C could be solved with a relatively low error. Other problems are difficult to learn, even the approximate shape of the solution. Here we only visualize two-dimensional cases,

| $\alpha$ | Burgers1d | GS | Heat2d-CG | Poisson2d-C |
|---|---|---|---|---|
| 0.2 | 9.88E-2(1.75E-2) | 8.57E-2(3.14E-3) | 1.05E+0(1.68E-1) | 5.81E-1(1.01E-3) |
| 0.4 | 9.01E-2(1.43E-2) | 8.09E-2(7.63E-4) | 7.36E-1(7.23E-2) | 2.85E-1(9.30E-2) |
| 0.6 | 1.75E-1(7.97E-2) | 7.95E-2(6.30E-4) | 6.79E-1(1.17E-1) | 5.54E-2(1.23E-3) |
| 0.8 | 1.61E-1(1.08E-1) | 8.04E-2(1.03E-3) | 6.96E-1(1.50E-1) | 4.19E-2(4.71E-3) |

Table 26: Performance (L2RE) comparison of FBPINN with different overlap ratios $\alpha$.

| Scale $L$ | Adam | MultiAdam | LRA | GePinn |
|---|---|---|---|---|
| 0.5 | 6.94E-1(1.76E-2) | **5.71E-1(6.11E-2)** | 6.93E-1(1.48E-2) | 7.06E-1(2.94E-3) |
| 1 | 6.92E-1(1.79E-2) | **3.56E-2(1.25E-2)** | 3.88E-1(2.61E-1) | 6.89E-1(1.41E-2) |
| 2 | 4.41E-1(9.57E-2) | **3.81E-2(9.38E-3)** | 1.68E-1(6.78E-2) | 6.76E-1(3.86E-2) |
| 4 | **1.77E-2(4.66E-3)** | 3.38E-2(9.71E-3) | 1.11E-1(1.43E-1) | 3.13E-2(2.85E-3) |
| 8 | 2.39E-2(7.26E-3) | 4.40E-2(3.07E-2) | 1.41E-1(7.10E-2) | **1.95E-2(6.42E-3)** |
| 16 | 1.83E-2(8.19E-3) | 3.62E-2(1.10E-2) | 9.45E-2(2.05E-2) | **1.59E-2(6.03E-3)** |

Table 27: Performance comparison of vanilla PINNs, Multi-Adam, PINN-LRA, and gPINN on Poisson2d-C different domain scales.

| L2RE | | MultiAdam | L-BFGS |
|---|---|---|---|
| Burgers | 1d-C | 4.85E-2(1.61E-2) | **1.33E-2(5.30E-5)** |
| | 2d-C | **3.33E-1(8.65E-3)** | 4.65E-1(4.69E-3) |
| Poisson | 2d-C | **2.63E-2(6.57E-3)** | NaN |
| | 2d-CG | **2.76E-1(1.03E-1)** | 2.96E-1(4.77E-1) |
| | 3d-CG | 3.64E+0(2.74E-2) | 3.51E+0(9.33E-2) |
| | 2d-MS | **5.90E-1(4.06E-2)** | 1.45E+0(4.75E-3) |
| Heat | 2d-VC | 4.75E-1(8.44E-2) | **2.32E-1(5.29E-3)** |
| | 2d-MS | 2.18E-1(9.26E-2) | **1.73E-2(4.74E-3)** |
| | 2d-CG | **7.12E-2(1.30E-2)** | 8.57E-1(6.69E-4) |
| | 2d-LT | 1.00E+0(3.85E-5) | 1.00E+0(6.69E-5) |
| NS | 2d-C | 7.27E-1(1.95E-1) | **2.14E-1(1.07E-3)** |
| | 2d-CG | **4.31E-1(6.95E-2)** | NaN |
| | 2d-LT | 1.00E+0(2.19E-4) | 9.70E-1(3.66E-4) |
| Wave | 1d-C | **1.21E-1(1.76E-2)** | NaN |
| | 2d-CG | 1.09E+0(1.24E-1) | 1.33E+0(2.34E-1) |
| | 2d-MS | 9.33E-1(1.26E-2) | NaN |
| Chaotic | GS | **9.37E-2(1.21E-5)** | NaN |
| | KS | 9.61E-1(4.77E-3) | NaN |
| High dim | PNd | 3.98E-3(1.11E-3) | **4.67E-4(7.12E-5)** |
| | HNd | 3.02E-1(4.07E-2) | **1.19E-4(4.01E-6)** |

Table 28: Mean L2RE comparison between MultiAdam and L-BFGS.

| L2RE | – | Burgers-P | Poisson-P | Heat-P | NS-P | Wave-P | High dim-P |
|---|---|---|---|---|---|---|---|
| Name | – | 2d-C | 2d-C | 2d-MS | 2d-C | 1d-C | HNd |
| Vanilla | PINN | 4.74E-1(1.93E-1) | 1.73E-1(2.40E-1) | 7.66E-3(3.61E-3) | 3.89E-1(4.40E-1) | 2.24E-1(3.03E-1) | 5.22E-1(3.56E-2) |
| Reweighting | LRA | 4.36E-1(1.99E-1) | 1.23E-1(1.56E-1) | 6.53E-3(6.12E-3) | 0.00E+0(0.00E+0) | 7.07E-2(1.14E-1) | 3.44E-1(1.81E-1) |
| | NTK | **4.13E-1(1.82E-1)** | **1.50E-1(1.86E-1)** | 9.04E-3(6.52E-3) | 4.52E-1(3.01E-1) | **1.66E-2(4.52E-3)** | 2.69E-1(1.88E-1) |
| Sampling | RAR | 4.71E-1(1.98E-1) | 1.53E-1(2.11E-1) | 8.07E-3(1.75E-3) | 3.91E-1(4.46E-1) | 2.33E-1(3.10E-1) | 5.05E-1(6.10E-2) |
| Optimizer | MultiAdam | 4.93E-1(1.94E-1) | 4.00E-1(3.20E-1) | 2.22E-3(1.55E-3) | 9.33E-1(4.32E-2) | 8.24E-2(9.22E-2) | **6.89E-1(8.46E-2)** |
| Loss functions | gPINN | 4.91E-1(2.01E-1) | 4.59E-1(4.57E-1) | 7.87E-3(2.82E-3) | 7.19E-1(2.89E-1) | 4.03E-1(3.44E-1) | 7.66E-1(3.30E-2) |
| | vPINN | 2.82E+0(1.79E+0) | 5.12E-1(2.43E-1) | – | 3.76E-1(6.90E-2) | 5.51E-1(6.09E-1) | – |
| Architecture | LAAF | 4.37E-1(1.77E-1) | 6.27E-2(4.65E-2) | **6.97E-3(5.23E-3)** | 3.63E-1(4.38E-1) | 1.84E-1(2.91E-1) | 4.03E-1(1.27E-1) |
| | GAAF | 4.34E-1(1.85E-1) | 1.89E-1(2.54E-1) | 1.94E-1(8.63E-2) | 4.85E-1(4.09E-1) | 2.97E-1(2.38E-1) | 9.00E-1(1.68E-1) |
| | FBPINN | – | 2.46E-1(4.50E-1) | – | 3.99E-1(2.97E-1) | 2.87E-2(2.81E-2) | 1.15E+0(1.06E+0) |

Table 29: L2RE (mean/std) of different methods on parametric experiments.

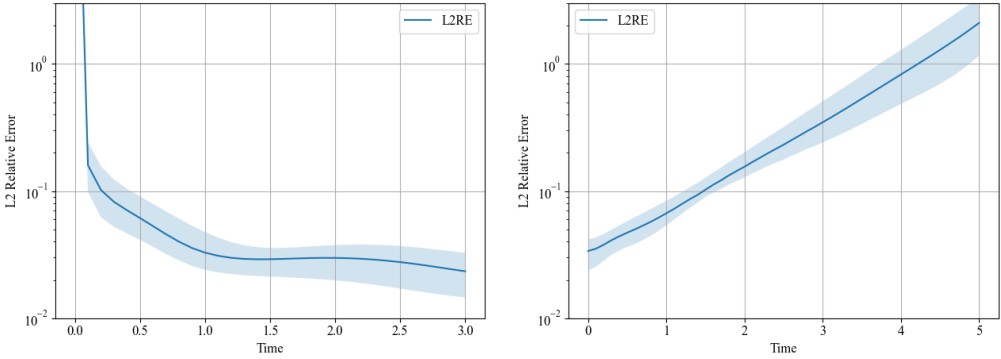

Figure 20: L2RE varying with time for PINNs on Heat2d-CG, Heat2d-MS.

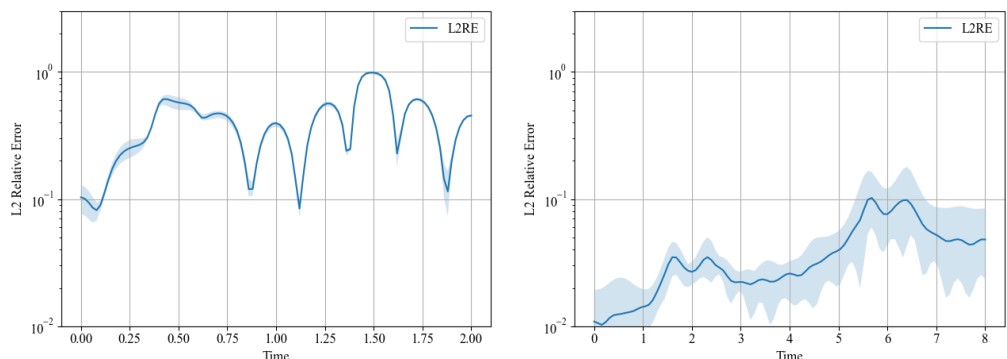

Figure 21: L2RE varying with time for PINNs on Wave1d-C-scale2 and Wave1d-C-Scale8.

which are easier to display in the paper. Note that we also support different forms of three-dimensional plot functionals in our code.

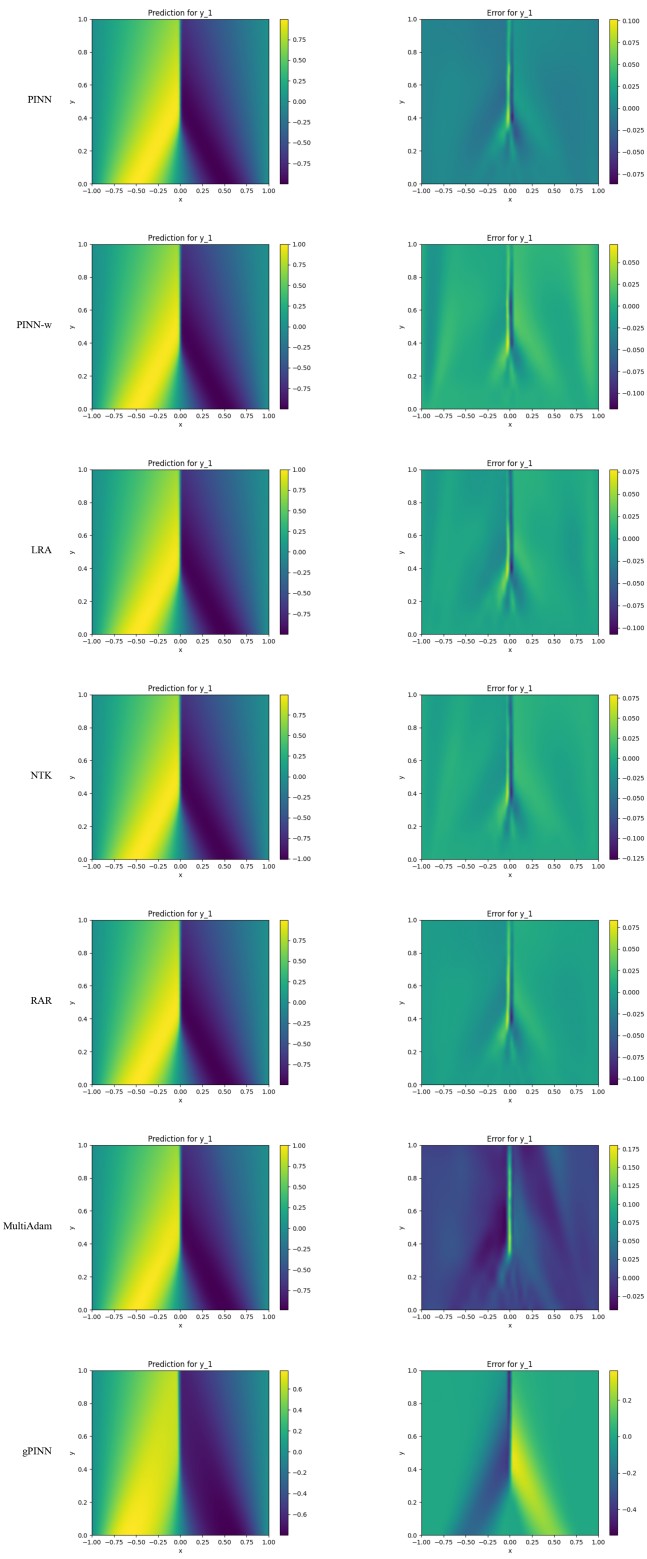

Figure 22: Visualization of Burgers1d. The left pictures are the prediction of PINN methods. The right pictures show the error between the prediction and the ground truth.

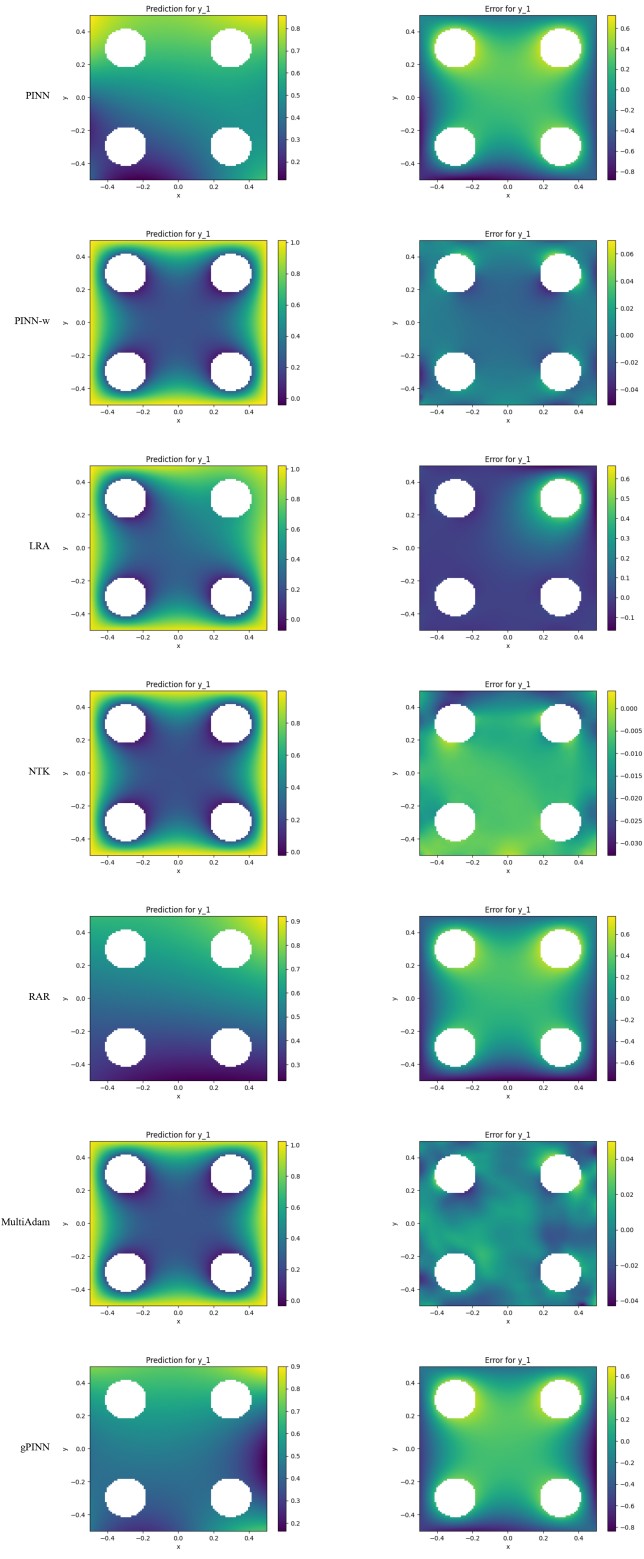

Figure 23: Visualization of Poisson2d-C. The left pictures are the prediction of PINN methods. The right pictures show the error between the prediction and the ground truth.

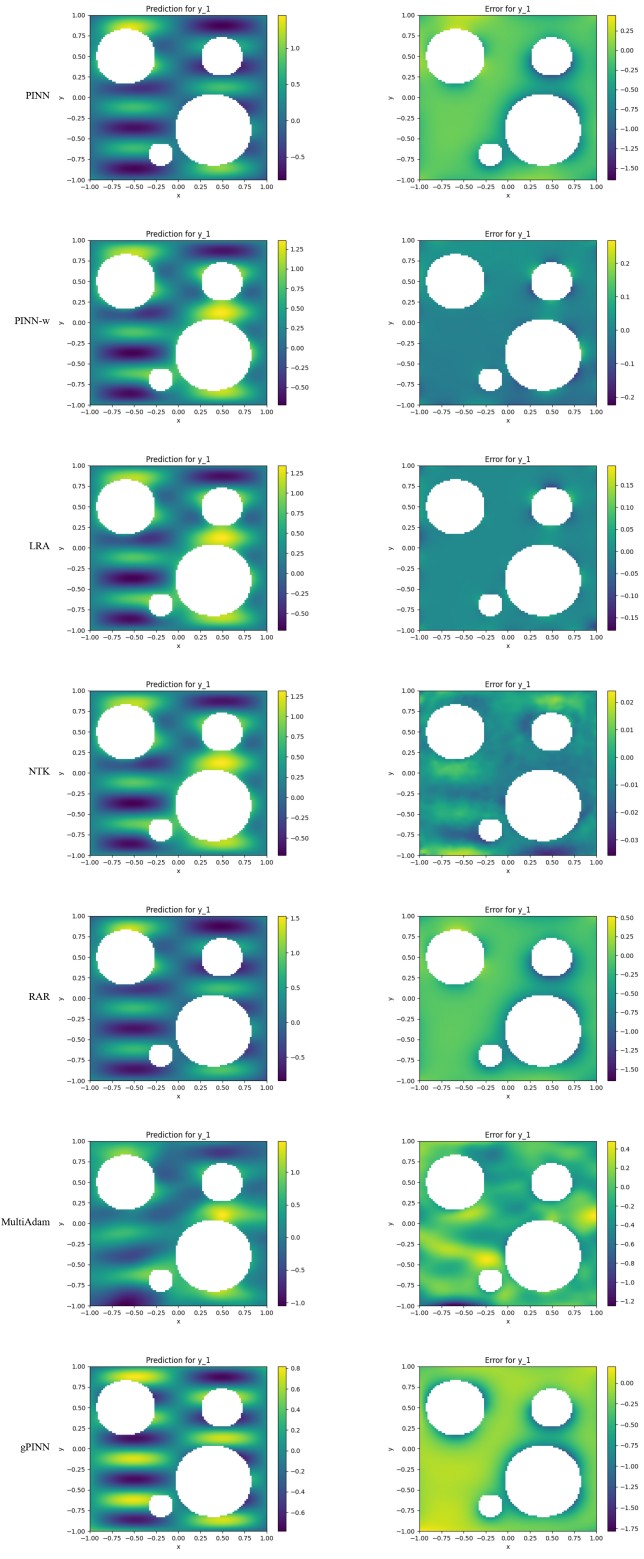

Figure 24: Visualization of Poisson2d-CG. The left pictures are the prediction of PINN methods. The right pictures show the error between the prediction and the ground truth.

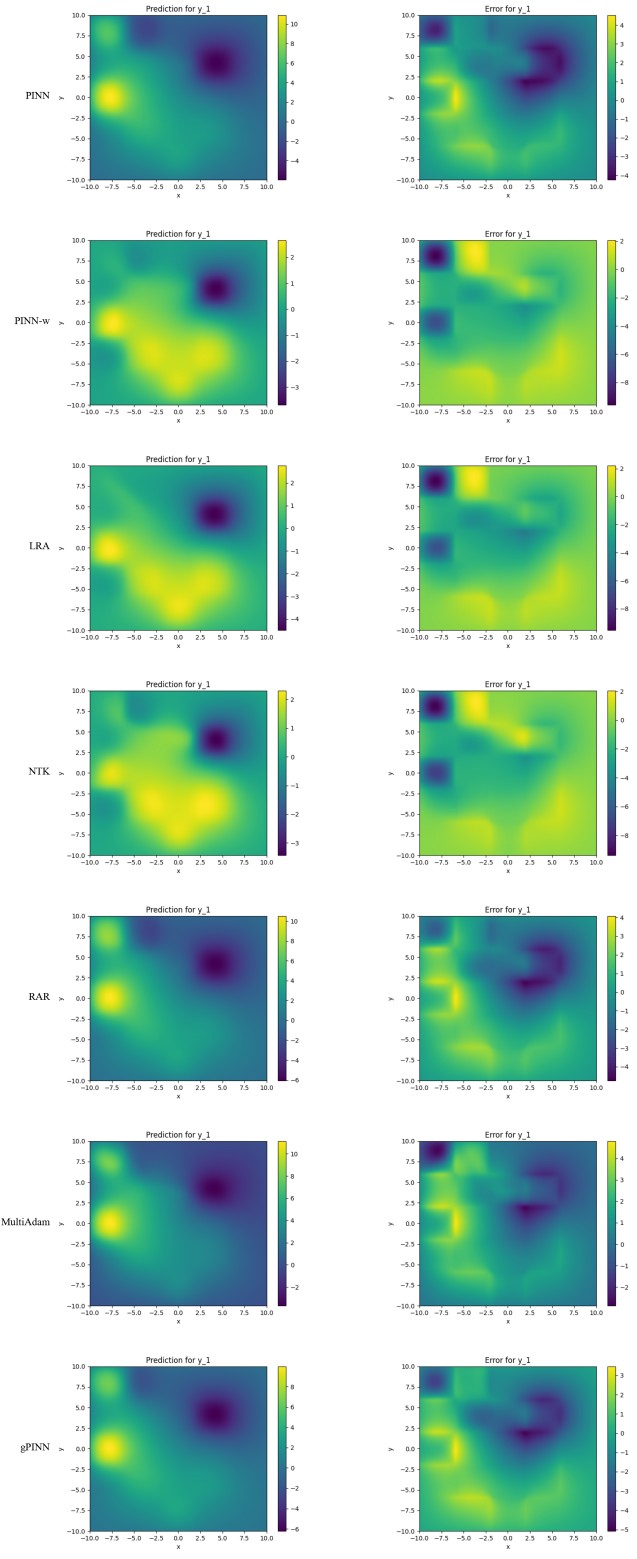

Figure 25: Visualization of Poisson2d-MS. The left pictures are the prediction of PINN methods. The right pictures show the error between the prediction and the ground truth.

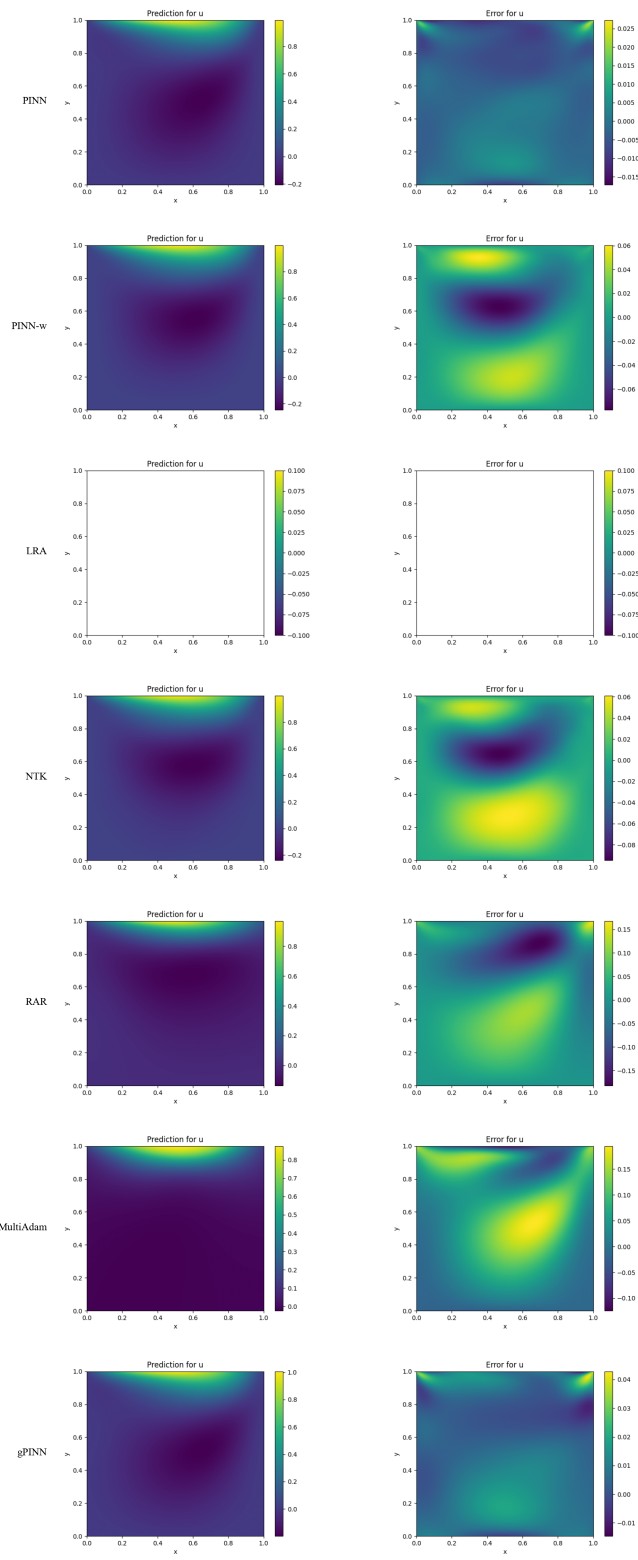

Figure 26: Visualization of NS2d-C. The left pictures are the prediction of PINN methods. The right pictures show the error between the prediction and the ground truth. Note that PINN-LRA diverged in this case.

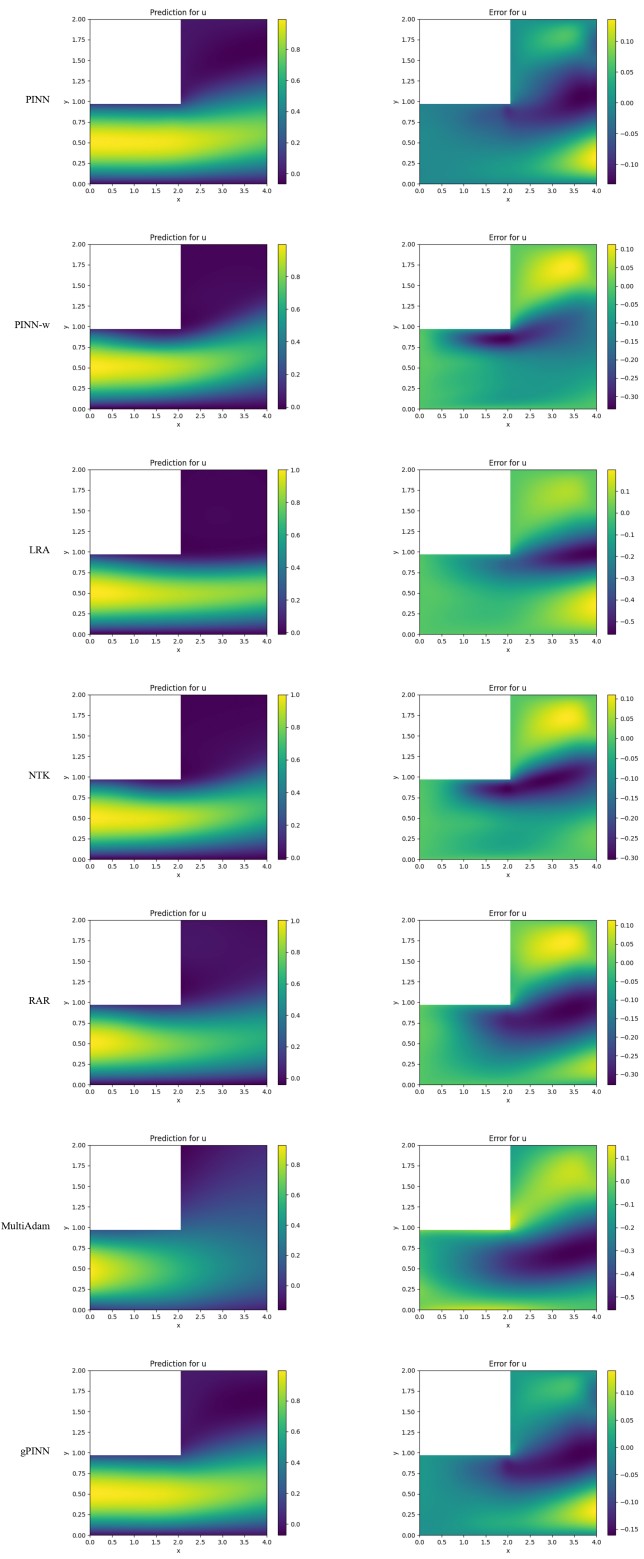

Figure 27: Visualization of NS2d-CG. The left pictures are the prediction of PINN methods. The right pictures show the error between the prediction and the ground truth.

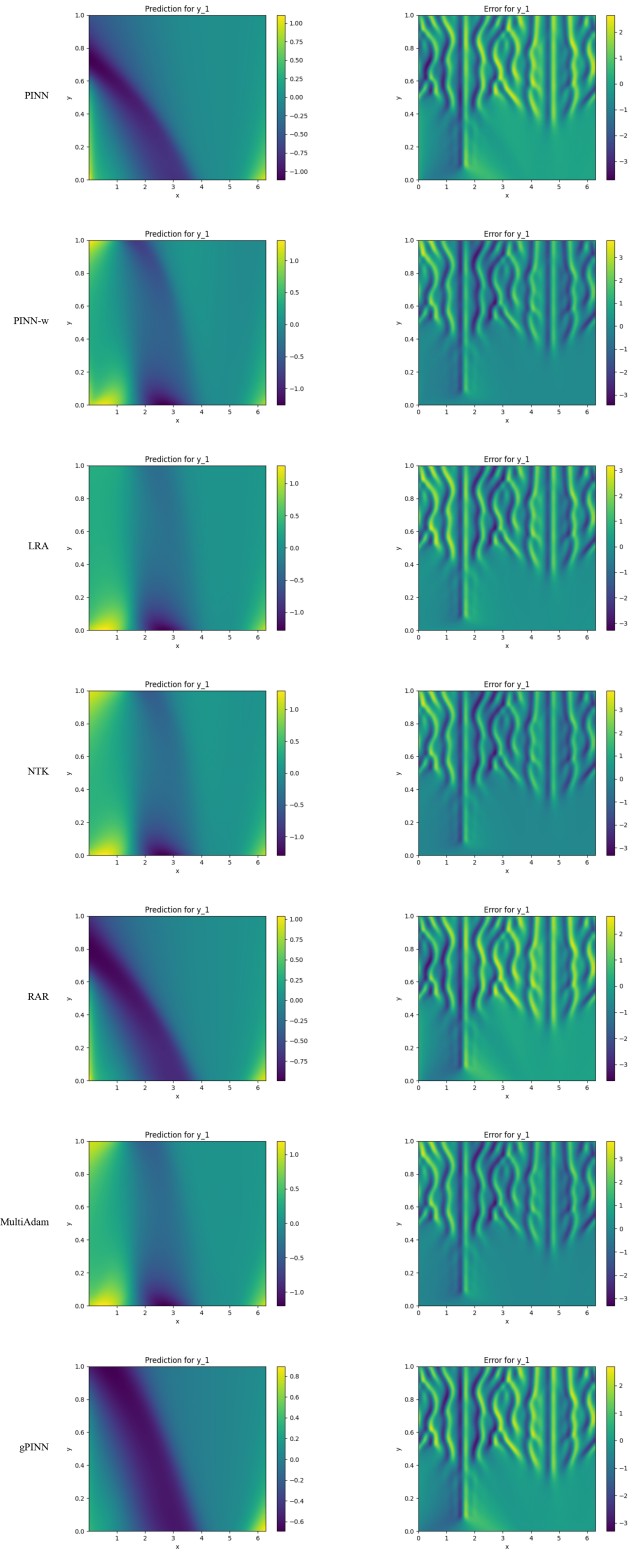

Figure 28: Visualization of KS. The left pictures are the prediction of PINN methods. The right pictures show the error between the prediction and the ground truth.

