# OpenReview forum: "PINNacle: A Comprehensive Benchmark of Physics-Informed Neural Networks for Solving PDEs"
_NeurIPS.cc/2024/Datasets_and_Benchmarks_Track — NeurIPS 2024 Track Datasets and Benchmarks Poster_

### Official Review · Reviewer_unji · 2024-07-23
**Initial review**

**Rating:** 6
**Confidence:** 4
**Correctness:** Yes.
**Clarity:** Yes.

**Review:**

**Quality:** This paper includes many types of PDEs and many PINN methods. The overall quality is good.

**Clarity:** This paper is well-clarified. The details and documents of the dataset are good.

**Originality:** The dataset itself is original, but there are many similar existing works, such as PDEBench and PDEArena. The authors provide a section in the Supplementary to differentiate the proposed work and the existing benchmark works.

**Significance:** This benchmark focuses on the PINN family. The significance seems limited.

**Strengths:**

- This paper is well-documented.

- More than 20 PDE problems are defined and over 10 PINN variants have been tested.

- Comprehensive experiments have been conducted to test the baseline models on the proposed datasets.

**Additional Feedback:**

- In Lines 308-309, this paper claims that the loss landscape of PINN methods is largely unexplored. The failure modes of PINNs have been investigated through the lens of loss landscape in [1] (also Ref [28] of the main text).

- In Lines 487-489, there is an existing paper [2] that has provided the theoretical convergence for PINNs.

**Refs:**

[1] Krishnapriyan, Aditi, et al. "Characterizing possible failure modes in physics-informed neural networks." Advances in neural information processing systems 34 (2021): 26548-26560.

[2] Gao, Yihang, Yiqi Gu, and Michael Ng. "Gradient descent finds the global optima of two-layer physics-informed neural networks." International Conference on Machine Learning. PMLR, 2023.

**Documentation:**

Yes.

**Ethics:**

No.

**Limitations:**

Yes.

**Opportunities For Improvement:**

- I feel that the scope of this paper can be broader, not just limited to PINN methods. PINN methods are a subset of the PDE learning. The impact might be limited if only testing PINNs. Since some of the datasets have quite different geometries, all mesh-free methods might be quite suitable. Therefore, including neural operator-type models might also be ok, which might broaden the application spectrum of the proposed benchmark. However, the concern is that this paper needs to think about more differences compared to PDEBench [1] and PDEArena [2] if aiming for general PDE learning.

- There are different types of PDE datasets, which is good. However, I think it would be better to include some datasets from the same PDE type but with different PDE coefficients, like what PDEBench has done. The failure models issue of PINNs [3] is challenging. Showing the performance of PINN variants on different PDE coefficients would give readers a better sense of the current situation of physics-constrained ML.

---

**Refs:**

[1] Takamoto, Makoto, et al. "Pdebench: An extensive benchmark for scientific machine learning." Advances in Neural Information Processing Systems 35 (2022): 1596-1611.

[2] Gupta, Jayesh K., and Johannes Brandstetter. "Towards multi-spatiotemporal-scale generalized pde modeling." arXiv preprint arXiv:2209.15616 (2022).

[3] Krishnapriyan, Aditi, et al. "Characterizing possible failure modes in physics-informed neural networks." Advances in neural information processing systems 34 (2021): 26548-26560.

**Relation To Prior Work:**

Yes.

**Summary And Contributions:**

This paper provides a benchmark dataset for the PINN family. More than 20 PDE problems are defined and over 10 PINN variants have been tested. Extensive experiments have been done to evaluate the baseline performance.

---

> ### Author Rebuttal · Authors · 2024-08-14
>
> Dear Reviewer unji,
>
> Thank you for your valuable feedback and suggestions on improving our paper. Here we provide responses to your questions.
>
> **Q1.** The scope of this paper can be broader, not just limited to PINN methods.
>
> **A1.** While both PINNs and neural operators use neural networks to solve PDEs, the problems they address are quite different. Additionally, the field of PINNs is evolving rapidly, with numerous methods being proposed, making it essential to distinguish advancements. We agree that solely comparing different PINN approaches may have limitations. However, directly comparing PINNs with neural operators which are two fundamentally different methods makes it difficult to draw meaningful conclusions. For example, in PDEBench, a very simple comparison was made, showing that with a large amount of data, PINNs struggle to outperform neural operator methods. While in the cases when there is only PDE form available, neural operators fail to work.
>
> **Q2.**  Add some datasets from the same PDE type but with different PDE coefficients, like PDEBench.
>
> **A2.** Thank you for your suggestion. We have already included parameterized PDE experiments in the main experiment (**Table 4**). In **Appendix B.2**, we provide a detailed explanation of how different parameters (e.g. parameters in Poisson's equation) or initial conditions (e.g. Burgers' equation) are selected within the same class of PDEs to comprehensively test the effectiveness of PINNs on these types of PDEs.
>
> We will deeply appreciate if you find our response helpful! We will try our best to improve our paper if you have further questions!
>
> best regards,
>
> authors

---

> > ### Author Rebuttal · Authors · 2024-08-15
> >
> > **Q3.** Additional feedback and references.
> >
> >
> > **A3.** Thank you for providing the references. We have addressed the issue of the PINN loss landscape in the revised version and have cited the relevant papers accordingly.

---

### Official Review · Reviewer_dS4W · 2024-07-24
**Review on PINNacle, a comprehensive PINN benchmark**

**Rating:** 6
**Confidence:** 3
**Clarity:** The paper is very clearly written.

**Review:**

This benchmark is a great attempt at bringing together a complete package for developing PINNs. It contains a multitude of equations, testing different phenomena, error metrics and a good collection of included methods. The organization is very good and it even contains a sandbox to write PINNs. While not the first benchmark, its comprehensiveness is really impressive. There are only a few areas for improvement, which are listed below.

**Strengths:**

* The dataset contains a good mix of equations relevant to different applications, something which is often overlooked.
* The inclusion of tests on complex geometry and inverse methods are good additions to the other PDE aspects tested and at least in the complex geometry case, often overlooked.
* There is a good discussion and analysis of failure modes of PINNs in different equations, such as long timescale ones.
* Often such benchmarks will contain only a few methods as the focus is really on the data, but here the authors do a good job of really benchmarking a comprehensive list of PINN approaches.

**Additional Feedback:**

N/A

**Correctness:**

The evaluation methods are reasonable and the design of the dataset is very well done. The consideration of different features of PDEs tested such as multi-scale phenomena and nonlinear behavior is very thorough.

**Documentation:**

The documentation seems very nice and easy to use.

**Limitations:**

The authors have adequately discussed limitations.

**Opportunities For Improvement:**

* I think that adding basic traditional methods, such as even finite difference schemes and how they perform, would ground performance of PINNs to reality, make problems of certain PDEs more evident and ultimately be helpful to those working in the space (even if they initially may not look great). The end goal is not to have the best PINN, but to have the best PDE solver.
* The ground truth needs to have some kind of discussion on how it was produced. Was it generated on a much smaller grid spacing? It would be useful to get an error bound on the generated solutions, perhaps even by comparing to the solution with smaller grid spacing and reporting a ground truth upper bound.
* For complex geometry, how is the data represented? Do you have a mesh representation or is the data on a grid? How does this work with PINNS? If the data is on a grid, why use FEM at all, as it boils down to Finite Difference anyway? I think some of these discussions belong in the paper.

**Relation To Prior Work:**

There is a good discussion on the state of the field as far as PINNs go, as well as benchmarks of PDE solvers in general.

**Summary And Contributions:**

This paper presents a benchmark for PINN style methods, specifically addressing 3 main parts:
* A varied dataset of PDEs along with their solutions, tackling common issues in PDE theory such as multiscale phenomena, irregular geometry and nonlinear behavior.
* A comprehensive evaluation of current PINN methods on this dataset, highlighting strengths and drawbacks of each network on the different PDEs.
* A standard set of error metrics.

---

> ### Author Rebuttal · Authors · 2024-08-15
>
> Dear Reviewer ds4W,
>
> Thank you for your valuable feedback and suggestions on improving our paper. Here we provide responses to your questions.
>
> **Q1.**  Adding basic traditional methods, such as even finite difference schemes and how they perform.
>
> **A1.** We agree with your point that our ultimate goal is to find the best PDE solver. However, directly comparing numerical methods with PINNs within our benchmark is somewhat problematic. For most PDEs, our approach uses the finite element method implemented in COMSOL. The solvers and parameters used vary depending on the specific PDE (see supplementary file). These traditional methods can achieve very accurate results for forward problems, but unlike PINNs, a single solver cannot universally solve different types of equations. Therefore, we did not include these methods in our experiments.
>
> **Q2.** The ground truth needs to have some kind of discussion on how it was produced.
>
> **A2.1 (Discussion about ground truth solvers)** In the **supplementary file ** and **Appendix B.3** (the revised version), we have added details about the solver configurations used for these problems and an analysis of the ground truth obtained.  Given that our benchmark includes various types of PDEs, we generated the reference data using different types of numerical solvers, including the FEM/FVM solvers and pseudo spectral methods.
>
> **A2.2 (Mesh convergence analysis)**. In the **supplementary file ** and **Appendix B.4** (the revised version), we conducted a mesh convergence study for two difficult examples, i.e., Poisson3d and NS2d-CG equations using grids with varying spacing. We use  Richardson extrapolation to estimate the error bound. Specifically, if the error reduces as a power of the grid size $h$, the extrapolated solution $u_{\text{extrapolated}}$ can be calculated as:
> $$
> u_{\text{extrapolated}} = \frac{h_2^p u_{h_1} - h_1^p u_{h_2}}{h_2^p - h_1^p}.
> $$
> We use the following equation to estimate the error bound,
> $$
> \text{Error Bound} = \frac{||u_{\text{extrapolated}} - u_{h_2}||}{||u_{\text{extrapolated}}||}.
> $$
> The **Figure 1** in the supplementary file shows the results of the error bound analysis.  The error for the reference data we used was below 0.1%, indicating that it is highly reliable and can serve as a valid reference for the PINN solutions.
>
> **Q3.** For complex geometry, how is the data represented? Do you have a mesh representation or is the data on a grid?
>
> **A3.** For complex geometries, we only need to define a method for sampling points within the geometric domain to solve using PINNs. In practice, we can achieve this using DeepXDE for certain simple geometric operations. So we do not have a mesh representation of the data as the points can be resampled every few epochs. For generating the ground truth, we used the solvers provided by COMSOL, which offer more efficient FEM/FVM methods.
>
> We sincerely thank the reviewers for their feedback again. We hope that our responses have addressed your concerns, and we are more than willing to continue improving the paper if you have any further questions or suggestions.
>
> best regards,
> authors

---

> > ### Comment · Reviewer_dS4W · 2024-09-01
> > **Comment**
> >
> > The authors have addressed questions 2 and 3 adequately. I do believe that some indications of traditional solver performance should go into benchmarks like this (Q1) and the reasons given are largely orthogonal to this. Even so, methods like finite element and even finite difference are used for a large variety of systems so the universality of PINNs is not entirely unique.

---

### Official Review · Reviewer_ffbg · 2024-07-24
**Benchmark for PINNs - good contribution with some shortcomings**

**Rating:** 6
**Confidence:** 4
**Correctness:** I believe the results to be correct.

**Review:**

The paper matches the revised submission to last year's benchmark
track apart from minor changes. For example, the LBFGS results are
new.

As PINNs are tricky to use in practice (they might utterly fail for
related, but not identical problems), there is a zoo of variants of
PINN methods out there, and a benchmark that allows to evaluate their
use for different problems is a valuable contribution. It therefore
sheds some light on the question when to use which variant.

An extensive appendix complements the work. Unfortunately, only the
appendix answers some of the questions that are to be answered in the
main paper according to the track's requirements (see opportunities
for improvement below).

Correspondingly, the required (URL to) Croissant metadata record could
not be found.

**Strengths:**

* A comprehensive list of results benchmarking different PINN variants
  (if including the appendix).
* Good descriptions in the appendix.
* An easy to use Python toolbox to reproduce the results.

**Additional Feedback:**

See above.

What do the  x/v entries in Table 1 tell me?

**Clarity:**

The paper is in general well written.

* line 206: framework To -> framework.
* Overivew (caption table 2)
* Please check spelling (captial letters) in the references; in particular for names and acronyms (such as pde -> PDE).

**Documentation:**

The documentation to reproduce the problems is good. Though following
the instructions on https://github.com/i207M/PINNacle spoiled my
Python installation. Please replace the Installation instructions with
those that you provide on
https://pinnacle-docs.vercel.app/start/install . A virtual environment
is really crucial.

The documentation to extend / run own experiments or PINN variants,
is, however, insufficient. The API is not well-documented,
e.g. https://pinnacle-docs.vercel.app/API/pde .

**Ethics:**

No ethical concerns

**Limitations:**

* Limitations are discussed, but not within the page limit.

**Opportunities For Improvement:**

* According to the requirements, the paper should discuss limitations
  of the proposed approach. Chapter 6, however, is sneaked in before
  the checklist and after the end of the official paper.
* Similiarly, the paper itself should provide information, according to the
  checklist, about the hardware used and the computational ressources
  spent. The former one is only in the appendix, the latter one
  missing.
* In the main paper (table 3), LBFGS and RAR had been added compared
  to previous versions. The runtime results and FLOPs are, however,
  missing in the appendix (e.g. tables 12 and 13).
* Documentation (see below)
* Reading the paper, I had the impression that one of the claims is to
  be the first to provide a set of PDE problems. It would be great if
  the main paper would have a table highlighting which of the
  established SimML benchmarks covers which of the problems discussed
  here. Or, in other words, to specify clearly what is the novel
  contribution in this regard.

**Relation To Prior Work:**

Prior work is addressed in Section 2 and related to the present work -
apart from the selection of datasets/problems (see improvements above).

**Summary And Contributions:**

The authors provide a tool to benchmark several PINN variants against
a set of 8 different PDE problems (some in several variants), and they
provide datasets for the latter. The PDEs cover a range of typical
problems such as nonlinearities, non-trivial geometries, and different
dimensionalities.

The paper furthermore shows benchmark results. In the
assessment, some characteristics of PDEs that are difficult
to learn with PINNs are identified.

The Python tool provided allows users in an easy way to reproduce the
experiments and to reproduce the benchmark results.

---

> ### Author Rebuttal · Authors · 2024-08-16
>
> Dear Reviewer ffbg,
>
> Thank you for your valuable feedback and suggestions on improving our paper. Here we provide responses to your questions.
>
> **Q1.** The paper should discuss limitations of the proposed approach.
>
> **A1.** Thank you for the reminder.  In the revised version, we have moved the limitations to the section before the conclusion. Due to space constraints, we have used more concise language to describe the limitations. Following NeurIPS policy, we are unable to upload the revised full text directly.  **Modified part of limitations**:  First, real-world problems are often more complex, with giant geometric domains or chaotic behaviors. Good performance on PINNacle does not guarantee it solves practical problems. We could explore larger-scale PINN training methods or efficient domain decomposition methods. Second, the issues of safety in the PINN methods pose potential roadblocks. Developing theoretical convergence for PINN like stability and convergence analysis \cite{eiras2023provably} could help resolve these limitations.
>
> **Q2.** The paper itself should provide information, according to the checklist, about the hardware used and the computational ressources spent.
>
> **A2.** We have updated **Appendix E** in the paper and provided additional details regarding the GPU resources required for the primary experiments in the paper.  The total computational cost for the paper is as follows and is also updated in the revised paper:
>
> **We run all experiments on a Linux server with 20 Intel(R) Xeon(R) Silver 4210 CPUs @ 2.20GHz and eight NVIDIA GeForce RTX 2080 Ti each with 12 GB GPU memory. All experiments in Table 3 require a total about 776 GPU hours, which can be completed in about 4 days on our cluster.**
>
> **Q3.**  The runtime/FLOPs results of LBFGS and RAR are missing in the appendix.
>
> **A3.** Thank you for pointing out this oversight. We have added the results for these two methods to the corresponding tables, **Table 12** and **Table 13** in the revised paper.
>
> **Q4.**   Installation instructions causing issues with Python packages.
>
> **A4.**  We have updated our GitHub README to highlight the importance of using a virtual environment and provide revised installation instructions.
>
> **Q5.** The documentation to extend experiments is insufficient.
>
> **A5.**  We have added detailed documentation for extending PDE problems at https://pinnacle.scientific.ml/pde/create.
>
> **Q6.** The required (URL to) Croissant metadata record could not be found.
>
> **A6.** Our work primarily focuses on a PINN benchmark, where the datasets consist only of reference solutions. This differs significantly from datasets in other machine learning fields, as our reference data is generated by solvers and is used solely to evaluate the quality of the solutions. Additionally, the scale of this data is quite small. Therefore, we did not treat the reference data as a standard dataset but rather as part of the toolbox.
>
> **Q7.** The API documentation is not well-documented.
>
> **A7.** The API section is intended to provide a quick reference and is automatically generated to show class hierarchies and methods. For detailed explanations, please refer to the Trainer/PDE/Method/Callback sections in our documentation.
>
> **Q8.**  It would be great if the main paper would have a table highlighting which of the established SimML benchmarks covers which of the problems discussed here.
>
> **A8.** In the **supplementary file** and the revised version of **Appendix B.3**, we have added the following **Table** summarizing the equations used in current PDE-related benchmark work. Additionally, we have described the relationship between our work and datasets/benchmarks such as PDEBench and PDEArena.
>
> | PDE type/Number of PDEs | PINNacle | PDEBench | PDEArena | Wang et. al\cite{wang2023expert} |
> | :---------------------: | :------: | :------: | :------: | :------------------------------: |
> |         Burgers         |    2     |    1     |    0     |                0                 |
> |         Poisson         |    4     |    1     |    0     |                0                 |
> |  Convection-Diffusion   |    4     |    3     |    0     |                2                 |
> |      Shallow water      |    0     |    1     |    1     |                0                 |
> |      Naiver-Stokes      |    3     |    4     |    2     |                4                 |
> |          Wave           |    2     |    0     |    0     |                0                 |
> |         Chaotic         |    2     |    0     |    0     |                1                 |
> |        High dim         |    2     |    0     |    0     |                0                 |
> |    Inverse problems     |    2     |    0     |    0     |                0                 |
>
> **Q9.** Typos and minor mistakes in the paper.
>
> **A9.** Thank you for pointing out these details. In the revised version, we've corrected these typos and issues related to references.
>
> **Q10.** The meaning of x/v entries in Table 1.
>
> **A10.** Table 1 highlights whether a particular equation in the benchmark exhibits certain challenges or phenomena we've identified, such as complex geometry or strong nonlinearity. The purpose of this table is to provide a clear, visual overview of the main difficulties associated with different types of PDE problems.
>
> We sincerely thank the reviewers for their feedback again. We hope that our responses have addressed your concerns, and we are more than willing to continue improving the paper if you have any further questions or suggestions.
>
> best regards,
>
> authors

---

> > ### Comment · Reviewer_ffbg · 2024-08-31
> >
> > The authors have addressed my concerns well enough. Though for several additions, they refer to the appendix. But, for example, a comparison with the state of the art (Q8) should be in the main text, not in the appendix.
> >
> > If the other reviewers are happy with the rebuttal, too, I would be willing to increase my rating by one.

---

### Official Review · Reviewer_uVbz · 2024-07-31
**A comprehensive PINN dataset and benchmark tool**

**Rating:** 7
**Confidence:** 3
**Correctness:** Good.
**Clarity:** Good.

**Review:**

See Strengths and Opportunities For Improvement.

**Strengths:**

- The paper is well written.
- The paper considers a wide range of PDEs and PINN variants.
- The paper provides a complete flow from data to implementation.

**Additional Feedback:**

None,

**Documentation:**

The documentation is sufficient.

**Ethics:**

None.

**Limitations:**

The authors addressed limitations adequately.

**Opportunities For Improvement:**

- If I understand correctly, although the authors also showed the results for parametric PDEs, in the main results each type of PDE only considers 1 initial condition. Is this setting practical? For example, the evaluation on one particular initial condition may not be able to represent the entire PDE class. I wonder how other PDE benchmarks handle this issue.

- Minor: there could be some potential confusion with the headers in Table 3 and 4. E.g., why there is a line between PINN-2 and LBFGS in table 3, and why there is not a line between PINN and LRA in table 4?

**Relation To Prior Work:**

Good.

**Summary And Contributions:**

The paper introduces PINNacle, a dataset and benchmark for PINNs. The proposed dataset provides a wide range of applications and PINN variants.

---

> ### Author Rebuttal · Authors · 2024-08-14
>
> Dear Reviewer uVbz,
>
> Thank you for your valuable feedback and suggestions on improving our paper. Here we provide responses to your questions.
>
> **Q1.** In the main results of parametrized PDEs, each type of PDE only considers 1 initial condition.
>
> **A1.**  For time-dependent PDEs like Burgers' equation, we selected multiple different initial values for the parametrized PDE experiments. We selected different parameterization methods for different classes of PDEs according to their types and characteristic(as detailed in **Appendix B**). For some steady-state problems, we chose different boundary conditions or parameters; and for higher-dimensional problems, we varied the dimensions, among other factors. Therefore, our parameterization experiments considered various initial conditions during testing.
>
> **Q2.** Some confusion with the headers in Table 3 and 4.
>
> **A2.** Thank you for pointing out this typo. We have corrected these issues in the revised version and have checked the other tables.
>
>  We will deeply appreciate if you find our response helpful!
>
> best regards,
>
> authors

---

### Decision · Program_Chairs · 2024-09-26

**Decision:**

Accept (Poster)

**Comment:**

This paper was reviewed by four experts and all of them are positive about the work to some extent. Thus an accept is recommended.